# Defective HNF4alpha-dependent gene expression as a driver of hepatocellular failure in alcoholic hepatitis

Josepmaria Argemi ⓘ et al.[#]

Alcoholic hepatitis (AH) is a life-threatening condition characterized by profound hepatocellular dysfunction for which targeted treatments are urgently needed. Identification of molecular drivers is hampered by the lack of suitable animal models. By performing RNA sequencing in livers from patients with different phenotypes of alcohol-related liver disease (ALD), we show that development of AH is characterized by defective activity of liver-enriched transcription factors (LETFs). TGF$\beta$1 is a key upstream transcriptome regulator in AH and induces the use of HNF4$\alpha$ P2 promoter in hepatocytes, which results in defective metabolic and synthetic functions. Gene polymorphisms in LETFs including HNF4$\alpha$ are not associated with the development of AH. In contrast, epigenetic studies show that AH livers have profound changes in DNA methylation state and chromatin remodeling, affecting HNF4$\alpha$-dependent gene expression. We conclude that targeting TGF$\beta$1 and epigenetic drivers that modulate HNF4$\alpha$-dependent gene expression could be beneficial to improve hepatocellular function in patients with AH.

Liver-related mortality has increased in the last decade, partially due to the higher incidence of addictions in the form of alcohol-related cirrhosis[1–3]. The prognosis of ALD depends on the development of liver failure, mainly in the form of AH[4]. The burden of AH has increased in many countries and represents an important public health problem[2,5]. The genetic and epigenetic factors involved in the development of AH in heavy drinkers are not well known[6]. GWAS studies have shown that variations in *PNPLA3*, *MBOAT7*, and *TM6SF2* loci confer risk for alcohol-related cirrhosis[7], but the association of specific loci with AH is unknown. Because alcohol abuse has been associated with DNA methylation changes in humans[8,9] and epigenetic dysregulation in experimental liver injury[10,11], it is conceivable that epigenetic factors play a role in AH. Liver failure in the setting of AH was traditionally considered to be secondary to a flare in intrahepatic inflammation[12]. Consequently, therapies have been directed towards decreasing inflammatory mediators (i.e. prednisolone), with limited efficacy[13]. We recently showed that bilirubinostasis, inefficient regeneration of hepatocytes and a compensatory ductular reaction may play a pathogenic role in AH[14–17]. However, the mechanisms of liver failure in the setting of AH remain obscure. This human-based translational study combined integrated multi-OMICs from a large cohort of human samples along with in vitro and experimental animal models with a goal to address this knowledge gap (see "Data Analysis Workflow" in Supplementary Fig 1). In the present work, we describe that livers from patients with AH undergo profound transcriptomic reprogramming, with downregulation of HNF4α and other LETFs. We detect the expression of a fetal isoform of HNF4α in these patients and describe the epigenetic landscape of HNF4a dependent transcriptome. We propose TGFβ1-mediated HNF4α de-regulation and the epigenetic changes in HNF4α-depending genes as potential new therapeutic avenues to treat this lethal disease.

## Results and discussion

**Patients with AH undergo deep transcriptional reprogramming.** In order to uncover the mechanisms involved in progression to AH in patients with ALD, we first performed a comprehensive analysis of liver RNA sequencing (RNA-seq) data from a large series of patients ($N = 92$) with different disease stages including normal liver (Normal, $N = 10$), early alcoholic steatohepatitis (ASH, $N = 12$), AH with liver failure (AH, $N = 18$) and a unique set of explants from patients with AH that underwent urgent liver transplantation (exAH, $N = 10$)[13] (Fig. 1a). As diseased controls, we included patients with non-alcoholic fatty liver disease (NAFLD, $N = 9$), chronic hepatitis C (HCV, $N = 9$) and compensated HCV cirrhosis (CIRR, $N = 9$). The principal component analysis (PCA) showed patient clustering according to the progressive clinical phenotypes (Fig. 1b). Thus, while early ASH clustered along with chronic hepatitis C and NAFLD close to normal livers, patients with AH showed a much more deregulated transcriptome. We then performed a comparative analysis between normal livers and different ALD phenotypes. As shown in Fig. 1c, analytical parameters of liver injury (i.e. AST) and hepatocellular synthetic function (i.e. INR, serum bilirubin and albumin) as well as clinical scoring systems (i.e. Child-Pugh and MELD) were markedly impaired after the onset of AH (Supplementary Table 1). Unbiased clustering and Short Time Expression Miner (STEM) algorithm identified 13 profiles of gene expression across the 4 selected disease stages (Fig. 1d, Supplementary Fig. 2a–d and Supplementary Data 1). These profiles were grouped into 4 main patterns along ALD progression including compensatory transient gene expression changes in early stages, genes progressively up or down-regulated along

disease progression or genes up or down-regulated only after the onset of liver failure (Fig. 1e). Top upstream regulators and target genes belonging to these 4 patterns are depicted in Fig. 1f. A detailed gene set enrichment analysis revealed down-regulation of genes related to basic hepatocyte functions (i.e. metabolism of amino acids and lipids, biological oxidations, mitochondrial function and bile acid metabolism), while cell proliferation, extracellular matrix regulation and inflammation related pathways were enriched among up-regulated genes (Supplementary Fig. 3). Overall, these changes could explain key features in AH including massive fibrosis, proliferation of immature ductular cells and bilirubinostasis[14,15]. In order to gain insight into the main drivers of gene expression that could result in the development of hepatocellular failure in AH, we analyzed the predicted activity of transcription factors using a complementary approach, by combining the search of transcription factor binding motifs in the promoter of differentially expressed genes (DEG) and by the use of Ingenuity Pathway Analysis (IPA) software to uncover predicted upstream transcription factor activity (see Material and Methods section). Early compensated state of ALD was characterized by an increased predicted activity of the hepatoprotective transcription factor PPARγ (Fig. 2a). In contrast, development of AH was associated with a profound decrease in the activity of LETFs, especially HNF4α (Fig. 2b, Supplementary Data 2). To illustrate the transcription factor footprint in Early ASH and in AH, a set of PPARγ and HNF4α target genes are shown in Fig. 2c, d.

The results obtained in human livers were assessed in several animal models of early and advanced ALD using the same approach to infer transcription factor activity from liver RNA sequencing (Supplementary Fig. 4a). Mice subjected to early experimental ALD (High Fat Diet -HFD- plus intragastric Ethanol administration -EtOH- for 3 weeks) showed increased liver damage and hepatocyte steatosis in the absence of significant fibrosis (Supplementary Fig. 4b,c). In these mice, we found a marked predicted activation of PPARγ resembling our analyses in patients with early stages of ALD (Supplementary Fig. 4d). Mice subjected to the model of severe ALD ($CCl_4$ for 9 weeks and then EtOH after a wash-up period) showed increased liver damage (Supplementary Fig. 4b, c) and pericellular ("chicken-wire") fibrosis similar to the findings that we described in humans (Supplementary Fig. 4d, e)[14] but without the parameters of liver failure (i.e. jaundice, coagulopathy). The transcription factor predicted activity analysis was characterized by decreased FOXA-1, but not HNF4α (Supplementary Fig. 4f). Interestingly, while the expression of some HNF4α target genes was decreased in these mice (i.e. PCK1), other well described targets were increased (i.e. coagulation factor VII F7) (Supplementary Fig. 4g). The relatively preserved HNF4α-dependent gene expression could partially explain why these mice do not develop liver failure. It is therefore plausible that manipulating HNF4α could favor the development of alcohol-induced liver failure in these mice. These results could be beneficial in developing a useful preclinical model of true AH. The fact that HNF4α is still active in these mice could also be partially due to defective TCF3/4 repression activity over HNF4α[18].

**Fetal P2-dependent HNF4α isoforms are increased in AH.** Because HNF4α was the most inhibited LETF found in our analysis of human AH, we decided to focus on its potential role in mediating liver failure in AH. HNF4α is responsible for the transcription activation of mature hepatocyte specific genes[19–23] and it is able to reverse established liver cirrhosis[24], suggesting a role in preserving hepatocellular homeostasis during chronic liver injury[25]. We studied the correlation between parameters

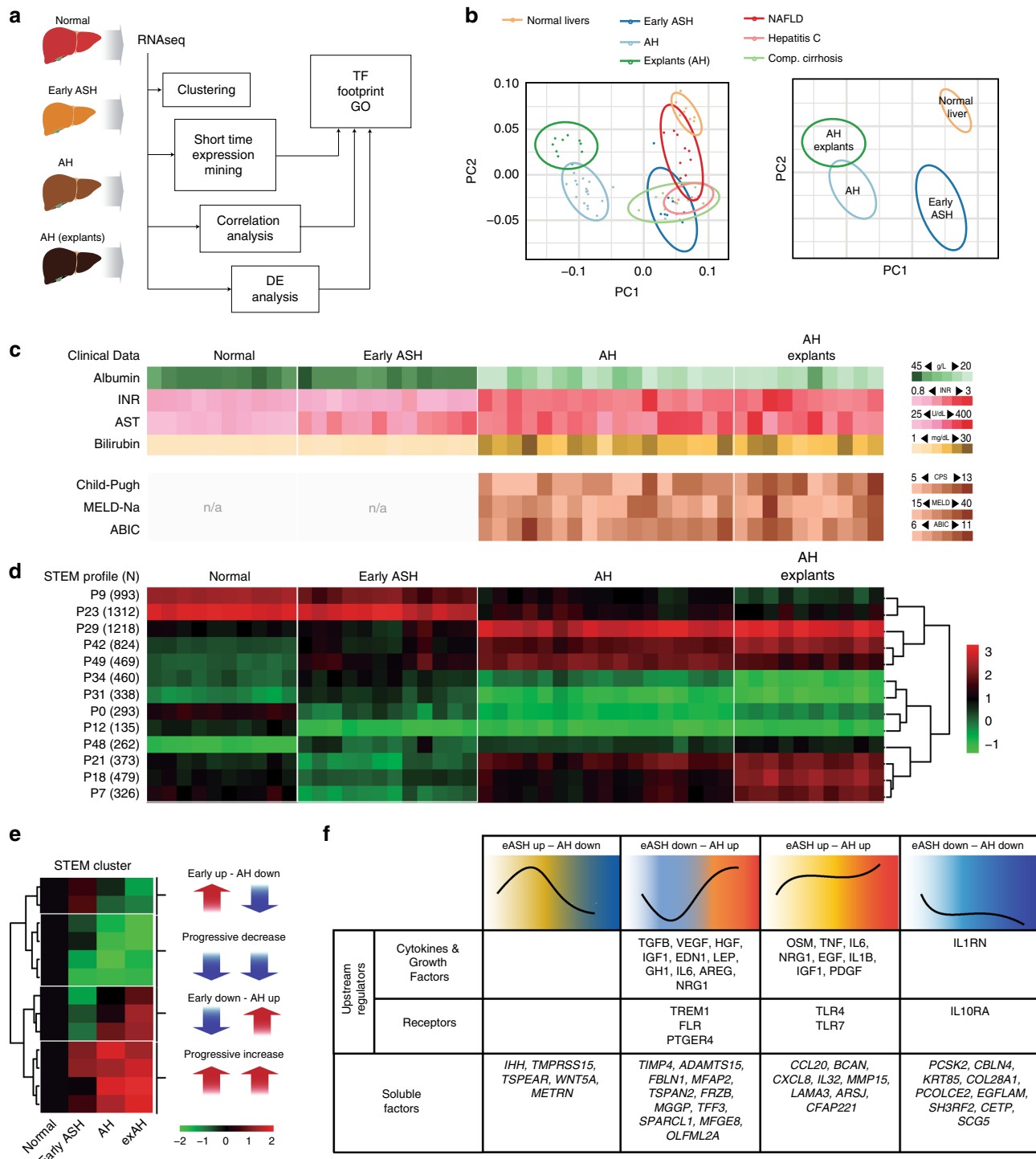

indicative of liver synthetic function and HNF4α activity. As shown in Fig. 3a, development of liver failure in the setting of AH, as indicated by elevated serum bilirubin levels and INR and decreased albumin synthesis, was strongly associated with a negative HNF4α Z-Score on IPA analysis. HNF4α is known to have two types of isoforms: the adult isoform is expressed in the liver during adulthood (HNF4α-P1) while the fetal isoform is driven by a ~45 kb upstream alternative promoter (HNF4α-P2). During embryonic development, the P2 promoter is used and an alternative splicing of the first exon is produced, originating the fetal isoforms α7–12. These variants lack the AF-1 domain in the N-terminal of the protein resulting in less transactivation activity,

affecting its interaction with coregulators (Fig. 3b)[26–28]. The relevance of P2 derived isoforms in adult human liver disease is not well-known. We then studied the expression of N-terminal isoforms in normal and AH human livers. HNF4α-P1 mRNA remained unchanged in AH, while there was a dramatic up-regulation in the expression of the fetal HNF4α-P2 isoform in livers from patients with AH (Fig. 3c). We found that the expression of the lncRNA *HNF4A-AS1*, which shares the P1 promoter region with *HNF4A*, was downregulated in patients with AH (Fig. 3c). The function of this antisense lncRNA is unknown. The expression of *HNF4A-AS1* measured by real time PCR was higher in human primary hepatocytes than in HepG2

**Fig. 1** Liver transcriptome encompasses disease progression in patients with ALD. **a** Human phenotypes included in the RNA-seq analysis: normal human livers ($n = 10$), early ASH ($n = 12$), AH ($n = 18$) and explants from AH patients ($n = 11$). Diseased controls: liver biopsies from patients with NAFLD ($n = 9$), non-cirrhotic HCV ($n = 9$) and compensated cirrhosis ($n = 9$). Unbiased clustering and Short Time Expression Miner (STEM) algorithm were used to group patients by RNA profiling and to identify main time-correlated patterns of expression. Kendall rank correlation coefficient and differential expression analysis (*limma*) between "Normal" and "Early ASH" and between "Early ASH" and "AH" groups was performed. Motif enrichment analysis (Opossum) and network analysis (Ingenuity Pathway Analysis) were used to identify main transcription factors involved in gene expression changes. **b** A schematic summary of Principal component analysis (PCA). **c** Heatmap of clinical and laboratory data of ALD patients: (Top) liver function tests: albumin serum levels, International Normalized Ratio (INR), aspartate aminotransferase (AST) and total serum bilirubin levels; (Bottom) Liver prognostic scores including Child-Pugh, MELD and ABIC; The color scale on the right indicates the range of each laboratory or clinical parameter. **d** Heatmap of STEM results, showing average expression (normalized log counts) of main groups of genes based on gene enrichment profile expression. Left column: STEM profile and number of genes. On top, patient phenotypes. Right panel, hierarchical clustering of profiles. See Supplementary Fig 2 for additional data from STEM analysis. **e** Heatmap of STEM results showing mean counts for all pattern-grouped genes for patients belonging to each disease stage. In the right panel, schematic representation (thick arrows) of main time-related expression patterns. **f** IPA analysis showing upstream regulators and soluble factors for each of four general expression pattern clusters. Regulators identified as cytokines, growth factors and receptors with a threshold ZS of 2 are presented (top-middle). Among the most 100 differentially expressed genes for each analysis, genes encoding secreted proteins are presented (bottom)

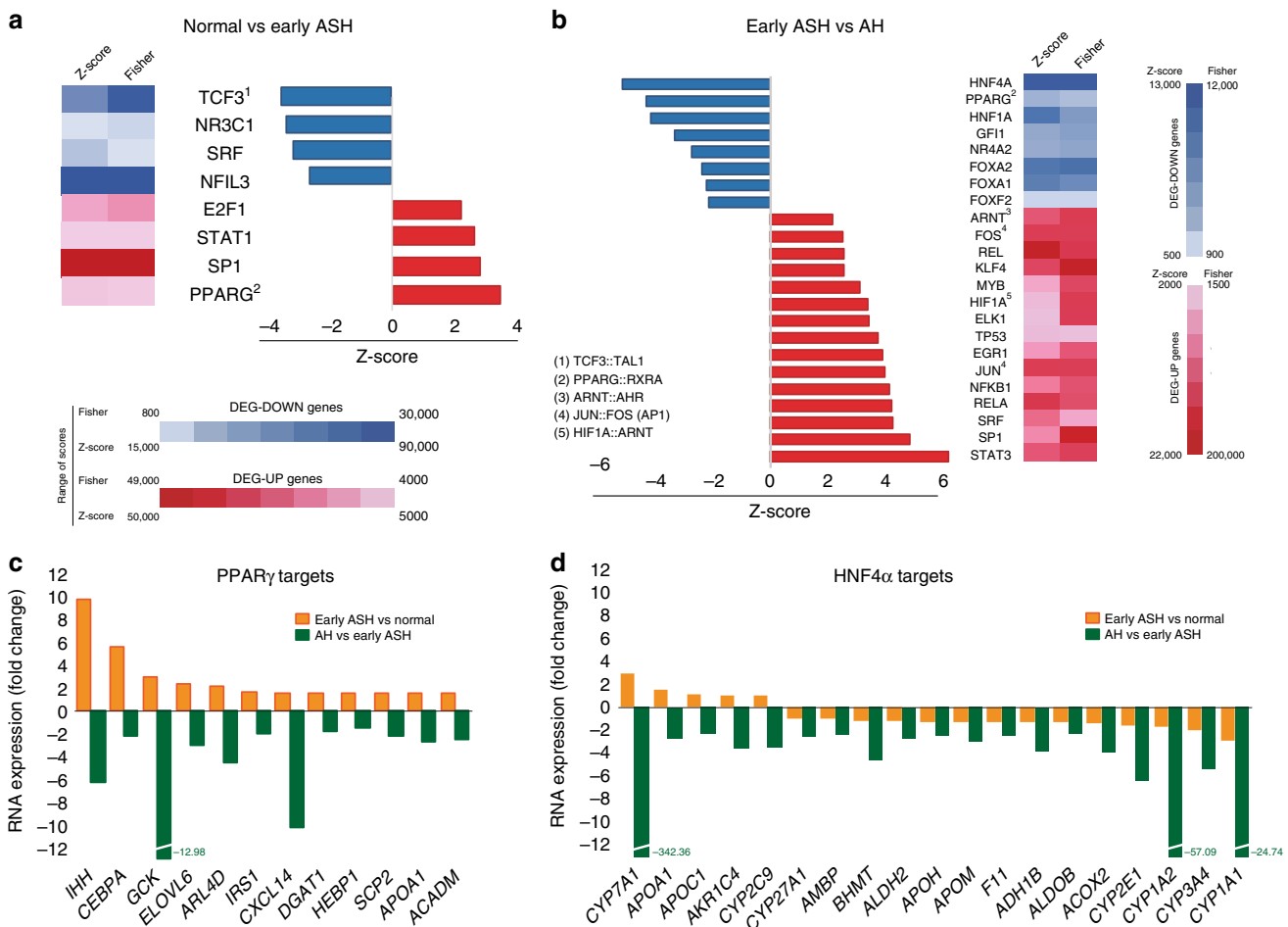

**Fig. 2** The predicted activity of liver-enriched transcription factors is defective in AH patients. **a, b** Transcription factor transcriptomic footprint inferred using Ingenuity Pathway Analysis (IPA) and Opossum analyses. Top differentially expressed (DE) genes between **a** normal livers and early ASH and **b** between early ASH and AH patients were the input for these analyses. Blue/Red indicates predicted activation/inhibition or motif enrichment in top 2000 downregulated/upregulated DE genes. **c** Selected target genes of PPARγ identified by IPA analysis. **d** Selected target genes of HNF4α identified by IPA analysis. Fold Changes (FC) in Normal vs Early ASH and between Early ASH and AH are presented. All genes in **g** and **h** had a FDR <$10^{-6}$ in DE analysis

and Hep3B cell lines (mean of 33 vs 37 and 38 cycles, respectively). The expression of this lncRNA could be related to HNF4α regulation and cell differentiation. Further studies should evaluate the functional role of this lncRNA in hepatocyte function and, in particular, in patients with AH. Importantly, up-regulation of

HNF4α-P2 was not seen in early forms of ALD or in other types of liver diseases such as NAFLD and chronic hepatitis C (Fig. 3c). In order to further explore the regulation of the HNF4α locus, we used a specific computational tool (i.e. Multivariate Analysis of Transcript Splicing -MATS-)[29] to assess differences in HNF4α

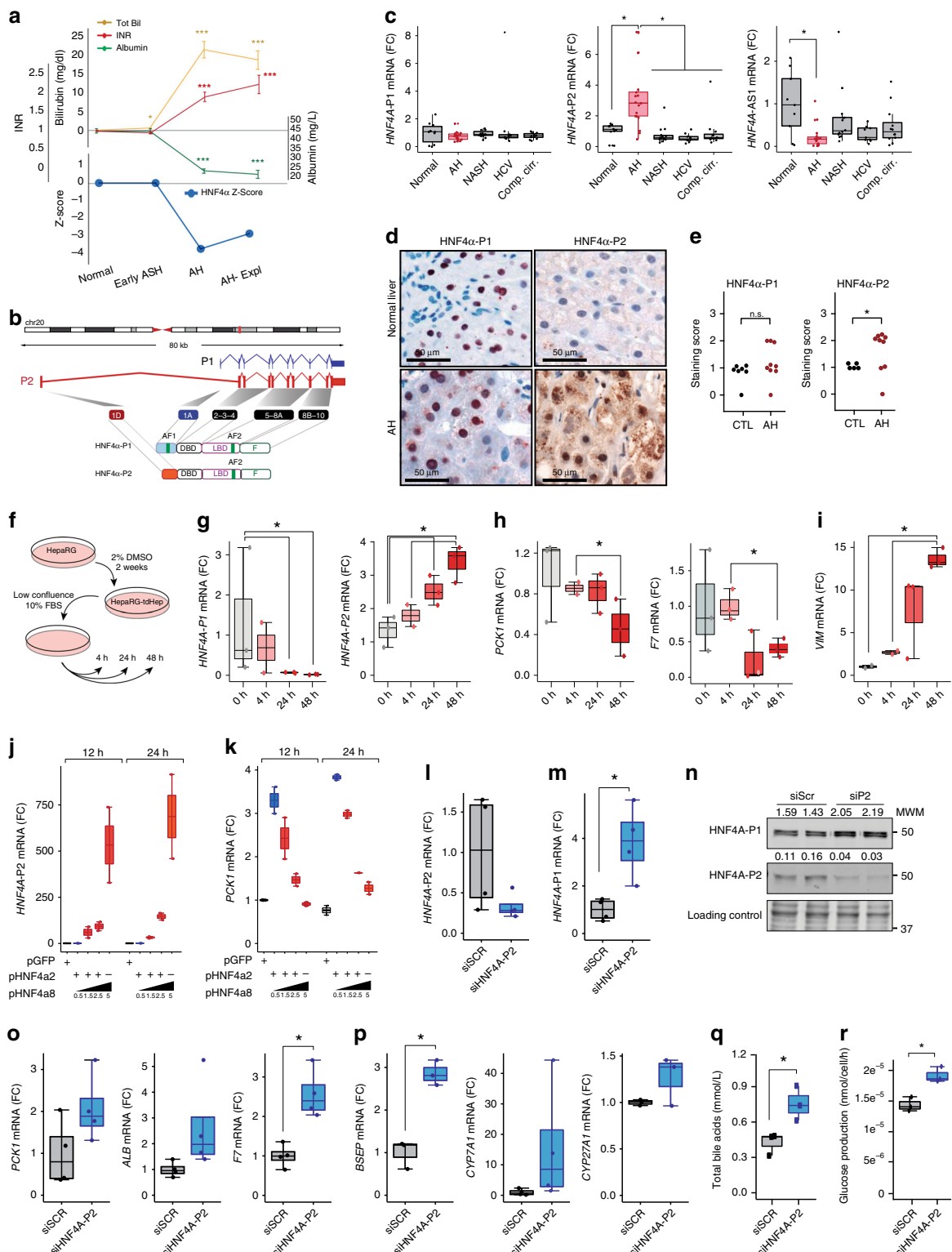

splicing between normal and AH livers. AH livers showed increased expression of exon 1D, 4, 5, 6, 9, and 10 (Supplementary Fig. 5a, b). The correlation of the expression of exon 1D with any of the other 10 exons was higher in patients with AH (Supplementary Fig. 5c–e). These differences suggest a profound deregulation of *HNF4A* gene splicing. The analysis of the exon exclusion events also showed an increase in the exclusion of exon 7 and a decrease of in exclusion of exon 8 (Supplementary Fig. 5f). Exon 8 encodes for a fraction of the AF-2 domain, which

is essential for post-translational regulation and activity of HNF4α[30]. Alterations of splicing in this region could thus affect HNF4α stability and/or activity. Further studies should evaluate the functional role of these C-terminal variants.

The hepatic expression of HNF4α isoforms in patients with AH was then assessed by immunohistochemistry (IHC) with specific N-terminal antibodies. HNF4α-P1 signal was detected in the nuclei of both normal and AH hepatocytes. Conversely, the HNF4α-P2 isoform, barely detected in the nucleus of normal

**Fig. 3** Fetal HNF4α-P2 isoform increase in patients with AH and its effect in HNF4α-P1. **a** Levels of bilirubin, INR and albumin levels in serum along ALD progression (values expressed as Mean ±SEM) and HNF4α footprint Z-Score. **b** Scheme of HNF4A gene fetal (P2) and adult (P1) isoforms structure and protein variants. **c** Real-Time quantitative PCR (qPCR) of HNF4A-P1 and P2 dependent isoforms, and lncRNA HNF4A-AS1 in the cohort of patients in Fig. 1. **d** Immunohistochemical detection of adult and fetal HNF4A protein variants in patients with AH (n = 9), and controls (n = 9), using N-terminal specific antibodies. **e** Semi-quantitative assessment of IHC signal for each antibody for nuclear staining. **f–i** HepaRG cells were retro-differentiated into tumor-derived Hepatocyte-like cells (HepaRG-tdHep); de-differentiation was induced with FBS and RNA was extracted at 4, 24, and 48 h (n = 3 for each time point); qPCR of **g** HNF4α-P1 and P2 isoforms, **h** phosphoenol-pyruvate carboxy-kinase (PCK1), clotting Factor VII (F7) and **i** vimentin (VIM) **j**, **k** HepG2 cells were transfected with plasmids encoding P1 (HNF4α2) and P2 (HNF4α8) variants. P1 was maintained at same dose while P2 was increased as indicated. RNA was extracted 12 h and 24 h after transfection (n = 3); qPCR of **j** HNF4α-P2 isoform and **k** PCK1. **l–p** HepG2 cells were transfected with siRNA targeting the first exon (1E) of HNF4α-P2 isoforms (n = 3), and RNA and protein was extracted at 48 h after transfection. qPCR of **l** HNF4α-P2 and **m** HNF4α-P1. **n** Western blot of HNF4α-P1 and HNF4α-P2 in nuclear extracts. qPCR of HNF4α-P1 targets related to **o** metabolic functions (PCK1, ALB and F7) and **p** bile acid synthesis and transport (BSEP, CYP7A1 and CYP27A1). **q**, **r** Primary human hepatocytes were silenced with siRNA-HNF4A-P2. **q** Supernatant was collected 48 h after transfection (n = 3 for each group) and total bile acids were quantified. **r** Glucose production in P2-silenced primary human hepatocytes. Significance was determined by unpaired, two-tailed Student's t-test in **a** and **c**, by Fisher exact probability test in **d**, **e** and by two-tailed Mann–Whitney U test in **g**, **i**, **l**, **m**, **o**, **p**, **q**, **r**: *P < 0.05. For box-and-whisker plots: perimeters, 25th–75th percentile; midline, median; individual data points are represented

livers, was markedly up-regulated in AH hepatocytes (Fig. 2d, e). Other important LETFs inhibited in AH such as HNF1α and FOXA-1 showed decreased nuclear expression and increased cytoplasmic localization (Supplementary Fig. 6a–c). In contrast, RXRα, whose heterodimer with PPARγ was predicted to be inhibited (Fig. 2b), did not show differences in the IHC of these patients compared to normal livers (Supplementary Fig. 6d, e).

**Knockdown of HNF4α-P2 ameliorates HNF4α-P1 expression.** We then sought to determine whether P2 expression in hepatocytes contributes to the loss of mature hepatocyte biological functions during AH including bile acid homeostasis, as well as metabolic and synthetic functions. To address this question, we used a well-characterized model of hepatocyte de-differentiation of HepaRG-tdHep into HepaRG cells[31]. HepaRG cells were first differentiated into the hepatocyte lineage by a two-week treatment with 2% DMSO. Then, these so-called tumor-derived HepaRG hepatocyte-like cells (HepaRG-tdHep) were cultured in the absence of DMSO and at low confluence. The expression of HNF4α isoforms and HNF4α targets was analyzed at different time points (Fig. 3f). Hepatocyte de-differentiation resulted in a rapid decline of HNF4α-P1 isoform expression with a constant upregulation of HNF4α-P2 isoforms (Fig. 3g). Hepatocyte-specific genes such as PCK1 and F7 were downregulated (Fig. 3h), while Vimentin (VIM), a known EMT marker, was upregulated (Fig. 3i). In AH patients, we found increased expression of progenitor cell markers and markers of epithelial-to-mesenchymal transition (EMT), suggesting a de-differentiation of hepatocytes (Supplementary Fig. 7a). This was also suggested by a correlation analysis with published tissue and cell type published gene sets (Supplementary Fig. 7b,c). We then performed gain and loss-of function studies to elucidate the role of P2 in hepatocyte biological functions. Overexpression of HNF4α-P2 resulted in decreased expression of the HNF4α-target gene PCK1 (Fig. 3j, k). In contrast, abrogation of HNF4α-P2 resulted in increased HNF4α-P1 gene and protein expression (Fig. 3l–n). The expression of HNF4α target genes involved in hepatocyte metabolic, secretory and synthetic functions such as PCK1, F7, Albumin (ALB), CYP7A1, CYP27A1 and biliary salt export pump (BSEP) was also increased (Fig. 3o, p). Moreover, this maneuver restored bile acid synthesis and the formation of glyco-cheno-deoxycholate conjugated bile acid (Fig. 3q, Supplementary Fig. 8), and also stimulated glucose production (Fig. 3r) in human primary hepatocytes. Overall, these results suggest that P2 over-expression negatively regulates HNF4α-dependent gene expression and several biological properties of mature hepatocytes that are commonly lost in AH.

**TGFB1 mediates HNF4α dysregulation.** We next explored the potential mechanisms involved in HNF4α P1-P2 imbalance during the development of liver failure in AH. Unbiased analysis of transcriptomic changes in patients progressing to AH uncovered potential main upstream regulators (Fig. 4a). Transforming growth factor β1 (TGFβ1) was found to be the most relevant factor, followed by epidermal growth factor (EGF). Expression of TGFβ1 and its receptors 1 and 2, as well as the EGF receptor ligand Amphiregulin (AREG), were markedly increased in AH livers (Fig. 4b, c). We then hypothesized that TGFβ1 and AREG regulate the relative expression of HNF4αP1-P2 in hepatocytes. In HepG2 and in Hep3B cells, TGFβ1 and AREG synergistically decreased HNF4α-P1 protein levels and RNA expression while TGFβ1 but not AREG increased HNF4α-P2 levels and expression (Fig. 5a, b, Supplementary Fig 9a, b, Hep3B experiments are shown). HNF4α-P1 is known to inhibit the expression of HNF4α-P2[32]. Nevertheless, in our experiments HNF4α-P1 down-regulation upon TGFβ1 treatment occurred several hours later than HNF4α-P2 upregulation, indicating a direct action of TGFβ1 (Fig. 5a, Supplementary Fig. 9a, b). Accordingly, knockdown of HNF4α-P1 expression did not increase HNF4α-P2 levels at baseline or in the presence of TGFβ1 (Fig. 5c). The effect of TGFβ1 was TGFβ1R1-dependent (Fig. 5d). Surprisingly, the inhibition of the nuclear translocation of SMAD family proteins by SMAD4 knockdown blocked TGFβ1-mediated HNF4α-P1 downregulation and transcriptional function but did not inhibit HNF4α-P2 upregulation. These results suggest that the action of TGFβ1 on HNF4α includes SMAD-dependent and SMAD-independent signaling pathways (Fig. 5e). TGFβ1 activated kinase 1 (TAK1) is essential for hepatocyte proliferative response and survival[33]. Strikingly, the selective inhibition of TAK1 reduced TGFβ1-mediated induction of HNF4α-P2 expression without affecting HNF4α-P1 downregulation (Fig. 5f). Cellular Src (c-Src) selectively decreases HNF4α-P1 levels in response to EGF[34]. But it can also be activated by TGFβ1[35] and transduce the signal through TAK1 in the context of hepatocyte protection[36]. The pharmacological inhibition of c-Src completely reverted the effects of TGFβ1 on both HNF4α-P1 and HNF4α-P2, indicating an essential role in HNF4α deregulation (Fig. 5g, h). One of the multiple effectors of c-Src and TAK1 is AP-1 transcription factor. We then scanned the genomic region around HNF4α-P2 Transcription Start Site (TSS, ±1 kb) in search of AP-1 transcription factor binding sites (TFBS). Only 6 FOS::JUN TFBS were found with high Relative Score (>85%, Supplementary Fig 10a, b). Next, we performed chromatin immunoprecipitation using RNA Polymerase II (RNA Pol II) and phospho-c-JUN antibodies. We studied several genomic regions. Interestingly, upon TGFβ1

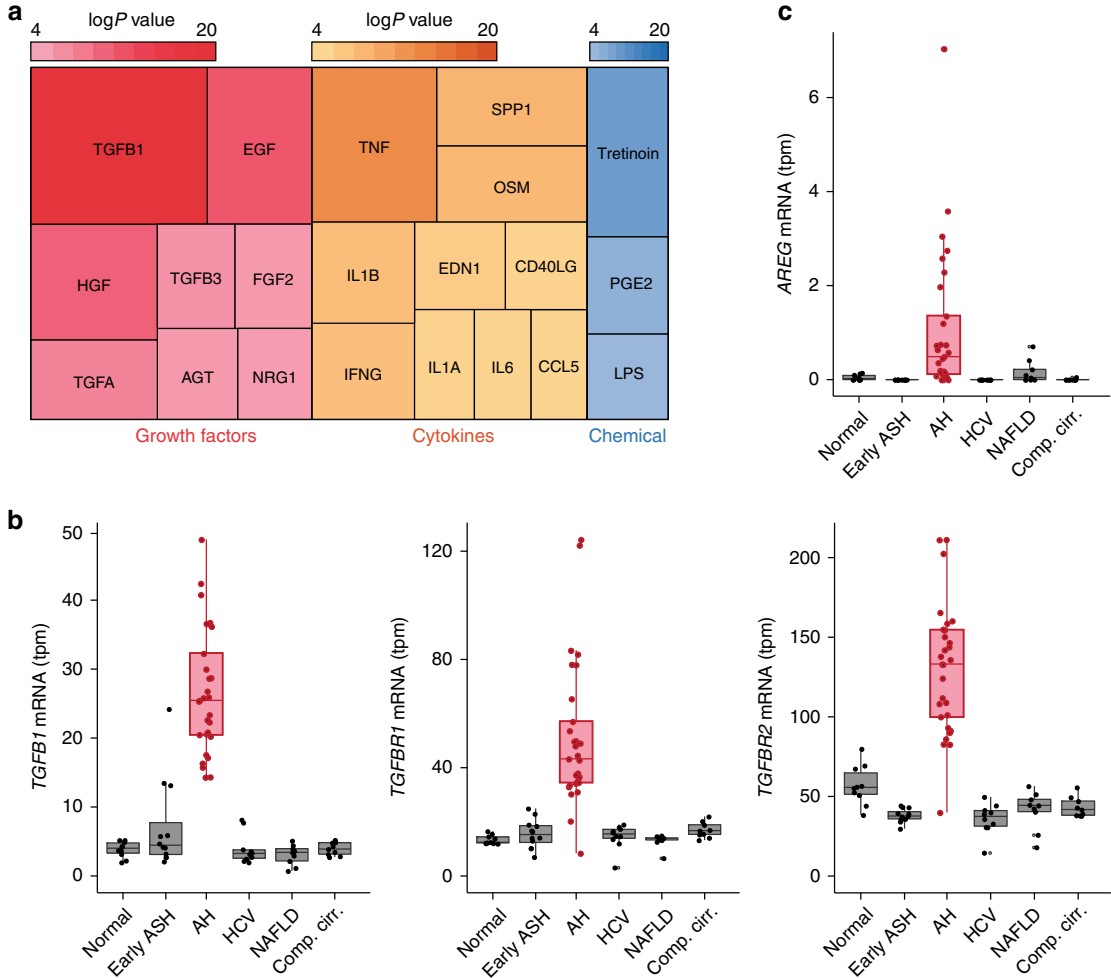

**Fig. 4** TGFβ1 is the main upstream regulator of transcriptomic reprogramming in ALD. **a** Treemap of the top predicted activated growth factors, cytokines and chemicals as detected by IPA. Color and box areas are related to *p*-values, indicated in top-right color-scale. Most significant hits ($P < 10^{-4}$) are shown. **b**, **c** mRNA abundance in transcripts per million (tpm) from normal livers, AH livers and livers of non-alcohol-related chronic disease of **b** TGFβ1, TGFβRI, and TGFβRII and **c** Amphiregulin (AREG). For box-and-whisker plots: perimeters, 25th–75th percentile; midline, median; whiskers, minimum to maximum values; individual data points are represented. Gene expression levels are presented in transcripts per million reads (tpm)

treatment, RNA Pol II was found to be bound to a proximal region of intron 1 that contains one FOS::JUN site. The same region was also pulled down with the c-JUN antibody specially under TGFβ1 treatment (Fig. 5j). These results indicate that TGFβ1 promotes the recruitment of c-JUN and RNA Pol II to the proximal intron 1 region. The effect of AREG on HNF4α-P1 was blocked by the EGFR inhibitor PD15 (Supplementary Fig 9c). The integrity of the MEK/ERK pathway was required for AREG-mediated (Supplementary Fig. 9d) and TGFβ1-mediated (Supplementary Fig. 9h, i) HNF4α-P1 downregulation. In contrast, inhibition of c-Src did not restore HNF4α-P1 levels upon AREG or EGF treatment (Supplementary Fig. 9e, f). Of note, for both AREG and TGFβ1 to induce HNF4α-P1 downregulation, the function of the proteasome must be intact (Fig. 5i and Supplementary Fig. 9g), indicating a strong effect of these growth factors in HNF4α-P1 protein stability. We then explored if the detrimental effect of TGFβ1 on hepatocyte function is mediated by HNF4α-P2 increase. Transfection of primary human hepatocytes and several cells lines (HepG2 and Hep3B cells) with siRNA targeting P2 isoforms abolished TGFβ1-mediated suppression of HNF4α-P1 (Fig. 5k–m, HepG2 experiments are shown). TGFβ1-induced inhibition of HNF4α-P1 dependent genes, in particular F7 and CYP7A1, was significantly reverted by P2 silencing while

the effect on other genes such as PCK1 or CYP27A1 was limited (Fig. 5m, n, HepG2). The production of bile acids was similarly restored by P2 silencing (Fig. 5o, primary human hepatocytes). These results suggest that the re-expression of HNF4α fetal isoforms in AH could participate in TGFβ1-induced loss of hepatocellular function, pointing to these isoforms as potential therapeutic targets.

Next, we sought to identify potential mechanisms that maintain the normal HNF4α P1/P2 ratio during the compensated stages of ALD. Transcriptomic footprint analysis revealed a marked predicted activation of PPARγ in early phases of ALD (Fig. 2a). Interestingly, P2 silencing increased the expression of PPARγ, indicating a potential antagonism with P2 (Fig. 6a). Because of its hepatoprotective properties and its inhibitory action on TGFβ1[37], we explored if PPARγ antagonizes TGFβ1-mediated HNF4α dysregulation. The PPARγ agonist rosiglitazone increased P1 isoforms in all conditions and decreased the abundance of P2 isoforms when TGFβ1 was combined with AREG (Fig. 6b, c). TGFβ1-mediated ALB down-regulation was restored by PPARγ activation. The effect of rosiglitazone on HNF4α-P1 mRNA levels was dose dependent (Fig. 6d). Overall, these results suggest that, in hepatocytes, PPARγ counteracts TGFβ1-mediated HNF4α-P1 downregulation. This mechanism

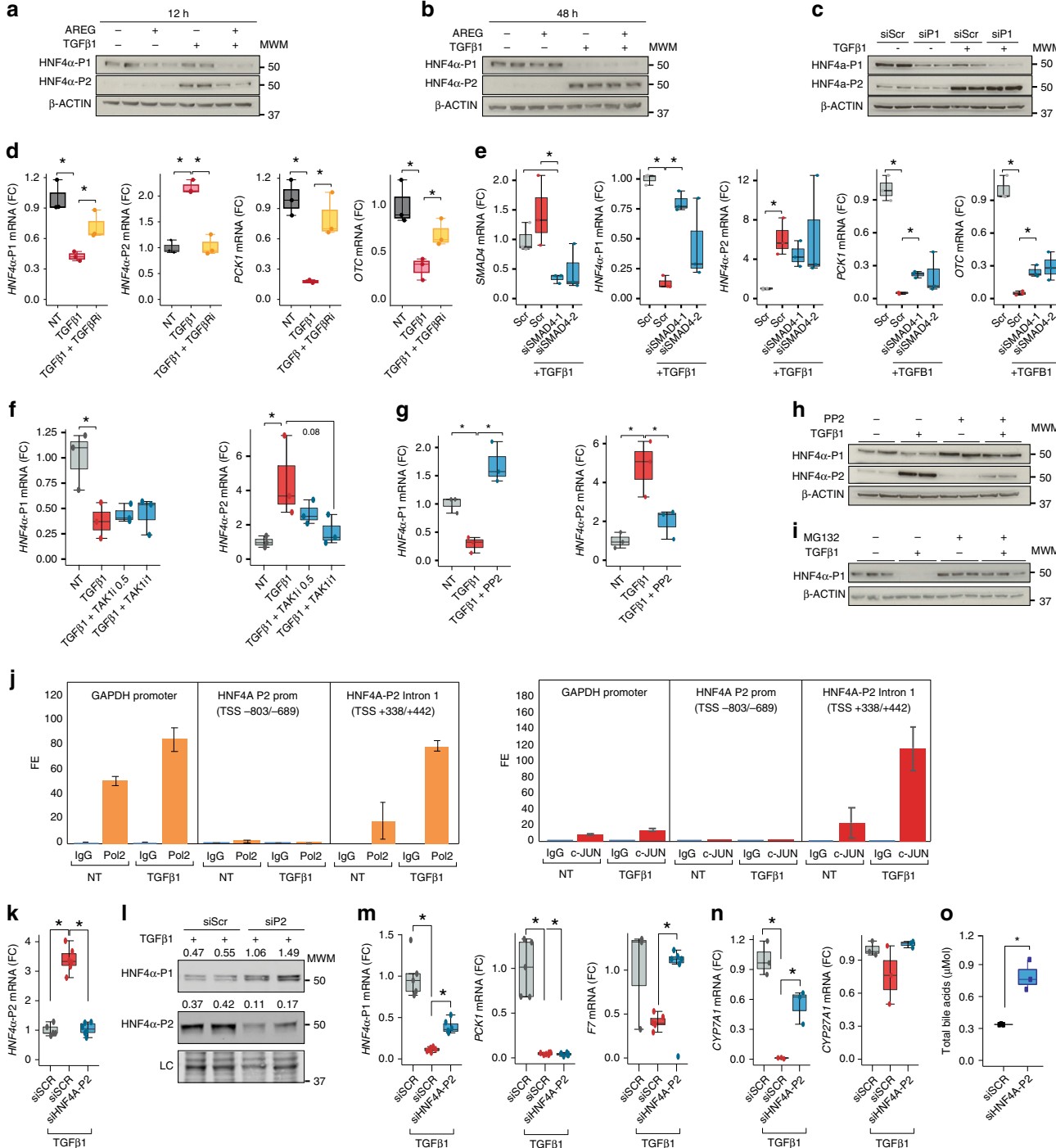

could partially explain the beneficial effects of PPARγ agonists in experimental alcoholic and non-alcoholic liver disease[38].

**miR122 is downregulated in patients with AH.** HNF4α is the main regulator of mir122 expression in hepatocytes through its binding to the hpri-miR-122 promoter[39]. In our RNAseq analysis, a number of patients with AH had low levels of liver miR122, while other AH patients having normal levels (Fig. 7a). The fact that HNF4α-P1 activity is suppressed in these patients suggests that it could play a role in the decrease of miR122 expression. Interestingly, some patients with early ASH had reduced levels of miR122, indicating that alcohol itself could reduce miR122

expression. Patients with AH have increased levels of GRHL2 expression (Fig. 7b), a transcription factor that has been recently associated with miR122 inhibition in a mouse model of ethanol + CCl₄ mediated liver injury[40]. There was a significant inverse correlation between these two genes (Fig. 7c). Whether this correlation denotes causal relationship requires further investigation. We used Ingenuity Pathway Analysis knowledge base to determine the significance (p value) and direction of the functional enrichment (Z-Score) of all human miRNA of the set of differentially expressed genes between early ASH and AH. Importantly, miR122 was found to have the most significant negative Z-score (Fig. 7d). HNF4α and miR122 dependent genes are only partially overlapping (Fig. 7e), suggesting that miR122 dysfunction in the

**Fig. 5** TGFβ1 induces the expression of HNF4α-P2 and binding by c-JUN to its promoter. **a**, **b** Immunoblots of HNF4α-P1 and HNF4α-P2 isoforms in Hep3B cells treated with TGFβ1 and or AREG (50 nM) for **a** 12 and **b** 48 h (n = 2). **c** Immunoblots of HNF4α-P1 and HNF4α-P2 from Hep3B cells transfected with an HNF4α-P1 specific siRNA for 48 h and treated with TGFβ1. **d** Hep3B cells were pre-treated with TGFβ-RI inhibitor SB431542 (5 nM) and treated with TGFβ1 (for 8 h (n = 3); qPCR of HNF4α-P1 and P2, PCK1 and Ornithine Carbamoyltransferase (OCT). **e** SMAD4-silenced Hep3B cells were treated overnight with TGFβ1; qPCR of SMAD4, HNF4α-P1 and P2 isoforms, PCK1 and OCT **f** Hep3B cells were pretreated with TAK1 inhibitor NG25 at 0.5 or 1 μM and then treated with TGFβ1 for 8 h (n = 3). qPCR of HNF4α-P1 and P2 isoforms. **g** Hep3B cells were treated with TGFβ1 overnight in the presence of cellular Src (c-Src) inhibitor PP2 (10 μM); **g** qPCR of HNF4α-P1 and P2 **m** Immunoblots of HNF4α-P1 and HNF4α-P2. **k** Chromatin immunoprecipitation of Hep3B cells treated with TGFβ1 overnight; RNA Polymerase II (orange), phospho-c-JUN (red) antibodies and normal mouse IgG (blue) were used. qPCR of GAPDH promoter, HNF4α-P2 promoter, and HNF4α-P2 proximal intron 1. Fold Enrichment of Pol II or c-JUN to control IgG is presented. **l** Hep3B cells were treated with TGFβ1 for 24 h and with the addition of proteasome inhibitor MG132 (10 μM) 2 h before collection when indicated (n = 3); immunoblot of HNF4α-P1. **o–r** HNF4α-P2-silenced HepG2 cells were collected 8 h (RNA) or 24 h (Nuclei) after TGFβ1 treatment (5 ng/ml) (n = 4–6); **o** qPCR of HNF4α-P2; **p** immunoblot of nuclear HNF4α-P1 and P2 isoforms **q** qPCR of HNF4α-P1 target genes PCK1, ALB, F7 and **r** CYP7A1 and CYP27A1. **s** Primary human hepatocytes were silenced with siRNA-HNF4α-P2 and supernatant was collected 48 h after transfection and 8 h after TGFβ1 treatment. Total bile acids in supernatant were quantified (n = 3). Significance was determined by two-tailed Mann–Whitney U test in **d**, **e**, **g**, **k**, **m**, **n**, **o** *P < 0.05. For box-and-whisker plots: perimeters, 25th–75th percentile; midline, median. The TGFβ1 dose used was 5 ng/ml

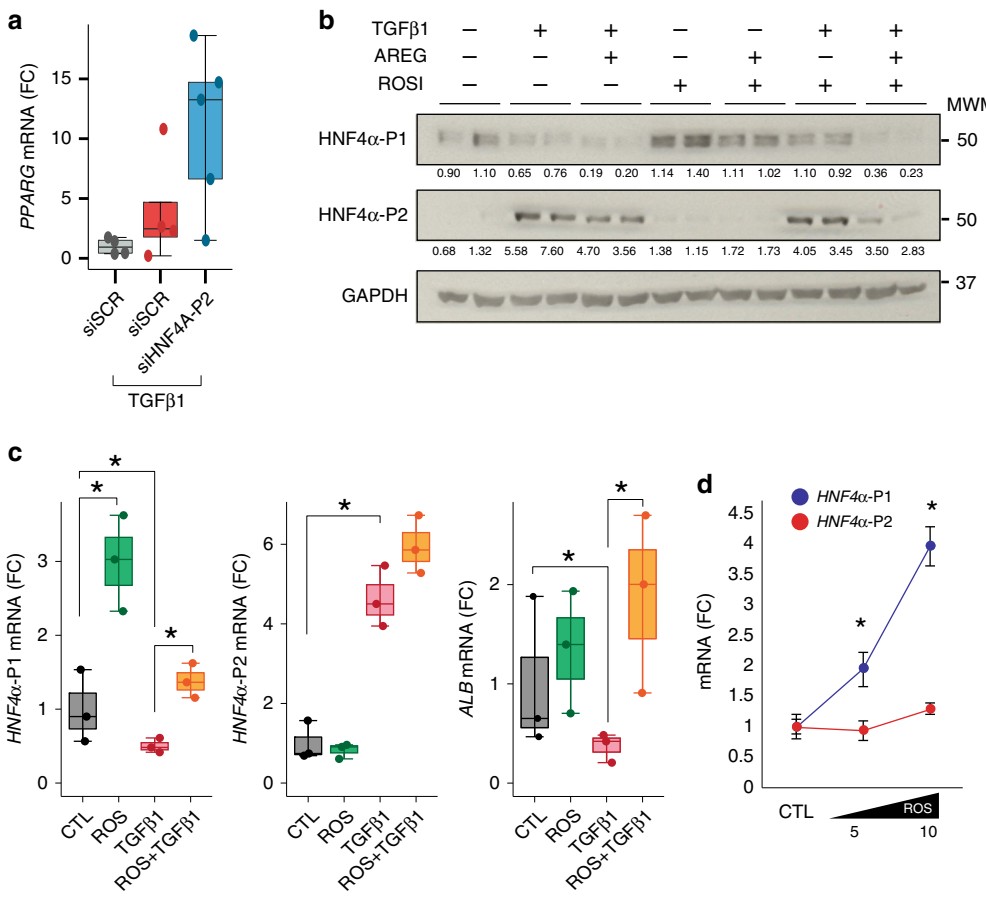

**Fig. 6** PPARγ agonist Rosiglitazone partially restores TGFB1-induced HNF4A de-regulation. **a** HepG2 cells were transfected with HNF4α-P2 siRNA for 48 h and collected 8 h after TGFβ1 treatment (5 ng/ml) (n = 5 for each condition);qPCR of PPARγ. **b**, **c** Hep3B cells were pretreated with rosiglitazone (10 μM) overnight and then treated with TGFβ1 (5 ng/ml) and/or AREG (50 nM) for 8 h (n = 3 for each condition); **b** Immunoblot of HNF4α-P1 and HNF4α-P2 **c** qPCR of HNF4α-P1 and P2 isoforms and ALB. **d** Hep3B cells were treated with rosiglitazone at doses of 5 and 10 μM, and harvested 16 h after treatment; qPCR of HNF4α-P1 and P2 isoforms (n = 3 for each condition). Significance was determined by two-tailed Mann–Whitney U test in **a**, **b** and **d**: *P < 0.05. For box-and-whisker plots: perimeters, 25th–75th percentile; midline, median

progression from early ASH to AH could involve HNF4α-independent pathways. Finally, we used miRTarBase[41], a curated database of miRNA targets, to select the top 10 most validated miR122 targets in humans. In patients with AH, 9 of those top targets were upregulated while none of them was found increased in early ASH (Fig. 7f). Although these new results support a

potential role for miR122 in AH, further functional experiments are needed to confirm this hypothesis.

**HNF4α-dependent genes are hypermethylated in AH patients.** Finally, we explored whether genetic or epigenetic factors are

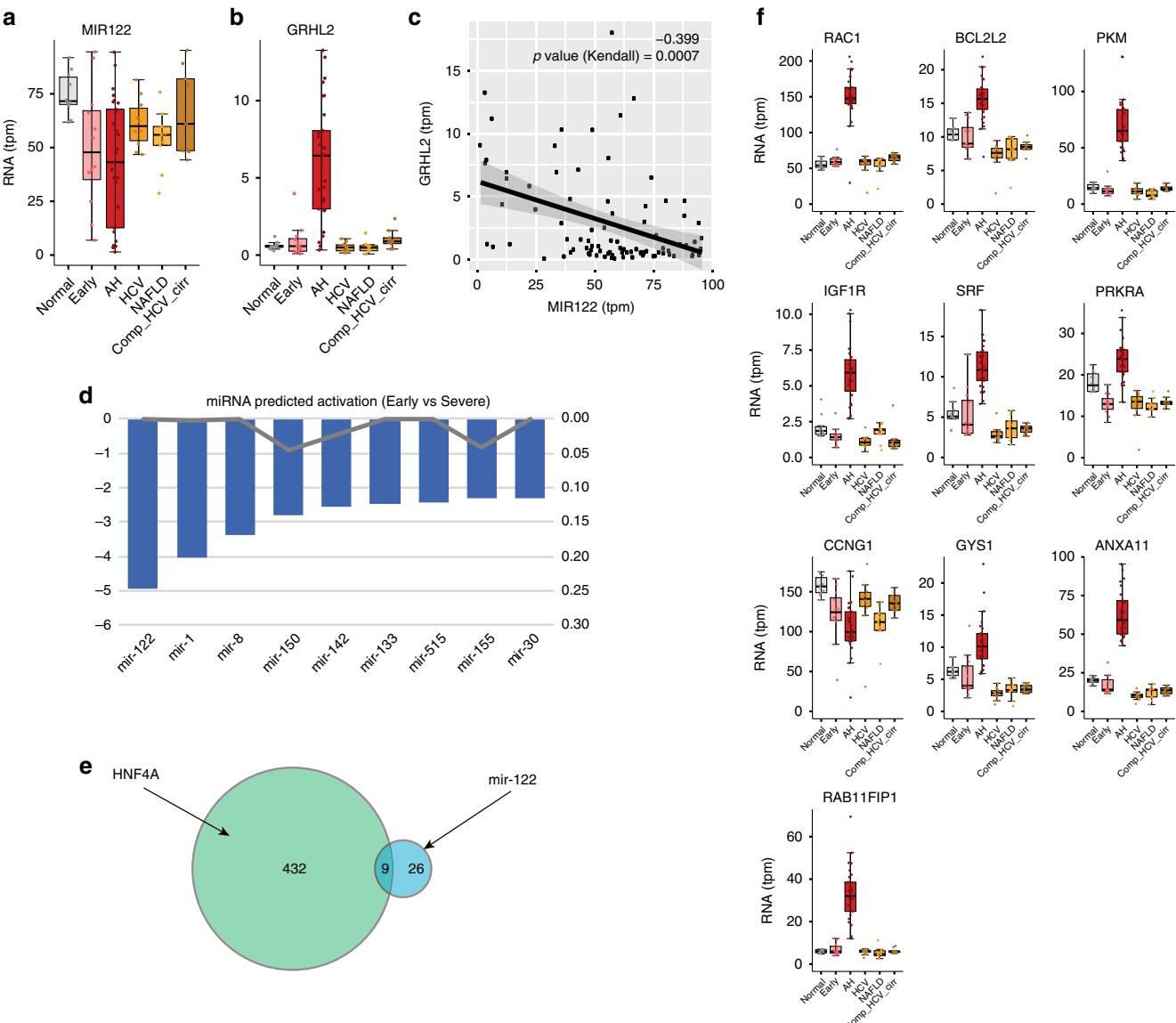

**Fig. 7** miR122 levels and miR122 predicted downregulation in AH patients. **a** RNA levels of hpri-miR122 in our cohort **b** levels of Grainyhead Like Transcription Factor 2 (*GRHL2*) in our cohort **c** correlation of GRH2L2 and MIR122 levels. *R* and *p* value (Kendall) are presented. **d** Results of miRNA predicted activity by means of IPA Upstream Regulator analysis when comparing early ASH vs AH. Top 8 miRNA are presented. **e** Venn diagram of the overlap between HNF4A and MIR122 targets among the differentially expressed genes in the comparison between early ASH and AH. **f** Box plot of most 10 validated miR122 targets (miRTarBase database) in our cohort. Box-and-whisker plots indicate 25th–75th percentile; midline, median; whiskers, minimum to maximum values; individual data points are represented. In bold, those genes that reached FDR < 10$^{-6}$ level of significance in DESeq2 differential expression analysis between early ASH and AH. For box-and-whisker plots: perimeters, 25th–75th percentile; midline, median. Gene expression levels are presented in transcripts per million reads (tpm)

involved in the defective LETFs function in AH. To address this question, we first analyzed GWAS data from a large cohort of AH patients ($N = 332$) and patients with alcohol abuse that never decompensated ($N = 318$) (Fig. 8a). None of the single nucleotide polymorphisms (SNP) detected in LETFs including HNF4$\alpha$, either genotyped or imputed, were significantly associated with AH development (Fig. 8b, c, Supplementary Data 3). Because exposure to either TGFβ1 or alcohol have been involved in DNA methylation and chromatin remodeling[42–45], we hypothesize that the disruption of the expression and activity of the transcriptional master regulators (i.e LETFs) in patients with AH could be part of a global epigenetic remodeling. In an unbiased fashion, we studied the overall expression analysis of genes encoding epigenetic modulators in patients with AH. For this purpose, we used the

EpiFactors database[46]. The top 5 hits of each family based on the differential expression comparing normal and AH patients are shown in Fig. 9a. Main genes found markedly deregulated included *HDAC 7*, *HDAC 11*, *PIWIL4* (*MIWI2*), *NCOR2*, *ZBTB33*, *PRDM6*, *PCGF2*, and *PHC2*. The DNA methyl transferases 1 and 3 A were notably increased in patients with AH (Fig. 9b, c). We then analyzed the methylation status of over 850,000 loci in normal livers ($N = 5$) and livers from AH patients ($N = 6$) and found around 3000 differentially methylated (DM) CpG-containing loci with an absolute change in beta value >0.3 and a false discovery ratio (FDR) <0.01 (Fig. 9d and Supplementary Data 4). Motif enrichment analysis of DM regions revealed the presence of HNF4$\alpha$ and PPARγ motifs in hypermethylated regions while hypomethylated regions were enriched

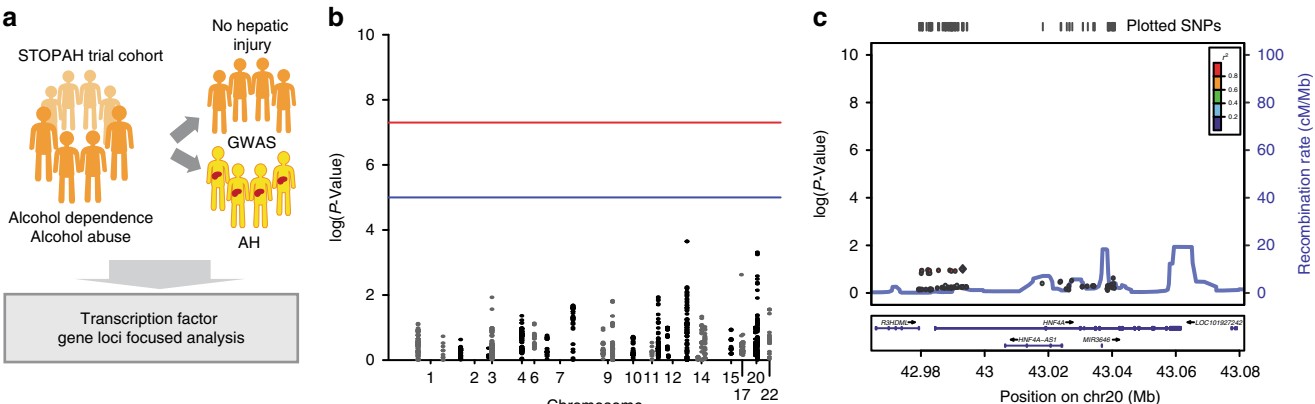

**Fig. 8** GWAS study does not show an association of LETF SNPs with the development of AH. **a** Detection of single nucleotide polymorphisms (SNP) associated to AH in transcription factor gene loci. In this study we compared patients had alcohol dependence but with no evidence of liver injury (n = 318) and patients with alcohol dependence and biopsy-proven severe AH (n = 332). **b** Manhattan plot of all the SNP present in the selected genomic regions (see also Supplementary Data 3). **c** LocusZoom plot of HNF4A locus

in motifs of inflammatory transcriptional regulators, such as STAT4 and AP1 complex (c-FOS, JUN) (Fig. 9e). The analysis of DM-CpG nearest genes with Ingenuity Pathway Analysis showed that among hypermethylated regions HNF4α footprint was the most enriched transcriptional regulator (Fig. 9f). These results mirrored data from RNA-seq analysis, showing a parallel between hypermethylation and down-regulation of regions controlled by LETFs (eg. HNF4α, HNF1α, CEBPα, SREBPs, CEBPβ) and other hepatoprotective factors such as PPARγ (Fig. 9f, g). These results were confirmed by RNA-seq of the same samples (Fig. 9h). The analysis of soluble upstream regulators revealed TNFα and TGFβ1 involvement in the expression of genes containing hypomethylated CpG (Fig. 9i). The presence of SNPs in differentially methylated regulatory regions could be involved in dysregulation of the HNF4A locus or the HNF4α-dependent transcriptome. Two annotated CpG islands near the HNF4A locus were identified using the UCSC human genome browser, which contained SNPs from the AH GWAS dataset (rs148377517 and rs13038786), although none of them were associated with the risk of developing severe AH (OR of 0.62 and 1.89; P value of 0.3809 and 0.24, respectively). We then analyzed the SNP located in Differentially Methylated Regions (DMR) located around the HNF4A locus. Five DMRs around HNF4A locus were identified and 20 SNPs fell within these DMR (Supplementary Data 5a, b). Next, we analyzed the SNPs within or near HNF4α binding motifs globally found within CpG islands and DMR. A total of 3214 DM CpG loci containing HNF4α binding motifs were found. The SNPs lying±75bp from the locus of a DM CpG locus were extracted from the AH GWAS dataset. In total 505 SNPs fulfilled these criteria. Of these, 18 demonstrated a potential association with the risk of developing AH (P < 0.05) (Supplementary Data 5c). Of note, the SNP rs942043 lies near E2F3 gene which encodes a transcription factor involved in cell cycle regulation. The variant rs846897 lies near the gene IGSF23, a member of the immunoglobulin superfamily. Finally, we analyzed the DMR around HNF4α binding motif-containing CpG loci. In total, 328 DMRs were extracted and 36 SNPs were found in these regions. Four variants, three of which were in perfect linkage disequilibrium, demonstrated a potential association with disease (P < 0.05) (Supplementary Data 5d). These lie within the coding region of the gene CLCN6 that encodes a chloride transporter. However, when viewed in the context of the number of tests performed (SNPs found) these associations are highly likely to represent false positives.

**AH patients show repressive chromatin in HNF4α targets**. Finally, we analyzed data from H3K27Ac, H3K27me3, H3K4me1 and H3K4me3 chromatin immunoprecipitation experiments coupled to DNA sequencing (ChIP-seq) of normal livers (N = 5) and livers from patients with AH (N = 8) (Fig. 10a). H3K27Ac, H3K4me1 and H3K4me3 marks are known to be enriched in active regulatory regions, while H3K27 trimethylation results in gene expression inhibition. As expected, in patients with AH, the promoter regions of HNF4α targets such as PCK1, CYP3A4 and F7 were poor in H3K27Ac, whereas other gene promoter targets of ICAM1 were rich in this mark (Fig. 10b, c). When focusing on the HNF4A genomic locus, we found enhanced H3K27Ac mark in the P2 promoter, in accordance with our RNA expression results (Fig. 10d). We then analyzed quantitatively the number and significance of the peaks called in each sample. In clear support of our findings in the RNA-seq cohort, patients with AH had a decreased number and significance of H3K27Ac, H3K4me1 and H3K4me3 peaks in the HNF4A P1 promoter and increased H3K27Ac enrichment in P2 promoter (Fig. 10e). A similar histone modification pattern was found in the promoter of HNF4α target genes such as PCK1 and CYP3A4. Interestingly, in F7 promoter, an increase in H3K4me3 was found (Fig. 10f). These results suggest that the epigenetic regulation of HNF4α target genes could be driven by different mechanisms. Other genes, such as ICAM1 did not show differences on histone peak fold enrichment (Fig. 10g). Further studies should identify molecular drivers of methylation and chromatin remodeling in AH, which could result in the development of novel targeted therapies.

Finally, we explored whether the defective LETFs-dependent gene expression in livers with AH results in an abnormal plasma footprint of the corresponding proteins. We thus collected plasma from controls (n = 15) and patients with AH (n = 10) and performed mass spectrometry. Among the 288 plasma proteins detected in plasma of both controls and AH patients (Supplementary Fig. 11a), 60 corresponded to liver-secreted proteins (Supplementary Fig. 11b) which gene expression was altered in AH livers (Supplementary Fig. 11c). Importantly, 21 of these proteins belong to the footprint of LETFs altered in AH (Supplementary Fig. 11d) and correlated with changes in hepatic gene expression (Supplementary Fig. 11e). Once validated in large cohorts, these peripheral footprints could be useful for prognosis, patient stratification or personalized treatment allocation in future clinical trials.

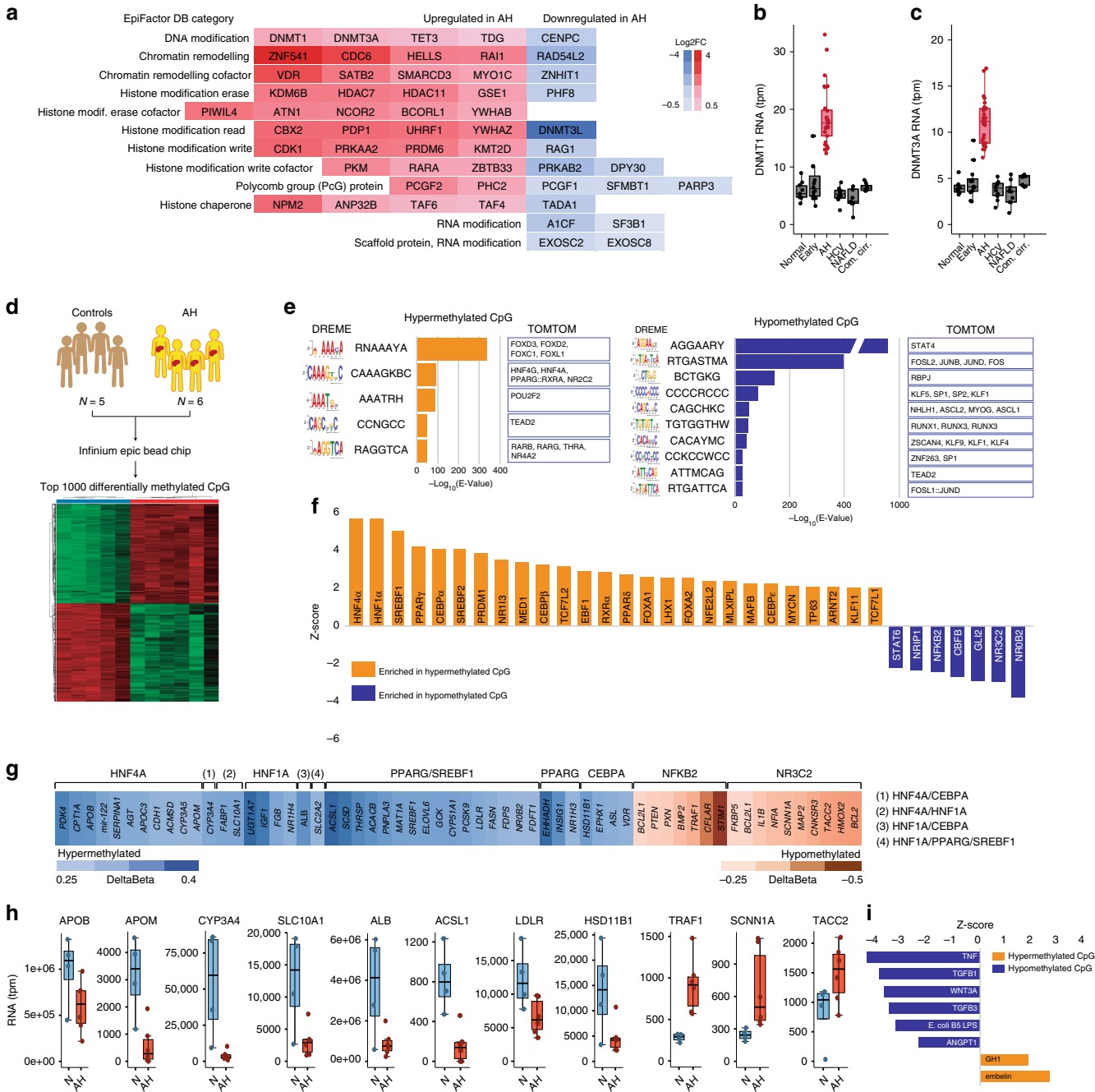

**Fig. 9** DNA hypermethylation of HNF4α-targets in AH patients. **a** Heatmap of Log Fold Changes in the expression of main epigenetic modulators in AH patients. Genes are organized by the 12 family of factors described in EpiFactor Database. **b**, **c** mRNA abundance in transcripts per million (tpm) from normal livers, AH livers and livers of non-alcohol-related chronic disease of **b** DNA Methyl-Transferases *DNMT1* and **c** *DNMT3A*. **d–i** DNA extracted from 5 Normal and 6 AH livers was bisulfite treated and hybridized in Illumina Infinium MethylationEPIC chip. **d** heatmap of top 2000 hyper or hypomethylated CpG islands. **e** DREME and TomTom algorithms (MEME-ChIP suite) were used to search for de novo transcription factor binding sites (tfbs) in hyper and hypomethylated regions and to identify transcription factors known to match these tfbs, respectively. **f** Differentially methylated regions were gene annotated (nearest-feature) and Ingenuity Pathway Analysis (IPA) was used to predict which transcription factor are predicted to be an upstream regulator genes with DM CpGs. Intensity of the enrichment is presented as Z-Score (*p* < 0.01). **g** Selected TF target genes delta-β changes: values are expressed with blue-color gradient if hypermethylated and brown-color if hypomethylated. **h** RNA sequencing of the same samples used in methylation chip was used to validate potential functional impact of hyper/hypomethylation on gene expression. **i** IPA analysis of soluble factors upstream the hyper and the hypomethylated region. Intensity of the enrichment is presented as Z-Score. For box-and-whisker plots in **b**, **c**, **h**: perimeters, 25th–75th percentile; midline, median. Gene expression levels are presented in transcripts per million reads (tpm)

In conclusion, this human-based translational study found that the development of hepatocellular failure in patients with AH is characterized by a dramatic decrease in HNF4α-dependent gene expression. The predicted decreased function was based on the integrated analysis of main target genes. TGFβ1, a key upstream transcriptome regulator in AH, induced the use of HNF4α P2 promoter in hepatocytes, which resulted in abnormal bile acid synthesis and defective metabolic and synthetic functions. In a

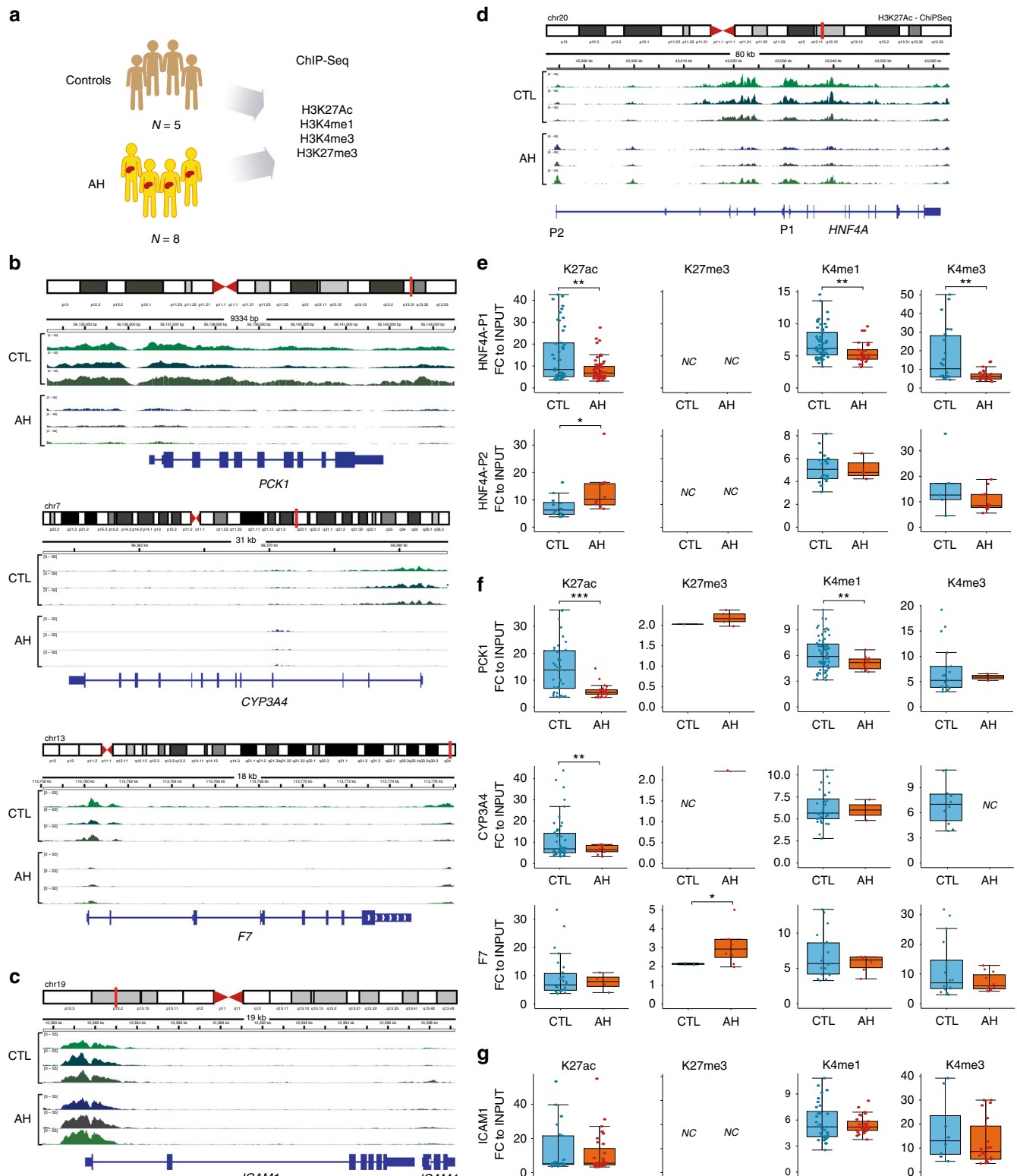

**Fig. 10** ChIP-seq shows decreased H3K27Ac and H3K4me1 in HNF4α-P1 and its targets and enhanced binding of H3K27Ac to HNF4α P2 promoter. **a** Data were obtained from ChIP-seq of Human Liver samples from normal (n = 5) and AH (n = 6) livers. Antibodies agains Histone 3 Lysine 27 acetylation (H3K27Ac), Histone 3 Lysine 4 mono and trimethylation (H3K4me1 and H3K4me3) and Histone 3 Lysine 27 trimethylation (H3K27me3) were used in the immunoprecipitation. Integrated Genome Viewer was used to visualize BigWig peak data. **b–d** Genomic view of sequencing reads present in loci of **b** HNF4α targets *PCK1*, *CYP3A4* and *F7*, **c** Non-HNF4α target *ICAM-1* and **d** *HNF4A*. **e–g** Box plot of fold changes (IP to Input) of all peaks called around the TSS of **e** HNF4A isoforms P1 and P2, **f** HNF4A targets *PCK1*, *CYP3A4* and *F7* and **g** *ICAM1*. Significance was determined by two-tailed Student *t* test in **e**, **f**, and **g**: *P < 0.05, **P < 0.01, ***P < 0.001. For box-and-whisker plots in **e, f, g**: perimeters, 25th–75th percentile; midline, median

recent work analyzing human samples by IHC, the authors describe a downregulation of HNF4α in patients with advanced decompensated cirrhosis[47]. In our study, patients with compensated cirrhosis and preserved synthetic function did not have a functional HNF4α deficiency. It is therefore plausible that other liver diseases characterized by decreased hepatocellular function and liver-related complications are characterized by defective HNF4α expression and/or function. Future analysis of transcription factor activity in AH should include decompensated patients as controls to better understand the specificity of these findings and mechanisms of liver failure. Gene polymorphisms in LETFs including HNF4α do not predispose to the development of AH, while AH livers are characterized by profound changes in DNA methylation state and chromatin remodeling in HNF4α-dependent genes. The results of this study suggest that targeting TGFβ1 and epigenetic drivers that modulate HNF4α-dependent gene expression could be beneficial in patients with AH.

## Methods

**Patients**. For Human RNAseq studies, Human liver samples were obtained from the Human Biorepository Core from the NIH-funded international InTeam consortium (7U01AA021908-05). Patients with early alcoholic steatohepatitis (ASH) were obtained from Cliniques Universitaires Saint-Luc (Brussels, Belgium). All patients included gave written informed consent and the research protocols were approved by the local Ethics Committees and by the central Institutional Review Board of the University of North Carolina at Chapel Hill. A total of 79 patients were included. Patients were selected according to different clinically relevant stage groups: (1) patients with early ASH, who were non-obese with high alcohol intake, and presented mild elevation of transaminases and histologic criteria of steatohepatitis (ASH, $N = 12$); (2) patients with histologically confirmed alcoholic hepatitis (AH) who were biopsied before any treatment (AH, $N = 18$) and (3) explants from patients with AH who underwent early transplantation following a well-defined protocol[48] (exAH, $N = 11$). These groups were compared with fragments of non-diseased human livers ($N = 10$), patients with non-alcoholic fatty liver disease (NAFLD) according to Keiner's Crieria[49] and without alcohol abuse ($N = 9$) and from patients with non-cirrhotic HCV infection ($N = 10$) and compensated HCV-related cirrhosis ($N = 9$). Patients with malignancies were excluded from the study. Clinical characteristics of patients are described in Supplementary Table 1 and depicted in Fig. 1c. A selection of liver samples from patients with AH ($N = 6$) and fragments of normal human livers ($N = 5$), were used for Methylome and ChIP seq analysis. For IHCs analyses, normal and AH liver samples were obtained at the Division of Gastroenterology and Hepatology, Medical University of Graz, Austria. All patients had clinically and histologically confirmed AH ($N = 10$) and did not have any concomitant causes of chronic liver disease ($N = 10$)[50]. The study was approved by the Ethics Committee of the Medical University of Graz and performed in accordance with the Declaration of Helsinki, and all patients gave written informed consent.

**RNA extraction, sequencing and bioinformatic analysis**. Total RNA from flash-frozen liver tissue was extracted by phenol/chloroform separation (TRIzol, Thermox). RNA purity and quality were assessed by automated electrophoresis (Bioanalyzer, Agilent) and was sequenced using Illumina HiSeq2000 platform. Libraries were built using TruSeq Stranded Total RNA Ribo-Zero GOLD (Illumina). Sequencing was paired end (2 × 100 bp) and multiplexed. Ninety-four paired-end sequenced samples obtained an average of 36.9 million total reads with 32.5 million (88%) mapped to GRCh37/hg19 human reference. Short read alignment was performed using STAR alignment algorithm with default parameters[51]. To quantify expression from transcriptome mappings we employed RSEM[52]. Principal component analysis (PCA) was done using *made4* library[53]. Analysis of differential expression was performed using the *Limma* package[54]. Cyclic loess normalization was applied, followed by log transformation of the counts per million and mean-variance adjustment using the voom function. The Jonckheere–Terpstratest and Kendall correlation was used to check ordered differences gene among progressive disease stages. To agglomerate gene patterns along disease stages, Short Time-course Expression Miner (STEM) algorithm was used through on-line platform[55]. The output of STEM analysis is shown in Fig. 1e, f, Supplementary Fig. 2 and Supplementary Data 1. To uncover biological functions related to gene expression changes, Gene Ontology (GO) enrichment through gene set overlapping computation was done by means of rMATS, using the Canonical Pathways (CP) collection, which includes 1329 gene sets[56]. To identify in an unbiased way the transcription factors predicted to be directly involved on transcriptomic changes we apply two methods: (1) Transcription factor motif searching in gene promoters and proximal 5′ regulatory regions (±2000 bp from TSS) by means of Opossum on-line tool[57] and (2) Functional prediction of differentially expressed genes (DEG) by the use of Ingenuity Pathway Analysis (IPA, Qiagen), selecting among predicted upstream regulators, those involved in transcriptional

regulation (categories: "transcriptional regulator", "ligand-dependent nuclear receptor"). Only those hits found in both analyses were considered for the overlap. A scheme of the methods used and a Venn diagram of the overlap between Opossum and IPA outputs is shown in Supplementary Fig. 12. The statistic approach used to calculate the predicted activation state (IPA) was Z Score (ZS) and is used to infer likely activation states of upstream regulators based on comparison with a model that assigns random regulation directions. An overlap p-value to determine statistically significant overlap between the transcription factor target gene dataset and the DEG for each comparison was also calculated using Fisher's Exact Test. For this study, the selected transcription factors (Figs. 2c, d, 5l and 9f) showed an overlap p-value <0.01 for the comparison between Normal livers and Early ASH and an overlap p-value <0.005 for the comparison between Early ASH and AH. Opossum calculates two complementary scoring methods to measure the over-representation of transcription factor binding sites: (1) Z-scores measures the change in the relative number of TFBS motifs in the DEG gene set compared with the background set, and (2) Fisher scores based on a one-tailed Fisher exact probability assessing the number of genes with the TFBS motifs in the foreground set *vs.* the background set. Since Opossum does not take into account the direction of the expression changes, the top upregulated and the top downregulated genes were scanned separately. Only those TF motifs with positive Z Score were considered. JASPAR database was used as the source of DNA binding profiles (Supplementary Fig. 12).

**HNF4α gene splicing analysis**. RNA-seq reads were trimmed to a uniform length of 75 bp using the FastxToolkit (http://hannonlab.cshl.edu/fastx_toolkit/). After read trimming, alignment of RNA-seq reads was performed with the STAR aligner (v2.5.2a) against the hg19 human genome. Resulting bam files were indexed with samtools for rMATS[58]. Differential expression of splice isoforms was completed using STAR alignment-StringTie-BallGown pipeline as described elsewhere[59]. To identify exon-specific expression, an alternate pipeline was used. First, reads were put through adapter trimming using TrimGalore (https://www.bioinformatics. babraham.ac.uk/ projects/trim_galore/). After the trimming, reads were aligned with the STAR aligner (v2.5.2a) against the hg19 genome. The resulting bam files were then put through the DEXSeq R Bioconductor package (v1.26.0 for DEXSeq and 3.3.1 for R) pipeline. To obtain raw read counts for each exon, we used a standard DEXSeq script for exon counting (dexseq_count.py), with minor modifications. The exons were categorized in the GenCode v19 release. After exon counting, individual R scripts were used to obtain the exon-specific expression profiles. All custom scripts are available upon request.

**Genomic DNA methylome analysis**. Genomic DNA (gDNA) was extracted from flash-frozen liver tissue with PureLink Genomic DNA Mini Kit (Thermo) and quantified using Nanodrop (Thermo). In total 1 µg of isolated gDNA was bisulfite converted, denatured and hybridized to Infinium Methylation Bead Chip, following the manufacturer protocol (Infinium MethylationEPIC kit, Illumina). BeadChips were imaged using an Illumina Scan System and intensity was determined by iScan Control Software (Illumina). Sample intensities were normalized using functional normalization from the *minfi* package (v1.24.0)[60]. Probes failing a detection p-value threshold (0.01) in at least 50% of samples were removed, as were probes identified as containing a SNP with a MAF >0.05. Differentially methylated probes were identified by applying *limma* (v3.34.3)[54] contrasts to M values (absolute change in beta value >0.1, FDR-corrected P-value < 0.05). Differentially methylated regions were identified using DMRcate (v1.14.0)[61] setting a threshold of absolute change in beta value in >0.1 and of Stouffer's value in <0.05.

**Chromatin immunoprecipitation-PCR (ChIP-PCR)**. Hep3B cells were plated in 150 mm dishes at semi-confluence ($10^6$ cells/dish) in 10% DMEM media. TGFβ1 (5 ng/ml) was added to the media 24 h before the fixation of the cells. The chromatin preparation and immunoprecipitation was performed using EZ-ChIP kit (Millipore, 17–371). Briefly, after treatment, cells were fixed with 1% formaldehyde (Sigma, F8775) for 10 min at room temperature. Quenching was performed with 125 mM glycine for 5 min. Cells were then washed twice with cold PBS and scrapped in 1.8 ml of cold PBS containing Protease Inhibitor Cocktail per dish. Cells were pelleted at low g and lysed with lysis buffer containing Protease Inhibitor Cocktail. Chromatin sonication was performed using a Misonix 100 W at 15% of power by giving 6 pulses of 5 s with intervals of 30 s in ice, to obtain DNA fragments between 200 and 800 bp. Each ml of sonicated chromatin contained the equivalent of 10 million Hep3B cells. 100µL of sonicated chromatin was used in each immunoprecipitation. Chromatin was then diluted to 1 ml with ChIP dilution buffer and pre-cleared with Protein G-agarose for 1 h at 4 °C in continuous rotation. Agarose pellet was then pelleted and discarded. A 10% of the sonicated, precleared chromatin was removed and saved as INPUT. The rest of the chromatin was immunoprecipitated using anti-RNA Polymerase II (Sigma-Aldrich, 05-623B, 1 µg) and anti-Phospho-c-JUN (Life, 711207, 5 µg), using normal mouse IgG as control (Sigma-Aldrich, 12-371B, 1 µg), overnight at 4 °C in continuous rotation. In total 60 µL of Protein G-agarose was then added and incubated for 1 h at 4 °C in continuous rotation. The agarose beads were then pelleted and washed with Low Salt Immune Complex Wash Buffer, High Salt Immune Complex Wash Buffer,

LiCl Immune Complex Wash Buffer and TE Buffer (twice). DNA was eluted in 200 μL of elution buffer. The same buffer was also added to INPUT samples. For reverse crosslinking samples, 8 mL 5 M NaCl was added to each sample and eluates were incubated for 5 h at 65 °C. Then, treatment with RNAse A (30 min at 37 °C) and with Proteinase K (2 h at 45 °C) was performed. DNA was then purified using spin columns, following manufacturer instructions. In order to avoid sonication and immunoprecipitation batch effect, DNA from two different experiments was pooled and aliquoted. The scanning of the genomic region around HNF4A-P2 Transcription Start Site (TSS), was performed by using JASPAR2018 Basic Sequence Analysis[62], selecting the matrix profiles for FOS:JUN heterodimer (MA0099.2 and MA0099.3) and for JUN (MA0488.1). We found 6 binding sites with high Relative Score (>85%) (Supplementary Fig 10). The oligonucleotides used for the amplification of HNF4a-P2 promoter and intron regions and of control GAPDH promoter can be found in Supplementary Table 2. Real time PCR was performed in triplicate.

**ChIP-seq of Histone marks.** ChIP-seq was performed in Mayo Epigenomics Development Laboratory (EDL)[63]. ChIP-seq with the liver tissue from 5 controls and 7 severe AH explants (provided by University of Lille, France) were done for four histone modifications, using antibodies against histone H3 Lysine 27 acetylation (H3K27ac, Cell Signaling #8173), histone H3 Lysine 27 tri-methylation (H3K27me3, Cell Signaling #9733), histone H3 Lysine 4 mono-methylation (H3K4me1, EDL, Mayo Clinic, Lot#1) and histone H3 Lysine 4 tri-methylation (H3K4me3, EDL, Mayo Clinic, Lot#1). For the next-generation sequencing, ChIP-seq libraries were prepared from 10 ng of ChIP and input DNAs with the Ovation Ultralow DR Multiplex system (NuGEN). The ChIP-seq libraries were sequenced to 51 base pairs from both ends using the Illumina HiSeq 2000 in the Mayo Clinic Medical Genomics Core. Data were analyzed by the HiChIP pipeline[64]. Briefly, reads were aligned to the hg19 genome assembly using BWA and visualized using the Integrative Genomics Viewer (IGV). Mapped reads were post-processed to remove duplicates and pairs of reads mapping to multiple locations. The MACS2 and Sicer algorithm was used for peak-calling in relation to the input DNA. IGV was then used to visualize H3K27ac peak changes on individual genes in this study.

**Human primary hepatocytes and cell lines.** Primary human hepatocytes were purchased from Lonza. They were thawed in thawing medium (MCHT, Lonza), plated in plating medium (MP, Lonza), and cultured in maintenance medium (MM, Lonza). PHH were seeded on collagen-coated 12- or 6-well plates (Corning), allowed to attach for 4 h, and then overlaid with Matrigel (0.3 mg/mL; Corning). In silencing experiments, transfection was done 6 h before Matrigel overlay and cells were kept in reduced serum media (OptiMEM, Gibco) during that time. Cells and/ or supernatant were collected at the indicated time points. HepG2 and Hep3B cells were purchased from ATCC and were mycoplasma-free. They were expanded in Dulbecco's Minimum Essential Media (DMEM, Gibco) supplemented with 10% Fetal Bovine Serum (FBS, Gibco), 1unit/mL Penicillin (Gibco), and 1 μg/mL Streptomycin (Gibco). When indicated, cells were serum-starved (1% FBS DMEM) 2 h prior drug incubation. In silencing experiments, transfection was done 24 or 48 h before treatment and cells were kept in OptiMEM for 6 h after transfection and then in 1% FBS DMEM until harvesting. HepaRG cells were purchased from ATCC and were mycoplasma free. They were incubated for 2 weeks in 2% DMSO Williams E. HepaRG cells were grown in William's E medium supplemented with 10% FBS, 100 U/mL penicillin, 100 μg/mL streptomycin, 5 μg/mL insulin, and 50 μM hydrocortisone hemisuccinate. After 2 weeks the medium was supplemented with 2% dimethyl sulfoxide (DMSO) and the cells were cultured for 2 more weeks. For hepatocyte de-differentiation experiments, HepaRG cells were detached and seeded at low confluence in the absence of DMSO[65].

**RNA extraction and Real Time Polymerase Chain Reaction.** RNA from human biopsies, for Real Time Polymerase Chain Reaction (RT-PCR) experiments was extracted with Qiagen AllPrep DNA/RNA/Protein kit (Qiagen) following manufacturer's instructions. For experiments with cell lines and primary hepatocytes, RNA was extracted by phenol/chloroform method (TRIzol, Invitrogen). Concentration and purity was assessed by spectrophotometry (Nanodrop, Thermo). In total 1 μg of total RNA was used for reverse transcription reaction using Maxima First Strand cDNA Synthesis Kit for RT-qPCR with dsDNase (Thermo) following manufacturer protocol. RT PCR of 50 ng of cDNA was performed in a 96 well plate, using a CFX96 Real Time PCR detection system (BIO-RAD) and fluorescent double-stranded DNA-binding dye (SsoAdvanced Universal Sybr Green Supermix, BIO-RAD). Sequence of custom designed primers (Primer3 software) are in Supplementary Table 2. The comparative CT method ($2-\Delta\Delta Ct$) was used to determine fold changes in mRNA expression compared a control group after normalization to an endogenous reference gene (Ribosomal Protein L4, RPL4).

**Protein extraction and Western Blot.** Liver tissue fragments and cell pellets were lysed in RIPA buffer (150 mM NaCl, 50 mM Tris pH 7.5, 0.1% SDS, 1% Triton X-100) with the addition of 40 mM DTT, protease inhibitor cocktail (Complete, Roche) and phosphatase inhibitors (1 mM $Na_3VO_4$, 2 mM NaF and 2 mM β-glycerophosphate) just before protein extraction. For liver extracts, ratio 1:20 (mg: μL) was used, and tissue was sonicated (5 cycles of 20 s with a 50 W probe sonicator

at 20% Amplitude). In indicated cases, nuclear/cytoplasm fractionation was made by using the NE-PER kit (Thermo), following the manufacturer protocol. For western blot, 20–40 ug of protein extract was denatured with Laemli buffer (AlfaAesar), boiled (95 °C for 3 min), loaded in SDS-PAGE system (BIO-RAD), run until complete separation, transferred to a nitrocellulose membrane (0.2μm pore diameter, BIO-RAD). Membranes were blocked for 1 h at room temperature with 5% non-fat milk in 0.1% Tween20-Tris Buffered Saline (T-TBS). After overnight incubation with primary antibodies (Supplementary Table 3), membranes were washed three times with T-TBS and incubated with Near-Infrared Florescent secondary antibodies (IRDye 680CW Goat anti-Rabbit and/or IRDye 800CW Goat anti-Mouse, LiCOR) for 1 h at room temperature and washed twice with T-TBS and finally rinsed with TBS. Membranes were imaged using an Odissey CLx Imager (LiCOR). For loading control of nuclear extracts, nitrocellulose membrane was stained with REVERT Total Protein Stain (LiCOR). A representative band was selected for western blot images in Figs. 2 and 3. Uncropped blots can be seen in Supplementary Fig. 13.

**Silencing of HNF4α isoforms P1 and P2 and of SMAD4.** The sense-strand sequences of the siRNAs were as follows: si-HNF4α-P1 (UUGAGAAU-GUGCAGGUGUU-dTdT), si-HNF4α-P2 (GCTCCAGTGGAGAGTTCTT-dTdT) and Scr (GCTGAGTAGAGTGTCCCTT-dTdT). SMAD4 siRNAs were purchased from Life (Assay Id s8403 and s8405). The effective working concentration of siRNA was 20pM in primary hepatocytes and 10pM in HepG2/Hep3B cells. Transfection of siRNAs was performed by the use of Lipofectamine-RNAiMAX (Invitrogen) following the manufacturer recommendations. This protocol showed 70–85% of silencing efficiency (mRNA and Protein level) at 24 and/or 48 h.

**Overexpression of HNF4α isoforms P1 and P2.** For overexpression of HNF4α-P1 dependent isoforms, ORFs of human HNF4α2 and α8 isoforms were cloned in pcDNA6 (Invitrogen) vectors under the CMV promoter. Plasmids were transfected at the indicated doses in HepG2 cells using Lipofectamine 3000 (Invitrogen) following standard manufacturer protocol.

**Cell culture treatments.** TGFβ1 (5 ng/mL, R&D Systems) or amphiregulin (AREG, 50 nM, Sigma Aldrich) were added immediately before Matrigel overlay and mRNA or protein were collected at the indicated time points. For was used. For proteasome inhibition, MG132 (10 μM, Calbiochem-EMD Millipore) was added 45 min prior to cell harvesting. Treatments with TGF-β RI Kinase Inhibitor VI (5 nM, SB431542, Calbiochem-EMD Millipore), TAKI Inhibitor (0.5 or 1 μM, NG25 trihydrochloride, Axon), EGFR inhibitor (3 μM, PD153035, Calbiochem-EMD Millipore), MEK Inhibitor (10 μM, UO126, Promega), c-SRC inhibitor (10 μM,PP2, Calbiochem-EMD Millipore), Rosigiltazone (10 μM, Sigma) and were performed after of 2 h starvation (1% FBS DMEM) and 45 min before TGFβ1 treatment.

**Biliary acid quantification.** Cryopreserved human primary hepatocytes (Lonza) were plated overnight on collagen-coated 96-well plates at $2 \times 10^4$ cells per well in MM (Lonza) and collected after 24 and 48 h of siRNA transfection. Total bile acids were measured following the protocol supplied in the Total Bile Acid Assay Kit available from Cell Biolabs (San Diego, Ca). Absorbance data was collected using the SpectraMax M2 (Molecular Devices, Sunnyvale, CA, USA) microtiter plate reader. The total bile acids were calculated by extrapolating test values to a calibration curve as described in the assay kit. The levels of glycochenodeoxycholate were measured by mass spectroscopy, as described elsewhere[66]. Briefly, 100 μL of acetonitrile was added to 50 μL of cell culture. The samples were vigorously vortexed and then centrifuged ($22,000 \times g$, 2.5 min). The supernatant fraction was diluted 1:10 in 20% acetonitrile in $H_2O$ for analysis by LC–MS/MS. Glycochenodeoxycholate (Sigma) was used to prepare a standard curve (1 nM–10,000 nM). The concentration of glycochenodeoxycholate in the media was determined by linear regression analysis.

**Glucose production assay.** Cryopreserved human primary hepatocytes (Lonza) were plated overnight on collagen-coated 12-well plates at $1 \times 10^5$ cells per well in MM (Lonza). 24 h after plating, cells were serum-starved in DMEM base medium (Sigma) supplemented with 1 g/L glucose (Sigma), 3.7 g/L sodium bicarbonate (Sigma), and 4 mM L-glutamine (Corning) overnight, followed by 24 h incubation in 0.3 ml glucose-production medium: DMEM base with 2 mM glutamine, 3.7 g/L sodium bicarbonate, 15 mM HEPES (ThermoFisher), 20 mM lactate (Sigma), 2 mM pyruvate (Fisher) and 0.1 mM pCPT-cAMP (Sigma). After 24 h, 50 μL of medium was removed for glucose detection with Invitrogen Glucose Colorimetric Detection kit (#EIAGLUC), according to manufacturer's protocol, and read on a plate reader (Multiskan GO, Thermo-Scientific). Because hepatocytes were extensively washed prior to cell incubation in glucose-free media for this assay, the only potential source of glucose in the media is hepatic production. The prolonged culture of cells in low glucose media prior to the assay depletes hepatocytes of glycogen stores. The media used during this assay contains high concentrations of gluconeogenic substrates, primarily lactate, favoring gluconeogenesis[67,68].

**Model of alcoholic liver disease**. Male mice (C57BL/6 J, 20–25 g, 12 weeks of age) were obtained from the Jackson Laboratory (Bar Harbor, ME) and housed in a temperature-controlled environment with a 12-h light-dark cycle and were given free access to regular laboratory chow diet and water. All studies were approved by the Institutional Animal Care and Use Committee at UNC-Chapel Hill. The model of acute on chronic alcoholic liver injury in mice was performed as described elsewhere[69]. CCl4 (>99.5% pure) and olive oil vehicle were from Sigma (St. Louis, MO), ethyl alcohol (EtOH) (190 proof, Koptec) was from VWR (Radnor, PA). Procedures for CCl4–induced liver fibrosis were as detailed elsewhere[70]. Mice were intra-peritoneally injected (15 ml/kg) with CCl$_4$ (0.2 ml/kg) or olive oil vehicle-alone 2 × week for 6 weeks. After 6 weeks of CCl$_4$ treatment, animals underwent surgical intragastric intubation[71]. Following surgery, mice were housed in individual meta-bolic cages and allowed 1 week to recover with ad libitum access to food and water. Animals had free access to water and non-nutritious cellulose pellets throughout the remaining study. Alcohol groups received high-fat diet containing ethyl alcohol as detailed elsewhere[71]. Alcohol was delivered continuously through the intragastric cannula initially at 16 g/kg/day and was gradually increased to 25 g/kg/day. All animals were given humane care in compliance with the National Institutes of Health guidelines and alcohol intoxication was assessed to evaluate the development of tolerance. Experimental groups are detailed in the legend to Supplementary Fig. 4. At the end of the study, mice were anesthetized with pentobarbital (50 mg/kg, i.p.) and sacrificed via exsanguination through the vena cava, which was the site of blood collection. Tissues were excised and snap-frozen in liquid nitrogen.

**Liver histopathological evaluation**. Tissues were embedded in paraffin, sectioned at 5 μm, and stained with hematoxylin and eosin (H&E) or Sirius red. For Oil red O staining, tissues embedded in optimal cutting temperature compound were sectioned at 10 μm. Liver pathology was evaluated in a blind manner by two independent pathologists and scored as detailed elsewhere[72]. For Sirius red and oil red O staining, quantitative analysis was performed using NIH ImageJ at 100 × magnification in 5 random fields.

**Immunohistochemistry**. Dewaxed 3 μm thick sections were stained with hematoxylin and eosin (H&E) or chromatrope aniline blue (CAB) connective tissue stain according to standard protocols. All slides were reviewed by a single pathologist (CL). For immunohistochemistry paraffin sections were dewaxed and rehydrated. Conditions of anti-human PPARγ, HNF1α, FOXA1 (HNF3α), RXRα, and HNF4α (P1 and P2-isoforms) immunohistochemistry are summarized in Supplementary Table 4. After immunohistochemical staining sections were counterstained with hematoxylin (Labonord, Templemars, France) and mounted with Aquatex (Merck, Darmstadt, Germany). Immunohistochemical signals were evaluated semi-quantitatively by the application of numerical scores, based on the intensity of the signal. For HNF4α, HNF1α and FOXA1, where the signal in AH patients was also cytoplasmic, scoring was made separately for cytoplasmic and nuclear signals.

**Single nucleotide polymorphism (SNP) cohorts**. The AH exploratory study data were obtained from a genome-wide association study of severe alcoholic hepatitis published in abstract form[73]. Patients with AH were recruited through the steroids or pentoxifylline for AH (STOPAH) trial[74]. Inclusion was based upon a clinical diagnosis of alcoholic hepatitis, modified Maddrey's discriminant (mDF) ≥ 32, current excess alcohol consumption, recent onset of jaundice and exclusion of other causes of decompensated liver disease. In order to reduce population admixture only patients with self-reported "white" ethnicity were included. In order to maximize phenotypic differences in the exploratory genome-wide association stage, in accordance with the study design, patients with biopsy-proven disease and the most severe liver injury, as indicated by the mDF, were preferentially selected for inclusion in the exploratory cohort (n = 332). Controls with a background of alcohol dependence but with no evidence of liver injury were recruited via the University College London Consortium (n = 318). The majority had been drinking hazardously for over 15 years and were actively drinking at the time of enrollment and the absence of significant alcohol-related liver injury was confirmed on liver biopsy. The remainder had no historical, clinical or radiological features suggestive of significant liver injury either at presentation or during prolonged follow-up. All were of English, Scottish, Welsh or Irish descent with a maximum of one grandparent of white European Caucasian origin. None of the individuals was related.

**SNP analysis**. Samples were genotyped using the Illumina HumanCoreExome beadchip at the Wellcome Trust Sanger Institute in Cambridge, UK. Quality control and analysis of data were performed in PLINK v1.90[75]. Individual data were quality controlled such that those with genotyping rate <98%, sample heterozygosity >3 standard deviations from the population mean, relatedness determined by pi-hat >0.185 or phenotypic and genotypic sex mismatch were excluded. Markers with genotyping rate <98% or with a probability of deviation from Hardy-Weinberg equilibrium <1 × 10-6 were also excluded. Population principal components were calculated using a linkage-disequilibrium pruned data set of common variants in PLINK v1.90, associations between principal components and case-control status were tested in R. The resultant dataset was phased using ShapeIt v2.r790 and imputed against the 1000 genomes project reference dataset using

IMPUTE v2.3.2[76]. The imputed genotypes were hard-called using a probability threshold of 0.9 and quality control filters were applied – missingness <5%, minor allele frequency >1% and deviation from Hardy–Weinberg equilibrium $<1 \times 10^{-6}$. Associations with case-control status were tested in PLINK v1.90 specifying the principal components associated with case-control status as covariates. Only autosomal data was analyzed. For the single marker analyses, key transcription factors and related genes were identified through the primary analysis of RNAseq data. Genomic coordinates for the coding regions of these genes, including 3′ and 5′ upstream regions, were obtained from Ensembl Biomart. Single nucleotide polymorphisms (SNPs) falling within these genetic loci were extracted from the AH study data. Analyses in the AH study data were limited to SNPs with a minor allele frequency >1%. In order to control the false discovery rate a Bonferroni correction was applied based upon 105 independent tests in the directly genotyped dataset. Thus, the study-specific threshold for significance for tests of single SNPs was 0.0005. For all other tests p < 0.05 was considered significant. For significantly associated SNPs predicted effects on protein structure were predicted using SIFT[77] and Polyphen[78], expression quantitative train locus (eQTL) tests were conducted using GTeX[79]. For gene- and pathway-based association tests were performed using study summary statistics in MAGMA v1.06[80] in accordance with recommended procedures using reference files available at https://ctg.cncr.nl/software/magma via the FUMA online server (http://fuma.ctglab.nl/,). Pathway-based association testing was achieved by defining a biological pathway incorporating the gene targets of interest.

**Mass Spectrometry of plasma samples for proteomic analysis (LC–MS/MS)**. Plasma samples from Control subjects (N = 10, 10 μL each) and plasma from patients with AH (N = 15, 10 μL each) were pooled and protein concentration of each group was determined by Qubit fluorometry. In total 10 μL of protein from each pooled sample was depleted on a Pierce™ Top 12 Abundant Protein Depletion Spin Column (Thermo Scientific) according to manufacturer's protocol. Depleted samples were buffer exchanged into water on a centrifugal concentrator (Spin X, Corning) using a 5 kD molecular weight cut off and quantified by Qubit fluorometry (Life Technologies). 50 μg of each sample was reduced with dithiothreitol, alkylated with iodoacetamide and digested overnight with trypsin (Promega). The digestion was terminated with formic acid. Each digested sample was processed by solid phase extraction using an Empore C18 (3 M) plate under vacuum (5in Hg). Briefly, columns were activated with 400 μL 95% acetonitrile/0.1% TFA X2, and then equilibrated with 400 μL 0.1% TFA X4. Acidified samples were samples were loaded and columns were washed with 400 μL 0.1% TFA X2. Peptides were eluted with 200 μL 70% acetonitrile/0.1% TFA X2 and then lyophilized for further processing. 2 μg of each sample was analyzed by nano LC-MS/MS with a NanoAcquity HPLC system (Waters) interfaced to a Q Exactive (Thermo-Fisher). Peptides were loaded on a trapping column and eluted over a 75 μm analytical column at 350 nL/min using a 3 h reverse phase gradient. Columns were packed with Luna C18 resin (Phenomenex). The mass spectrometer was operated in data-dependent mode, with the Orbitrap operating at 60,000 FWHM and 17,500 FWHM for MS and MS/MS respectively. The fifteen most abundant ions were selected for MS/MS. Data were searched using a local copy of Mascot with the following parameters: *Enzyme*: Trypsin/P; *Database*: SwissProt Human. *Fixed modification*: Carbamidomethyl (C); *Variable modifications*: Oxidation (M), Acetyl (N-term), Pyro-Glu (N-term Q), Deamidation (N/Q); *Mass values*: Monoisotopic; *Peptide Mass Tolerance*: 10ppm; *Fragment Mass Tolerance*: 0.02 Da; *Max Missed Cleavages*: 2. Mascot DAT files were parsed into Scaffold (Proteome Software) for validation, filtering and to create a non-redundant list per sample. Data were filtered using at 1% protein and peptide FDR and requiring at least two unique peptides per protein. Normalized Spectral Abundance Factor (NSAF) values were used to obtain the fold change between Normal and AH groups. For unbiased searching of secreted protein coding genes from RNA-seq data, Retrieve/ID mapping on-line tool of UniProt was used (filters "signal peptide" and "NOT transmembrane domain")[81].

## Data availability

The RNA-sequencing and Methylomic raw data (Figs. 1, 2, 3a, 4, 7, 9 and Supplementary Figs 2,3 and 5) have been deposited in the Database of Genotypes and Phenotypes (dbGAP) of the National Center for Biotechnology Information (United States National Library of Medicine, Bethesda, MD) under accession number phs001807.v1.p1. The GWAS Summary used to generate Fig. 8 and Supplementary Data 3 and 5 and the ChIP-seq peak calling data in Fig. 10 are publicly available (https://doi.org/10.5281/zenodo.3233952).

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

## Acknowledgements

CLC Genomics Workbench Software licensed through the Molecular Biology Information Service of the Health Sciences Library System, University of Pittsburgh was used for data analysis. This work was mainly supported by NIH/NIAAA funded Consortia "Integrated approaches for identifying molecular targets in alcoholic hepatitis" InTEAM (U01AA021908) (R.B., P.M., P.S-B.,I.R.,J.Cbl). This work was supported in part by: NIH /NIAAA (R01AA023781), USA (C.W.); Hepacare Project, Fundación La Caixa, Spain (M.A.A., C.B. and M.U.L); Fond national de la recherche scientifique (FNRS J.0146.17) and Fond de la recherche scientifique médicale (FRSM T.0217.18), Belgium (P.S.); NIH/ NCATS (UH3TR000503) and EPA (STAR 83573601), USA (D.L.V. and L.A.T.); MRC, UK (MK/K001949/1) and NIH/NIAAA, USA (UO1AA018663) (J.M.); NIH/NIAAA (1U01AA021908-01-33490), Instituto de Salud Carlos III (PI17/00673) and Miguel Servet (CPII16/00041) and "Una manera de hacer Europa" program, European Regional Development Fund (ERDF), EU (P.S-B.); National Institute for Health Research Imperial Biomedical Research Centre and NIHR Health Technology Assessment Grant 08-14-44 (M.R.T.); NIH T32, DK007052, USA (L.R.E.); NIH/NIAAA (1U01AA021908) and AFEF (P.M., L.D.,A.L.). *Acronyms:* NIH: National Institutes of Health; NIAAA: National Institute of Alcohol Abuse and Alcoholism; MRC: Medical Research Council; NCATS: National Center for Advancing Translational Sciences; EPA: United States Environmental Protection Agency; STAR: Science to Achieve Results; NIHR: National Institute for Health Research; AFEF: Association Française Pour l'Etude du Foie.

## Author contributions

R.B. and J.Ar. conceived and designed the study, analyzed and interpreted data and wrote the manuscript. J.Ar., V.M., J.Cbz, M.V.-C., C.F., G.O., L.D., A.L., J. Al., P.S.-B., J. Cbl., C.L. and P.M. provided and collected study materials, samples and patient data. J.Ar., M.U.L., J.P.G., V.M., and A.B. performed in vitro experiments. S.F. and I.R. performed in vivo experiments. J.J.L, J.Cbz, and J.Ar. analyzed RNA-sequencing data. I.O.B., D.V.B., C.W. performed exon quantification and analyzed exon exclusion. L.A.V. and D.L.T. performed in vitro experiments related to bile acid secretion. L.R.E. and M.J.J. performed in vitro experiments related to glucose production. S.R.A., M.Y.M. and M.R.T collected, analyzed and interpreted GWAS data. J.M. and J. Ar. analyzed and interpreted methylome data. S.C., J.P.A and V.H.S. collected, analyzed and interpreted ChIP-seq data. C.L. performed immunohistochemistry on human samples. T.A. and J.L.G. performed partial hepatectomy and in vitro proliferation assays, respectively. M.A.A., P.M., S.P.M., C.W., P.S-B, I.R., V.H.S., C.B., M.R.T., J.M. and L.V. interpreted the data, help in the study design and reviewed the manuscript.

## Additional information

**Competing interests:** The authors declare no competing interests.

Josepmaria Argemi [1,2], Maria U. Latasa [3], Stephen R. Atkinson [4], Ilya O. Blokhin [5], Veronica Massey [6], Joel P. Gue [1], Joaquin Cabezas [6,7], Juan J. Lozano [8,9], Derek Van Booven [10], Aaron Bell [11], Sheng Cao [12],

Lawrence A. Vernetti[13], Juan P. Arab [12,14], Meritxell Ventura-Cots [1], Lia R. Edmunds[15], Constantino Fondevila[16], Peter Stärkel [17], Laurent Dubuquoy [18], Alexandre Louvet[18], Gemma Odena[6], Juan L. Gomez[11], Tomas Aragon[19], Jose Altamirano[20], Juan Caballeria[8,9], Michael J. Jurczak [15], D. Lansing  Taylor[13], Carmen Berasain [3,8], Claes Wahlestedt [5], Satdarshan P. Monga [11], Marsha Y. Morgan[21], Pau Sancho-Bru [8,9], Philippe Mathurin[18], Shinji Furuya[22], Carolin Lackner[23], Ivan Rusyn[22], Vijay H. Shah[12], Mark R. Thursz[4], Jelena Mann[24], Matias A. Avila[3,8] & Ramon Bataller [1,6]

[1]Division of Gastroenterology, Hepatology and Nutrition, Pittsburgh Liver Research Center, University of Pittsburgh Medical Center (UPMC), Pittsburgh, PA 15261, USA. [2]Liver Unit, Clínica Universidad de Navarra, University of Navarra, Pamplona 31008, Spain. [3]Hepatology Program, Center for Applied Medical Research (CIMA), University of Navarra, Pamplona 31008, Spain. [4]Division of Digestive Diseases, Department of Surgery and Cancer, Imperial College London, London SW7 2AZ, UK. [5]Center for Therapeutic Innovation and Department of Psychiatry and Behavioral Sciences, University of Miami Miller School of Medicine, Miami, FL 33136, USA. [6]Division of Gastroenterology and Hepatology, Departments of Medicine and Nutrition and Bowles Center for Alcohol Studies, University of North Carolina at Chapel Hill, Chapel Hill, NC 27516, USA. [7]Departament of Hepatology, Marqués de Valdecilla University Hospital, Santander 39008, Spain. [8]Centro de Investigacion Biomedica en Red, Enfermedades Hepáticas y Digestivas (CIBERehd), Madrid 28029, Spain. [9]Institut d'Investigacions Biomèdiques August Pi i Sunyer (IDIBAPS), Barcelona 08036, Spain. [10]John P. Hussman Institute of Human Genomics. Miller School of Medicine, University of Miami, Miami, FL 33136, USA. [11]Departments of Pathology and Medicine, Pittsburgh Liver Research Center, University of Pittsburgh School of Medicine, Pittsburgh, PA 15261, USA. [12]Division of Gastroenterology and Hepatology, Mayo Clinic, Rochester, MN 55905, USA. [13]University of Pittsburgh Drug Discovery Institute, Department of Computational & Systems Biology, University of Pittsburgh, Pittsburgh, PA 15261, USA. [14]Departamento de Gastroenterologia, Escuela de Medicina, Pontificia Universidad Catolica de Chile, Santiago, Chile. [15]Department of Medicine, Division of Endocrinology and Metabolism, Center for Metabolic and Mitochondrial Medicine, University of Pittsburgh, Pittsburgh, PA 15261, USA. [16]Liver Transplant Unit, Department of Surgery, Hospital Clinic, University of Barcelona, Barcelona 08036, Spain. [17]Service d'Hépato-gastroentérologie, Cliniques Universitaires Saint-Luc and Laboratory of He-patogastroenterology, Institut de Recherche Expérimentale et Clinique, Université Catholique de Louvain, Brussels 1200, Belgium. [18]Service des Maladies de l'appareil digestif, CHU Lille. Inserm LIRIC - UMR995, University of Lille, Lille 59000, France. [19]Department of Gene Therapy and Regulation, Center for Applied Medical Research, University of Navarra, Pamplona 31008, Spain. [20]Liver Unit, Department of Internal Medicine, Vall d'Hebron Institut de Recerca. Internal Medicine Department, Hospital Quiron Salud, Barcelona 08035, Spain. [21]UCL Institute for Liver and Digestive Health, Division of Medicine, Royal Free Campus, University College London, London WC1E 6BT, UK. [22]Department of Veterinary Integrative Biosciences, College of Veterinary Medicine and Biomedical Sciences, Texas A&M University, College Station, TX 77845, USA. [23]Medical University of Graz, Institute of Pathology, Graz 8036, Austria. [24]Newcastle Fibrosis Research Group, Institute of Cellular Medicine, Faculty of Medical Sciences, Newcastle University, Newcastle upon Tyne NE2 4HH, UK

