## [Peer Review File · Nature Communications]

Reviewers' Comments:

Reviewer #1:

Remarks to the Author:

The manuscript is well-written and contains some novel, interesting findings on the role of HNF4a in alcoholic hepatitis. The authors included an excessive number of different parameters in this report. This negatively affects the value of the report since many of the described biological events were just observational and not investigated in the follow-up experiments.

Major concerns:

1. What is the reasoning of including the animal models in this report if the authors used only six lines (page 7) to describe a few observational findings and general statements? This part of the manuscript is out of place and findings in animal models deserve a greater in-depth investigation.
2. The authors reported down-regulation of the lncRNA HNF-4A-As1 in the livers from patients with alcoholic hepatitis (Page 8). What is the role of this lncRNA?
3. The in vitro part of the report requires substantial revision (pages 9-10). The authors attempted to study alterations of HNF4a isoforms during the de-differentiation of hepatocytes. They examined the expression of several genes in primary human hepatocytes in vitro. A simple culturing of primary hepatocytes is not a proper model to study hepatocyte de-differentiation. Additionally, altered expression of a few genes is not a sufficient evidence of de-differentiation.
4. The part of the manuscript focusing on epigenetic alterations describes few random epigenetic alterations only, i.e. altered expression of DNA methyltransferase and few selected histone acetyltransferase genes, the presence of differentially methylated regions without investigation of their functional consequences, analysis of one H3K27ac transcription-activating mark, each of the needed to be investigated in greater details. The inclusion of this part in the manuscript is preliminary.
5. In addition to several dysregulated HNF4a-associated molecular pathways, it will be beneficial to explore the HNF4a-miR-122 axis alterations in alcoholic hepatitis.

Reviewer #2:

Remarks to the Author:

Argemi et al reported a comprehensive study on ALD. The paper is data rich and presented in a logical way. I have the following concerns.

- 1) The paper is should be divided into sections, eg. discussion section.
- 2) It will be informative and more readable for audience if the authors can prepare a flowchart for all the analysis of this work (although Fig1a shows the brief flowchart for RNA-seq analysis)
- 3) Abstract should present key data and results (eg. Sample size profiled).
- 4) Many datasets are used throughout the paper, however, some conclusions/statements were made without specifying the dataset used. For example, "We then analyzed the methylation status of nearly 800,000 loci in normal livers and livers from AH patients and found around 3,000 differentially methylated (DM) CpG - containing loci (Fig. 4g and Supplementary Table 5)". What was the sample size and significance level used in the differential methylation study?
- 5) GWAS did not find signals around HNF4a, and the authors hypothesize the mechanism of HNF4a is via DNA methylation and chromatin remodeling. It is important to check eQTLs and methQTLs in liver for HNF4a locus.

Reviewer #3:

Remarks to the Author:

Argemi et al show the involvement of liver enriched transcription factors in the development of alcoholic hepatitis (AH) using human samples, RNAseq, ChIPseq, plasma proteomics as well as mouse models and cells in culture. In particular, they find that the expression of P2 isoform of

HNF4a is increased while that of the P1 isoform is decreased in AH. PPAR-g activity is increased in early stages of the disease but not during AH. They further show that the increase in P2 HNF4a is mediated by TGFb1 and that pharmacological modulation of TGFb1 and/or PPARg alleviate the dysregulation caused by AH, suggesting a potential therapy for patients with AH.

As noted by the authors, there are 2 promoters to the HNF4A gene that drive the expression of two sets of isoforms that differ in their N-termini. The major transcripts driven by the P1-promoter (P1-HNF4a, HNF4a1 and HNF4a2 in particular) are the best characterized and widely accepted to be the major driver of hepatocyte-specific gene expression in the adult liver; they also serve as tumor suppressors. P2-HNF4a in contrast is expressed in the fetal liver and liver cancer but generally not observed in the normal adult liver. The finding that P2-HNF4a is upregulated in alcoholic hepatitis is novel and has important clinical implications. The finding that TGFb1 upregulates P2-HNF4a is also tantalizing and opens up whole new areas of investigation.

Overall, the work is very well done and well presented. The authors perform a massive amount of work on human liver samples at various stages of ALD but there are many issues (mostly minor) that need to be addressed.

General:

1. Summary and elsewhere – The authors show that TGFb1 increases P2 protein expression and RNA, a very intriguing result. However, they have not shown that TGFb1 acts directly on the P2 promoter. It could be that TGFb1 decreases P1 expression, resulting in an increase in P2 expression. For example, Briancon et al JBC 2004 (PMID: 15159395) showed that P1-HNF4a represses the P2 promoter. The ambiguities in the mechanism responsible for the upregulation of P2-HNF4a TGFb need to be clarified.
2. The authors conclude that PPARg agonists are partially preventive of the TGFb1 effect, but that effect is not very convincing. Increase of P1 protein by PPARg agonists (ROSI and PIO) is not obvious from the blots Fig. in 3q nor is the rescue of the negative effect of TGFb1 on P1. There is no quantification and it is not clear how reproducible this result is. Suppression of P2 by ROSI/PIO is clearer as are changes in RNA in the other panels but again some sort of quantification is needed.
3. Down regulation of P1-HNF4a via EGF-like molecule AREG could be due to activation of Src kinase by EGF. Others have shown that Src (downstream of EGF) phosphorylates and subsequently down regulates the P1-HNF4a protein but not P2-HNF4a (Chellappa et al PNAS 2012 PMID: 22308320).
4. The antisense HNF4a between P1 and P2 is examined in various panels but the relevance is not completely clear and it is not discussed. For example, Fig. 2c shows the down regulation of the AS and upregulation of P2-HNF4a, suggesting that P2-HNF4a might be repressing the AS. (P1-HNF4a does not appear to be affected in 2c.) If this is in fact the case, what role in AH do the authors propose for the AS?
5. Likewise, Fig. S4 – the predicted splicing events are interesting but the relevance to AH is not discussed.

Specific:

6. P.10, Fig. 3f-i – synergistic action via the TGFbRI/TAK1 – the TAK1 inhibitor does not affect P1 expression, although it does affect P2 – but p values for the appropriate comparisons are not given (TGFb1 +/- TAK1 inhibitor) for P2 RNA (they look like they are significant but this needs to be proven)
7. Fig. S7 c-f. Which HNF4a is being probed for?
8. The effect of AREG (EGF-like) on P2 expression is not very convincing (Figure 3e). This needs to be quantified? Also, the actin blots are oversaturated impeding proper normalization. In contrast, in Fig. 3e and 3q the combined effect of TGFb1 and AREG is substantial especially on P1-HNF4a but there is no mention of how the EGF and TGFb pathway might be synergizing. Similarly, Fig. S7ab – TGFb1+AREG synergistically decrease P1 but, unlike the statement on p. 10, in Fig. S7b

does not show P2 RNA up in the presence of both. Rather it shows an increase only with TGFb.

9. Fig. 1e shows that PPARg is decreased in AH (second most downregulated TF after HNF4a) but this result is not discussed in the text. PPARg does not go down in early ASH, when there is apparently no effect on HNF4a expression (Fig. 1e), but does decrease in AH, when there is increased P2-HNF4a (Fig. 1f). Could P2-HNF4a be down-regulating PPARg activity??

10. Fig. 2m – an increase in P1 protein in the P2 KD is not apparent (the RNA effect is evident in 2n).

11. Fig. 2n suggests that P2-HNF4a represses P1 expression (but not the AS). Briancon et al JBC 2004 (PMID: 15159395) showed the reciprocal effect – namely, that P1-HNF4a represses the P2 promoter. This should be mentioned.

12. Fig. 2r, 2s, p. 10 – synthesis and secretion of bile acids and glucose per se have not been examined. The authors have only measured the levels of these compounds in the primary hepatocytes – changes in those levels could be achieved by a number of different mechanisms.

13. Fig. S6a – TCF3,4 and LEF1 are up in AH and AH explants; P2-HNF4a is also up. Others have reported that P1- and P2-HNF4a interact in a differential fashion with TCF4 in a colon cancer model (Vuong et al 2015 MCB, PMID: 26240283). The authors might want to see whether this paper is relevant to their story.

14. P. 11, Fig. 3l – need to show samples without TGFb to show that “Hepatocytes transfection [sic – should be transfected] with siRNA targeting P2 isoforms abolished TGFb1-mediated suppression of HNF4a-P1.”

15. Supp Table 7 – catalog numbers for the antibodies must be given

Editorial comments:

1. Abstract- last 2 sentences and elsewhere – “HNF4a-depending gene expression” should be “HNF4a-dependent (-driven) gene expression”

2. Some labels in the main figures are barely visible -- e.g., Fig. 3f-I – labels not very visible unless the pdf is zoomed in. In the paper version TGFbRI and TGFBRi cannot be easily distinguished. Fig. 4g labels are too small

3. Page 8- P2 HNF4a is introduced but P1 isoform is not described

4. Fig. 1b – text mentions NASH but the figure shows only NAFLD

5. Page 9 -“furtherly” ?

6. Fig. 1g, 1h – are not referenced in the text properly –Fig. 1g is referenced after Fig. 3o, Fig. 1h is not referenced at all

7. Fig. S4d – label for “AH livers” is missing

8. S4q – presentation is confusing. Means suggest more exon 8 in Control v. AH but AH ratio is 4.24

9. Fig. 2q – Cyp7A1 and 27A1 are not mentioned in the text

10. P. 10 -- “main” should be “potential” upstream regulators

11. Fig. S5d (RXR) is not discussed in the text

12. P. 11 Fig. 3p does not show: “The PPAR-g agonists rosiglitazone and pioglitazone decreased

the abundance of P2 isoforms and increased P1 isoforms (Fig. 3p).” Should be Fig. 3q?

13. Pg 11- “ The effect of rosiglitazone on HNF4a-P1 mRNA levels was dose dependent (Fig. 3t)” - this is shown in 3u, not 3t

14. P. 11 – Fig. 3s should be 3t

15. Fig. 3q – is the reason for the HNF4a1-6 and 7-9 nomenclature used in the bottom blots instead of the P1- and P2-HNF4a as in the rest of the paper?

16. Pg. 11 PCK1, ALB mRNA panels in Fig 3t not referenced

17. Fig. S9d and e are not referenced

18. Fig. S10 is not referenced in the text

19. Fig. S1 legend – not all p values are on the bottom left of the plots

20. Fig. S3 – red and blue should be defined. MEA should also be defined

21. Fig. S4e – padj and Pearson’s – up for Exon 1D compared to what?

22. Fig. 2 legend – in discussing genes, “that” should be used instead of “who”

Reviewer #4:

Remarks to the Author:

This is an ambitious study examining changes in gene expression and the underlying mechanisms associated specifically with alcoholic hepatitis. By combining multiple 'omics approaches (RNA-seq, ChIP-seq, DNA methylomics, GWAS, and plasma metabolomics) with several bioinformatics pipelines to study patients with liver disease, the authors identified a strong association between alcoholic hepatitis and down-regulation of HNF4a-dependent gene expression that appeared to be linked to increased TGFβ1 signaling. These associations were then tested in cultures of primary hepatocytes and hepatocarcinoma-derived cell lines. Genetic and pharmacological approaches verified that TGFβ1 signaling increased expression from an alternative promoter of HNF4a, and that isoforms derived from this (usually) fetal promoter were responsible for decreased expression of genes associated with mature hepatocyte functions. PPAR-gamma agonists were able to at least partially block these effects, providing an explanation for the reported benefits of these agents in the clinic. These findings are novel and should be of interest to the readers of Nature Communications.

Despite the complicated study design, the results are presented fairly clearly. This is achieved largely by restricting the conclusions to focus on the big picture or take home message from each experiment. However, there are some places where it would be better to include some additional details or discussion within the body of the manuscript. Specifically:

1. A weakness of this manuscript is that the major finding was not reproduced in the animal models of liver disease used in this study. Whereas earlier stages of ALD were recapitulated in the high fat diet/alcohol model with respect to activation of PPAR-gamma-dependent genes, later stages (modeled by alcohol/CCI4) did not show inhibition of HNF4a-dependent gene expression. This observation is very briefly addressed by the authors, but should be discussed more thoroughly.

2. Although the use of HepG2 and Hep3B cells is described in the Supplementary Materials and Methods and within the figure legends, it was not at all clear from reading the text of the manuscript that these cell lines were used. In fact, the way the cell culture experiments were described made it seem as though spontaneous de-differentiation of primary hepatocytes in

culture was the only in vitro model employed. It should be made clear when, and for what purpose, HepG2 and Hep3B cells were used. The impact of the differentiation state of these cells should be discussed in the context of comparing results to those obtained with primary cells.

3. The increase in HNF4a-P2 (fetal) isoforms is seen only in patients with AH (Figure 3C). Notably, this increase was not seen in patients with compensated HCV-related cirrhosis. The authors cited a recent study showing that forced overexpression of a mature HNF4a isoform can reverse cirrhosis in CCl4-treated rats (Nishikawa 2015). The present findings should be discussed in the context of the Nishikawa paper, and the meaning of the observed restriction of the P2/P1 imbalance to AH patients only should be addressed.

4. An increase in the fetal isoforms of HNF4a may reflect an increase in proliferation that would be beneficial in terms of liver regeneration. Is there any indication that the loss of biological functions is accompanied by increased indicators of proliferation?

5. Zhanxiang Zhou has shown that in animal models of ALD, the activity of HNF4a can be lost as a result of loss of zinc from the DNA binding domain zinc fingers. This post-translational modification is not seen by looking at mRNA or protein levels. The limitations associated with indirect measures of transcription factor activity should be addressed.

6. Please include the catalog numbers for the antibodies listed in Supplemental Table 7.

7. In the description of Protein Extraction and Western Blotting in the Materials and Methods section, 2 clarifications should be made. First, the units of the extract ratio are imprecise and should be changed to mg:µL or mg:L (whichever is correct). Second, it is unlikely that the "tissue was pestle and sonicated." Correct as necessary.

8. It is somewhat misleading to refer to 76 patients in groups of 9 to 18 as "a large series of patients" in the second paragraph of the manuscript.

9. The first line of the manuscript is confusing. This isn't a paper about (primarily) addictions, mortality or cirrhosis, so why are those the 3 topics introduced first? Also, the first 4 references are not cited.

POINT-BY-POINT ANSWERS TO THE REVIEWERS

Editor's comments

We have made a considerable effort including a high number of new experiments and analyses to address all questions raised by the reviewers. In particular, we have reasonably addressed the specific questions highlighted by the Editor:

1/ Concerns by referee #1 on *in vitro* experiments.

Answer: As detailed in the response to question #4 from referee #1 and to question #1 of referee #3, we have performed a substantial number of new *in vitro* experiments. The main results include:

- We used a well-characterized *in vitro* model of hepatocyte de-differentiation (HepaRG into HepaRG-tdHep cells) (Aninat C. et al. 2006, see references at the end of this document). Hepatocyte de-differentiation resulted in a rapid decline of HNF4A-P1 isoform expression along with upregulation of HNF4A-P2 isoforms. Hepatocyte-specific genes such as PCK1 and F7 were strongly downregulated, while genes related to EMT (eg. VIM) were upregulated (see figure in the specific response to the reviewer#1). These data strongly confirm our findings in primary hepatocytes, further reinforcing a role for HNF4A isoform dysregulation in hepatocyte de-differentiation.
- As suggested by reviewer #3 (question #1), we explored whether decreased HNF4A-P1 mediates P2 repression. First, we found that TGF β 1-induced P2 overexpression precedes HNF4A-P1 downregulation at both RNA and proteins levels (see Figure in the response to reviewer #3). Next, we found that knockdown of HNF4A-P1 did not result in the upregulation of HNF4A-P2 protein levels. Moreover, depletion of HNF4A-P1 did not affect TGF β 1- increased HNF4A-P2 protein. Altogether, these findings suggest that the stimulation of HNF4A-P2 protein expression by TGF β 1 is not mediated by HNF4A-P1 downregulation. Additionally, we explored the role of different TGF β 1-induced signaling pathways in P2 overexpression. We provide evidence that TGF β 1 signals through SMAD-dependent mechanisms to downregulate HNF4A-P1 and through SMAD-independent pathways to upregulate HNF4A-P2.
- All these new results and discussion have been incorporated into the revised version (**Fig 3, Supplementary Figure 8, Pages 11-14**).

2/ Concerns by referees #1 and 2 on methylomics analysis.

We agree with reviewers #1 and #2 on that the epigenetic data is preliminary. To address this question, we performed the following new analyses:

- To address question #4 from referee #1, in an unbiased fashion we studied the overall expression analysis of genes encoding epigenetic modulators in patients with alcoholic hepatitis (AH). We selected the top 5 hits of each family based on the differential expression comparing normal and AH patients. The main genes that were found markedly deregulated include HDAC 7, HDAC 11, PIWIL4 (MIWI2), NCOR2, ZBTB33, PRDM6, PCGF2 and PHC2. Moreover, we performed RNAseq from the same samples that were used for methylation chip and we analyzed the expression of the top differentially methylated genes that are targets of transcription factor deregulated in AH. There was a correlation between hypermethylation and gene downregulation and between hypomethylation and gene upregulation. We are currently investigating the functional relevance of DNA methylation in specific loci in AH, which we believe it is beyond the scope of this paper.
- To address question #5 raised by reviewer #2 (see tables in the specific response to the reviewer), we performed the following additional analyses:
 - Detection of SNP in CpG islands and in differentially methylated regions (DMR) near the HNF4A locus that are significantly associated with the development of AH.
 - Detection of SNP within/near HNF4A binding motifs globally found within CpG islands and DMRs.

These new analyses of the HNF4A locus do not support the hypothesis that genetic variation in differentially methylated regions is associated with the risk of developing severe AH.

All these new results and discussion have been incorporated into the revised version (**Supplementary Table 9 and pages 14-15**).

3/ To show that TGF β 1 acts on the P2 promoter (referee #3).

To address this question, we performed a number of additional experiments. See the detailed results in the response to the criticism #1 by referee #3. We provide evidence that: a) decreased HNF4A-P1 levels are not determinant for TGF β 1-mediated P2 induction, b) SMAD signaling as well as MEK/ERK pathway mediate the effect of TGF β 1 effect on P1 decrease c) HNF4A-P2 overexpression by TGF β 1 is SMAD-independent. d) Src-TAK pathway, and to a lesser extent, MEK/ERK mediate HNF4A-P2 overexpression. e) In the presence of TGF β 1, c-JUN binds HNF4A-P2 proximal intron 1. Further experiments including mutagenesis in a P2 promoter reporter and EMSA studies would be useful to further investigate P2 promoter regulation. All these new results and discussion have been incorporated into the revised version (**Fig. 3, Supplementary Fig 8, Pages 11-13**).

4/ To address the concerns by referee #4 on the lack of validation of the findings in animal models.

We agree with referees 1 and 4 on that the animal model was only briefly described in the manuscript, which was mainly due to space constraints. The development of a true model of AH is one of the most urgent unmet need in this field. Within the NIH-funded InTeam consortium, we have been working during the last 5 years to develop such model (Furuya et al 2016, Furuya et al 2018, see references at the end of this document). To reproduce the typical scenario in humans, mice with advanced fibrosis were challenged by heavy alcohol administration (i.e. CCl₄ for 9 weeks and then EtOH after a wash-up period). Although we found liver damage and pericellular (“chicken-wire”) fibrosis similar to the findings that we described in humans (Altamirano J. et al. 2014), our model did not show parameters indicative of liver failure. The finding that mice had a relatively preserved HNF4A-dependent gene expression could explain why they do not develop liver failure. It is therefore plausible that manipulating HNF4A could favor the development of alcohol-induced liver failure in these mice. We think that our results could be beneficial in developing a useful preclinical model in the near future. Moreover, to expand the information on this model, additional data on degree of steatosis, inflammation and fibrosis as well as liver function tests have been included in the revised manuscript (see figures on response #1 to referee #4). We also analyzed HNF4A isoform-specific targets by RNAseq in mice livers, as well as performed qPCR of selected genes. In summary, our animal model shows some features of AH by lack of liver failure, probably due to a preserved HNF4a dependent gene expression. These data could pave the way for future efforts to develop a useful preclinical model for this devastating disease for which new targeted therapies are urgently needed. These new results and comments have been added to the revised manuscript (**Results & Discussion pages 7 and 8, Supplementary Fig 3**).

Reviewer #1

Major concerns:

1. ***What is the reasoning of including the animal models in this report if the authors used only six lines (page 7) to describe a few observational findings and general statements? This part of the manuscript is out of place and findings in animal models deserve a greater in-depth investigation.***

Answer: We agree with the referee that the data obtained in the animal model was only briefly described in the manuscript, which was mainly due to space constraints. Because the development of a true model of alcoholic hepatitis (AH) is an urgent unmet need in this field, we think that these results can be relevant. It is well known ethanol itself does not cause advanced fibrosis and liver failure in mice. To overcome this point, we performed a model of acute-on-

chronic alcoholic liver disease in an attempt to reproduce the scenario in humans. For this purpose, mice with established cirrhosis were challenged by heavy alcohol administration (i.e. CCl₄ for 9 weeks and then EtOH after a wash-up period). Although we found liver damage and pericellular (“chicken-wire”) fibrosis similar to the findings that we described in humans (Altamirano J. et al, 2014, see references at the end of this document), our model did not show parameters indicative of liver failure (i.e. jaundice, coagulopathy). As explained below, the molecular characterization of these mice suggest that the relatively preserved HNF4A-dependent gene expression could partially explain why these mice do not develop liver failure. It is therefore plausible that manipulating HNF4A could favor the development of alcohol-induced liver failure in these mice. These results could be beneficial in developing a useful preclinical model.

To expand the information on this model, additional data on degree of steatosis, inflammation and fibrosis as well as liver function tests have been included in the revised manuscript. We analyzed HNF4A isoform-specific targets by RNAseq in mice livers, as well as performed qPCR of selected genes. Hnf4a mRNA levels were not dysregulated along disease progression across different animal models, confirming the results obtained in human samples (see Figure below). Interestingly, only a specific sub-group of HNF4A-P1 targets were downregulated in the acute-on-chronic alcohol-related liver injury model (i.e. Pck1, see Figure below), while other targets remained unchanged or upregulated. In contrast, some known HNF4A-P2 target genes in mice were upregulated. These results indicate a partial defective transcription of HNF4A-P1 in mice, along with transcriptional activation of HNF4A-P2 in mice with acute-on-chronic alcoholic liver injury. Our results suggest that more profound changes in the overall HNF4A transcription activity are probably required to develop liver failure. We are currently working on a long-term project to manipulate HNF4A to favor alcohol-induced liver failure in mice. We have modified Supplementary Figure 3 and extended our description of the animal model and the transcription factor dysregulation in these animals. The new results and comments have been added to the revised manuscript (**Results & Discussion pages 7 and 8, Supplementary Fig 3**).

Quantitative analysis of liver injury in alcohol- and fibrosis-associated mouse model. (A) Serum aminotransferase levels. (B) Liver injury score (see Methods). (C) Quantitative analysis of Oil RedO staining (five random fields at 200× magnification). (D) Triglyceride levels in liver tissue. (E) Quantitative analysis of Sirius Red staining (five random fields at 200× magnification). (F) Quantitative analysis of MPO-positive cell counts (five random fields at 200× magnification). (G) Representative liver sections from Hematoxylin-Eosin, Oil-Red-O and Sirius red stainings. (H) Detail of Sirius Red stained liver sections. All data are presented as mean±SD. Asterisks denote statistical significance as follows: a, p<0.05, compared to control group; b, p<0.05, compared to CCl₄(6w) group; c, p<0.05, compared to CCl₄(9w) group; d, p<0.05, compared to EtOH group.

Analysis of P1 and P2 isoforms and their targets in the CCL4+Ethanol animal model

12 week old C57Bl6J mice were treated with isocaloric diet + olive oil i.p. for 6 weeks (Control, N=3), intragastric ethanol + olive oil i.p. for 3 weeks (EtOH, N=3), isocaloric diet + CCL4 0.2 ml/kg i.p. for 6 weeks (CCL4 6W, N=2 for RNAseq and 3 for RT-PCR), isocaloric diet + CCL4 0.2 ml/kg i.p. for 9 weeks (CCL4 9W, N=2 for RNAseq and 3 for RT-PCR) or CCL4 0.2 ml/kg i.p. for 9 weeks + intragastric ethanol the last 3 weeks (CCL4 9W+EtOH, N=3). (a-f) Box Plot of RNA counts from liver RNA sequencing data. (a) Hnf4a levels, (b) HNF4A-P1 specific targets downregulated in mice treated with CCL4 and Ethanol. (c) HNF4A-P2 specific targets according to Vuong LM et al [REF] that were found upregulated in mice treated with CCL4 and Ethanol. (d) HNF4A-P1 specific targets not modified in mice treated with CCL4 and Ethanol. (e) HNF4A-P1 specific targets upregulated in mice treated with only ethanol 3 weeks. (f) Genes related to fibrosis (Col1a1, Lox), proliferation (Ccnb1), ductular reaction (Krt7) and macrophage infiltration marker (Cd68). (g) Real Time PCR of P1 and P2 dependent Hnf4a isoforms (h) Real Time PCR of HNF4A-P1 targets related to gluconeogenesis (Pck1), urea cycle (Otc) and clotting factor synthesis (F7).

2. The authors reported down-regulation of the lncRNA HNF-4A-As1 in the livers from patients with alcoholic hepatitis (Page 8). What is the role of this lncRNA?

Answer: We agree with the reviewer that this lncRNA could play a role in AH. In the original manuscript, we showed that HNF4A-AS1 lncRNA expression was decreased in patients with AH. This antisense lncRNA uses the same promoter region as the HNF4A adult isoform (i.e. P1 promoter) and may play a role in deregulated P1 and P2-dependent gene expression in AH. The sequence of the 3'-end of a recent annotation of HNF4A-AS1 overlaps to the 5' end of the exon 1E. Through *Cis* and/or *Trans* mechanisms, AS1 could recruit repressors (transcription factors, transcription factor interacting proteins, histone modifiers or DNA methylases) to the P2 or exon 1E regions and regulate the expression and/or the splicing. Silencing a lncRNA is challenging due to its low expression and nuclear localization. In order to address the question raised by the reviewer, we studied the expression of HNF4A-AS1 in three cell lines (HepG2, Hep3B and HepaRG). HNF4A-AS1 was expressed at low levels and we failed to silence HNF4A-AS1. Due to time constraints, it is not possible to provide new data using Locked Nuclear Acid GapmeR technology. Due to the sequence length and the fact that the 3' end is not well known, we would need to perform 3'-RACE, possibly from human liver samples, before deciding the specific sequence to be cloned for gain of function experiments.

In summary, we agree that further studies should evaluate the functional role of this intriguing lncRNA in AH. Due to the complexity of the experiments and time constraints, we think that this aim is beyond the scope of the study. We think that this lncRNA represents a potential target for therapy and that by keeping our data we can stimulate other investigators to explore its pathogenic role. These comments have been included in the revised manuscript (**Results & Discussion section, page 9**).

3. The in vitro part of the report requires substantial revision (pages 9-10). The authors attempted to study alterations of HNF4a isoforms during the de-differentiation of hepatocytes. They examined the expression of several genes in primary human hepatocytes in vitro. A simple culturing of primary hepatocytes is not a proper model to study hepatocyte de-differentiation. Additionally, altered expression of a few genes is not a sufficient evidence of de-differentiation.

Answer: We agree with the reviewer that a specific protocol of hepatocyte de-differentiation is needed to confirm the results obtained in primary hepatocytes. To address the point raised by the reviewer, we used a well-characterized model of hepatocyte de-differentiation of HepaRG into HepaRG-tdHep cells (Aninat C. et al. 2006). Hepatocyte de-differentiation resulted in a rapid decline of HNF4A-P1 isoform expression with a constant upregulation of HNF4A-P2 isoforms. Hepatocyte-specific genes such as PCK1 and F7 were strongly downregulated, while genes related to EMT (VIM) were upregulated (see figure below). Interestingly, the observed changes were gradual and did not show the plateau at 24h and 48h that was seen in primary hepatocytes. These data strongly confirm our findings in primary hepatocytes, further reinforcing a role for HNF4A isoform dysregulation in hepatocyte de-differentiation. These new results have been included in the revised manuscript (**page 10 and Fig. 2**).

HNF4A isoform regulation in HepaRG-tdHep model of hepatocyte retrodifferentiation

HepaRG cells were cultured for 2 weeks in 10% FBS Williams E medium and then for additional 2 weeks supplementing the media with 2% DMSO. After this period of differentiation, cells were plated at low confluence and switched to media without DMSO. Cells were collected at the indicated time points. Significance was determined by two-tailed Mann-Whitney U. *P < 0.05, **P < 0.01. Box-and-whisker plots indicate 25th–75th percentile; midline, median; whiskers, minimum to maximum values; individual data points are represented. Gene expression is presented as relative values normalized to the mean of the control.

4. The part of the manuscript focusing on epigenetic alterations describes few random epigenetic alterations only, i.e. altered expression of DNA methyltransferase and few selected histone acetyltransferase genes, the presence of differentially methylated regions without investigation of their functional consequences, analysis of one H3K27ac transcription-activating mark, each of the needed to be investigated in greater details. The inclusion of this part in the manuscript is preliminary.

Answer: We agree with the reviewer that the epigenetic data is somewhat preliminary. To address this question, we have performed the following new analyses:

- In an unbiased fashion, we studied the overall expression analysis of genes encoding epigenetic modulators in patients with AH. For this purpose, we used the EpiFactors database (Medvedeva YA. et al. 2015). This comprehensive database classifies epigenetic regulator genes in families based on their known function. We selected the top 5 hits of each family based on the differential expression comparing normal and AH patients. Main genes that were found markedly deregulated include HDAC 7, HDAC 11, PIWIL4 (MIWI2), NCOR2, ZBTB33, PRDM6, PCGF2 and PHC2 (**Figure 4d, page 17**).
- We obtained ChIP-seq data for other three histone marks: H3K4me1, H3K4me3 and H3K27me3. We reviewed the fold change and the significant enrichment comparing to the INPUT for the selected regions of interest, for each of the four histone modifications. Histone acetylation in lysine 27 was downregulated in the promoter P1 and in classic targets like PCK1 and CYP3A4, with similar findings for H3K4 mono-methylation, clearly indicating the loss of transcriptional activation. Interestingly, F7, a HNF4A target gene, showed increased trimethylation of H3K27, indicating an active inhibition of this genomic locus. This suggests the existence of several mechanisms of gene silencing in patients with AH. These results have been included in the revised manuscript (**Supplementary Figure 8, page 18**).
- We performed RNAseq from the same samples that were used for methylation chip and we analyzed the expression of the top differentially methylated genes that are targets of those transcription factor predicted to be inhibited or activated in AH. There was a correlation between hypermethylation and downregulation and between hypomethylation and upregulation. In the new version of Figure 4, we replaced the panel k to insert a series of representative genes to show this correlation. We are currently investigating the functional relevance of DNA methylation in specific loci in AH, which is beyond the scope of this paper. (**Figure 4k, page 17**).
-

Quantification of ChIP-seq fold increases for H3K27Ac, H3K27me3, H3K4me1 and H3K4me1. Gene expression of epigenetic modulators. Hyper and hypomethylated genes and its expression in the methylation chip cohort

(a) Peaks with a significant calling were annotated. Those significant peaks around the TSS were taken into account. Box plot of the fold increase respect to the INPUT for each ChIP for each one of the selected genes is presented. (b) Fold Change of epigenetic modulators in the differentially expressed genes when comparing Normal livers with AH. We have selected the top 5 genes in each family of EpiFactor database. (c) RNA was extracted from the same samples used for DNA methylation chip analysis. Top hyper and Top hypomethylated targets were reviewed. Most of them showed a trend their mRNA levels to be in agreement with the methylation status of their promoter regions, with some exceptions. In revised figure 4, we have included a selection of these genes for representative purposes. Significance was determined by two-tailed Mann-Whitney U. *P < 0.05. Box-and-whisker plots indicate 25th–75th percentile; midline, median; whiskers, minimum to maximum values; individual data points are represented. Gene expression is presented as relative values normalized to the mean of the control.

5. In addition to several dysregulated HNF4a-associated molecular pathways, it will be beneficial to explore the HNF4a-miR-122 axis alterations in alcoholic hepatitis

Answer: We think that this is a very interesting question, given that HNF4A is one of the main regulators of miR122 expression in hepatocytes through the hpri-miR-122 promoter (Li ZY. Et al. 2011). Although a definitive overview of the mechanistic implications of miR122 and HNF4A-miR122 axis in AH will require complex studies, we have performed the following analyses to address this request:

1. In our RNAseq analysis, we found that most patients with AH had low levels of liver miR122 (see figure below). The fact that HNF4A-P1 activity is suppressed in these patients suggests that it may play a role in decrease miR122 expression. Interestingly, patient with early ASH have reduced levels of miR122, indicating that alcohol itself is able to reduce miR122 expression. Patients with AH have increased levels of GRHL2 transcript (see figure below), a transcription factor that has been recently associated with miR122 inhibition in a mouse model of ethanol + CCl₄ mediated liver injury (Satishchandran A. et al 2018, see references at the end of the response to the reviewers) Patients with early ASH have slightly higher levels of GRHL2. There was a significant inverse correlation between these two genes. Whether or not this correlation denotes causal relationship requires further investigation.
2. We used Ingenuity Pathway Analysis knowledge base to determine the significance (p value) and direction of the functional enrichment (Z- Score) of all human miRNA in the differentially expressed genes between early ASH and AH. Importantly, miR122 was found to have the most significant negative Z-score (see figure below). HNF4A and miR122 depending genes are only partially overlapping, suggesting that miR122 dysfunction in the progression from early ASH to AH could involve HNF4A-independent pathways.
3. Finally, we used miRTarBase (Chou CH et al. 2018), a curated database of miRNA targets, to select the top 10 most validated miR122 targets in humans. In patients with AH, 9 of those top targets were upregulated while none of them were found increased in early ASH (see Figure below).

Although these new results support a potential role of miR122 in AH, further functional experiments are needed to confirm this hypothesis. These results are in accordance with a recent study suggesting role of miR122 in alcoholic liver disease (Satishchandran A. et al 2018). These new findings have been included in the revised manuscript (**pages 14-15 and Supplementary Fig.9**).

miR122 levels and predicted activation in AH patients

(a) RNA levels of hpri-miR122 in our cohort (b) levels of Grainyhead Like Transcription Factor 2 (GRHL2) in our cohort (c) correlation of GRHL2 and MIR122 levels. R and p value (Kendall) is presented. (d) Results of miRNA predicted activity by means of IPA Upstream Regulator analysis when comparing early ASH vs AH. Top 8 miRNA are presented. (e) Venn diagram of the overlap between HNF4A and MIR122 targets among the differentially expressed genes in the comparison between early ASH and AH. (f) Box plot of miR122 most 10 validated targets in humans according to miRTarBase database.Box-and-whisker plots indicate 25th–75th percentile; midline, median; whiskers, minimum to maximum values; individual data points are represented. In bold, those genes that reached FDR<10-6 level of significance in DESeq2 differential expression analysis.

Reviewer #2:

Argemi et al reported a comprehensive study on ALD. The paper is data rich and presented in a logical way. I have the following concerns.

1. The paper should be divided into sections, eg. discussion section.

Answer: We have modified the manuscript organization, following this suggestion.

2. It will be informative and more readable for audience if the authors can prepare a flowchart for all the analysis of this work (although Fig1a shows the brief flowchart for RNA-seq analysis)

Answer: We thank the reviewer for this thoughtful suggestion. We included a summarizing figure showing a flowchart of all the techniques and samples used in this study (Supplementary Figure 1) of the revised manuscript

Flowchart of analyses performed with Human Samples in the present work.

3. Abstract should present key data and results (eg. Sample size profiled).

Answer: We followed the guidelines for authors in Nat Commun, which suggest that abstracts should give a “brief non-technical summary of your main results and their implication”. The abstract word limit is 150. In order to address, as much as possible, the reviewer’s comments we have included a phrase better describing the phenotypes included in the GWAS study.

4. Many datasets are used throughout the paper, however, some conclusions/statements were made without specifying the dataset used. For example, “We then analyzed the methylation status of nearly 800,000 loci in normal livers and livers from AH patients and found around 3,000 differentially methylated (DM) CpG - containing loci (Fig. 4g and Supplementary Table 5)”. What was the sample size and significance level used in the differential methylation study?

Answer: As suggested by the reviewer, we have specified the dataset used in all OMIC studies throughout the manuscript. The DNA methylation chip analysis was performed by bisulfite treatment of DNA from 5 normal livers and 6 alcoholic hepatitis (AH) livers. Differentially methylated probes were identified by applying limma contrasts to M values (absolute change in beta value > 0.1, FDR-corrected Pvalue < 0.05). Differentially methylated regions were identified setting a threshold of absolute change in beta value in > 0.1 and of Stouffer’s value in < 0.05

(DMRCate). We have added these comments to the results section, including the sample size of ChIPseq and GWAS cohorts (**pages 6 and 7 of the revised manuscript**).

5. GWAS did not find signals around HNF4a, and the authors hypothesize the mechanism of HNF4a is via DNA methylation and chromatin remodeling. It is important to check eQTLs and methQTLs in liver for HNF4a locus.

Answer: The presence of SNPs in differentially methylated regulatory regions could be involved in dysregulation of HNF4A-dependent transcriptome. To address the point raised by the reviewer, we have done the following additional analyses:

A. Detection of SNP in CpG islands and in differentially methylated regions (DMR) near the HNF4A locus that are significantly associated with the development of AH

- **CpG islands:** Two annotated CpG islands near the HNF4a locus were identified using the UCSC human genome browser which contained SNPs from the AH GWAS dataset:
 - CpG 64 (Chr20:42939450-42940043) containing rs148377517. This SNP was not associated with the risk of developing severe AH (OR 0.62, 95% CI 0.214 – 1.802, P=0.3809).
 - CpG 29 (Chr20:42955433-42955728) containing rs13038786. This SNP was not associated with the risk of developing severe AH (OR 1.89, 95% CI 0.65 – 5.53, P=0.24).
- **DMR:** All the DMR previously found were used to identify differentially methylated regions around the HNF4a locus (defined as Chr20:42979340-43062485 but extended in either direction by c.50kb to Chr20:42900000- 43100000).
 - Five DMRs around HNF4a locus were identified:

Chr	Start	Stop	no.cp gs	minfdr	Stouffer	betaAfc
20	42983727	42984878	13	3.31E-131	3.21E-43	0.1457292
20	43013307	43013380	2	1.80E-12	3.69E-06	0.10592164
20	43020158	43020776	4	1.71E-71	2.08E-12	0.153716222
20	43024441	43025611	5	4.87E-48	1.35E-16	0.156084941
20	43028501	43029997	11	1.70E-127	2.74E-32	0.135593661
20	43078742	43080013	6	4.17E-58	8.51E-20	0.195183667

- SNPs located +/-1.5kb from the start and end of these DMRs were extracted from the AH GWAS dataset. In total, 20 SNPs fulfilled these criteria. None were associated with the development of severe AH:

SNP	BP	A1	OR	L95	U95	P
rs6031544	42982347	T	1.22	0.9496	1.567	0.1199
rs6031545	42982491	G	1.208	0.9403	1.552	0.1394
rs7347680	42982517	C	1.176	0.9162	1.508	0.2035
rs137931608	42982762	C	1.151	0.8249	1.605	0.4085
rs147258225	42982871	G	1.151	0.8249	1.605	0.4085
rs4810425	42982912	T	1.182	0.8681	1.609	0.2884
rs146481055	42982970	A	1.151	0.8249	1.605	0.4085
rs7272344	42983032	C	1.176	0.9162	1.508	0.2035
rs11508793	42983122	C	1.151	0.8249	1.605	0.4085
rs11508794	42983135	G	1.151	0.8249	1.605	0.4085
rs11508795	42983213	C	1.151	0.8249	1.605	0.4085
rs74749453	42983338	T	1.149	0.8238	1.603	0.4132
rs11508796	42983587	T	1.149	0.8238	1.603	0.4132
rs6031546	42985355	G	0.9276	0.7103	1.211	0.5811
rs2144908	42985717	A	1.208	0.8912	1.638	0.2234
rs2425635	43023841	A	0.8914	0.7093	1.12	0.3245
rs2425637	43024049	T	1.109	0.884	1.391	0.3714

rs112467286	43024585	A	0.8467	0.2807	2.554	0.7677
rs717247	43025784	C	0.9687	0.7589	1.236	0.7983
rs2868094	43026616	C	0.9575	0.7508	1.221	0.7267

B. Detection of SNP within/near HNF4A binding motifs globally found within CpG islands and DMRs.

- **CpG Islands:** MEME-ChIP output was used to identify differentially methylated (DM) CpG loci which lay in/near a HNF4a binding motif (CAAAGKBC) enriched locus.
 - 3,214 DM CpG loci containing HNF4A binding motifs were found.
 - SNPs lying +/-75bp from the locus of a DM CpG locus were extracted from the AH GWAS dataset. In total 505 SNPs fulfilled these criteria.
 - Of these, 18 demonstrated a potential association with disease (P<0.05), these are listed below.
 - Of note, the SNP rs942043 lies near the gene E2F3 which encodes a transcription factor whilst the variant rs846897 lies near the gene IGSF23, a member of the immunoglobulin superfamily. The functions of genes near other associated variants could not readily be linked with the potential development of severe AH.

SNP	CHR	BP	A1	OR	L95	U95	P
rs79481290	1	53884364	A	0.5151	0.3024	0.8775	0.01465
rs6693632	1	162648343	C	0.4605	0.2264	0.937	0.0324
rs112015044	1	175376029	A	1.504	1.008	2.246	0.04576
rs74898415	1	248111311	A	2.233	1.138	4.382	0.01945
rs12693080	2	176594989	C	1.444	1.097	1.9	0.008703
rs2742345	2	179568682	T	0.5174	0.2876	0.931	0.02791
rs2036980	3	145940283	C	1.463	1.096	1.953	0.00981
rs12504154	4	183986205	C	1.343	1.046	1.726	0.0209
rs233381	5	13580367	C	1.658	1.061	2.593	0.02649
rs6888163	5	74212755	G	0.5694	0.3926	0.8258	0.002988
rs942043	6	20428184	C	0.5705	0.3688	0.8826	0.0117
rs34023503	7	56144079	C	3.818	1.057	13.8	0.04095
rs12339639	9	97786816	T	0.6079	0.397	0.9308	0.02201
rs10513460	9	129339886	G	1.507	1.167	1.947	0.001683
rs80262520	11	97224826	A	0.3267	0.1101	0.9691	0.04373
rs2304471	16	8895909	A	0.6866	0.5111	0.9224	0.01254
rs17689455	16	79746335	A	1.43	1.113	1.837	0.005178
rs846897	19	45116933	A	0.7356	0.572	0.946	0.01672

- **DMR:** The loci of the 3,214 DM HNF4A binding motif-containing CpG loci were used to identify HNF4A binding motif-containing DMRs. In total, 328 DMRs were extracted.
 - SNPs lying +/-1.5kb from the locus of a differentially methylated region were extracted from the AH GWAS dataset.
 - In total 36 SNPs were identified which fulfilled this criteria. Only four variants, three of which were in perfect linkage disequilibrium, demonstrated a potential association with disease (P<0.05). These lay in the region Chr1:11899470-11092149 which lies within the coding region of a gene CLCN6 which encodes a chloride transporter.

SNP	CHR	BP	A1	OR	L95	U95	P
rs41275504	1	11902149	G	0.5515	0.3308	0.9193	0.02245
rs111479605	1	11899470	T	0.5666	0.3414	0.9404	0.02798
rs79429328	1	11899803	T	0.5666	0.3414	0.9404	0.02798
rs77785799	1	11900142	T	0.5666	0.3414	0.9404	0.02798

- In summary, these analyses appear to demonstrate the following:

1. The density of available SNP data within or near to either DMR or differentially methylated CpG dinucleotides was generally, sparse
2. Analysis of the HNF4A locus does not support the hypothesis that genetic variation in differentially methylated regions is associated with the risk of developing severe AH;
3. Attempts to assess whether genetic variation near to differentially methylated regions or dinucleotides containing/near to a putative HNF4A binding motif reveals a handful of SNPs some of which, including those lying near genes *E2F3*, *IGSF23* and *CLCN6*, demonstrated a possible association with developing disease. However, when viewed in the context of the number of tests performed (SNPs found) these associations are highly likely to represent false positives.

These new data and comments have been included in the revised manuscript (**Supplementary Table 9 and pages 14-15**).

Reviewer #3:

General:

1. **Summary and elsewhere – The authors show that TGFβ1 increases P2 protein expression and RNA, a very intriguing result. However, they have not shown that TGFβ1 acts directly on the P2 promoter. It could be that TGFβ1 decreases P1 expression, resulting in an increase in P2 expression. For example, Briancon et al JBC 2004 (PMID: 15159395) showed that P1-HNF4a represses the P2 promoter. The ambiguities in the mechanism responsible for the upregulation of P2-HNF4a TGFβ1 need to be clarified.**

Answer: We agree that the mechanisms of TGFβ1-increased P2 expression were not fully elucidated. We did a number of additional experiments to address this question.

- First, we analyzed the time-course of TGFβ1-induced changes in P1 and P2 expression. We found that TGFβ1-induced P2 overexpression precedes HNF4A-P1 downregulation. This was seen both at RNA and proteins levels (6 and 12h, respectively) (see Figure below, panels a-c). Next, we explored the effect of HNF4A-P1 silencing in TGFβ1-induced P2 overexpression. Knockdown of HNF4A-P1 using a specific siRNA did not result in the upregulation of HNF4A-P2 protein levels. Moreover, depletion of HNF4A-P1 did not affect TGFβ1- increased HNF4A-P2 protein overexpression (see Figure below, panel d). Collectively, these findings suggest that the stimulation of HNF4A-P2 protein expression by TGFβ1 is not mediated by HNF4A-P1 downregulation.

HNF4A-P1 decrease is not the main driver of P2 induction by TGFB1

(a-c) Hep3B cells treated with TGFB1, AREG or a combination (same as in old Figure 3e) (a) RNA levels of HNF4A-P1 and P2 isoforms. (b-d) Protein was extracted. Western Blot of HNF4A Isoforms at early (12h) (b) and late (24-48h) time points(c). (d) Hep3B cells were seeded and silenced with specific P1 siRNA. After 48h of silencing and 24h of TGFB1 treatment, cells were collected and Total protein was extracted. Western blot of HNF4A isoforms.

- Additionally, we explored the role of different TGFβ1-induced signaling pathways in P2 overexpression. In particular we explored whether TGFβ1 upregulates HNF4A-P2 expression through SMAD-dependent or SMAD-independent pathways. To address this point, we performed SMAD4 silencing using siRNA. SMAD4 is used by SMAD transcription factors to translocate to the nucleus and bind their target genes. SMAD4 silencing was efficiently obtained with both siRNAs. SMAD4 silencing did not result in a significant inhibition of TGFβ1-mediated HNF4A-P2 upregulation. Importantly, HNF4A-P1 downregulation by TGFβ1 was partially rescued by SMAD4 silencing (see figure below). These results suggest that TGFβ1 signals through SMAD-dependent mechanisms to downregulate HNF4A-P1 and through SMAD-independent pathways to upregulate HNF4A-P2).

SMAD4 inhibition improves HNF4A expression and function but not reduces TGFβ1-mediated HNF4A-P2 induction

HepG2 cells were seeded and silenced with two siRNAs against SMAD. After 48h of silencing and 24h of TGFB1 treatment cells were collected and RNA was extracted. (a) RNA levels of SMAD4 (b) RNA levels of HNF4A-P1 and P2 isoforms (c) levels of RNA of HNF4A targets PCK1 and OCT. Significance was determined by two-tailed Mann-Whitney U. *P < 0.05 compared to siScr, & P<0.05 compared to Non-TGFB. Box-and-whisker plots indicate 25th–75th percentile; midline, median; whiskers, minimum to maximum values; individual data points are represented. Gene expression is presented as relative values normalized to the mean of the control.

- We next aimed at identifying SMAD-independent pathways that mediate TGFβ1-induced HNF4A-P2 upregulation. In the original manuscript, we found that TAK inhibition markedly decrease this biological effect. C-Src mediates HNF4A-P1 downregulation induced by EGF. TGFβ1 can also activate c-Src. Moreover, TAK1 can mediate c-SRC downstream actions. Strikingly, c-SRC inhibitor PP2 completely abrogated the effect of TGFβ1 on HNF4a-P1 and P2 (figure below, panels a,b). We also incubated Hep3B cells with MEK/ERK inhibitor (UO126). MEK/ERK modulated the effect of TGFβ1 on HNF4A-P1 expression but not on HNF4A-P2 expression (see figure below, panels c and d).

c-SRC and MEK/ERK mediate TGFβ1-induced HNF4A deregulation

Hep3B cells were pre-treated with c-Src inhibitor PP2 (a), MEK/ERK inhibitor UO126 (c) and JNK inhibitor SP600125 (d) for 24h and treated overnight with TGFβ1 (5ng/ml).

- We finally performed ChIP-PCR experiments to assess the binding of transcription factors to the HNF4A-P2 promoter. C-JUN and other members of AP1 family of transcription factors are common downstream effectors of c-SRC and MEK-ERK. We first scanned promoter region of HNF4A-P2 (+/- 1kb from Transcription Start Site) searching for C-JUN transcription factor binding sites. We used JASPAR2018 Basic Sequence Analysis tool (Khan A et al 2018), selecting the matrix profiles for FOS:JUN heterodimer (MA0099.2 and MA0099.3) and for JUN (MA0488.1). We found 6 binding sites with high Relative Score (> 85%) (**Table below and Figure below, panel a**).

Matrix ID	Name	Score	Relative score	Start	End	Strand	Predicted sequence
MA0099.2	FOS::JUN	7.57	0.887	68	74	+	TGAATGA
MA0099.2	FOS::JUN	6.74	0.856	314	320	+	TCACTCA
MA0099.2	FOS::JUN	7.57	0.887	341	347	+	TGAATGA
MA0099.2	FOS::JUN	8.77	0.930	690	696	+	TGACACA
MA0099.2	FOS::JUN	6.65	0.853	1391	1397	+	TGGCTCA
MA0099.2	FOS::JUN	6.69	0.855	1717	1723	+	TTACTGA

We then designed primers to amplify two different regions (TSS-800 and TSS+350) containing C-JUN sites (Figure below, panel a). We also designed primers for a control region 6kb downstream the TSS and a positive control (GAPDH promoter). 10 Million Hep3B cells were used for each condition (No treatment and TGFβ1 5 ng/ml overnight). Cells were fixed and chromatin was sonicated and immunoprecipitated using anti-RNA Polymerase II (POL2) and anti-phospho-c-JUN(T91-93-95) antibodies. Isotype IgG was used as control. We found binding of POL2 to the control GAPDH promoter in both treated and untreated cells. Interestingly we found that POL2 binds a proximal intronic region

(TSS+338/+442) of HNF4A-P2, which contains a JUN::FOS TFBS, under TGFβ1 treatment (**Figure below, panel b**). Moreover, we found that c-JUN binds the same region under TGFβ1 treatment (**Figure below, panel c**).

C-JUN binds to canonical TFBS in HNF4A-P2 proximal intron 1 in response to TGFβ1 Hep3B cells were plated in p150 dishes at confluence and treated with TGFβ1 (5 ng/ml) for 12h. Cells were then fixed with formaldehyde 1% for 10 min and chromatin was extracted, sonicated and immunoprecipitated using c-Jun, Polymerase II and control isotype antibodies. (a) schematic representation of JUN binding sites in HNF4A-P2 promoter region and in proximal intron 1 and the location of primers used in ChIP-PCR. (b) ChIP-PCR using Polymerase II antibody. (c) ChIP-PCR using phospho-c-JUN antibody. The experiment was done in duplicate. FE: Fold Enrichment of % INPUT regarding Isotype IgG.

In summary, our results suggest that 1) the induction of HNF4A-P2 occurs 6 hours before the levels of HNF4A-P1 start to decline, and the silencing of HNF4A-P1 doesn't alter HNF4A-P2 levels at baseline or under TGFβ1 2) SMAD signaling as well as MEK/ERK pathway mediate the effect of TGFβ1 effect on P1 decrease 3) HNF4A-P2 overexpression by TGFβ1 is SMAD-independent. 4) Src-TAK pathway, and in a lesser extent, MAPKs, mediate HNF4A-P2 overexpression. 5) In the presence of TGFβ1, POL2 and c-JUN binds HNF4A-P2 proximal intron 1. We have added these results in the revised manuscript (**Figure 3, Pages 11-13**).

2. The authors conclude that PPAR γ agonists are partially preventive of the TGF β 1 effect, but that effect is not very convincing. Increase of P1 protein by PPAR γ agonists (ROSI and PIO) is not obvious from the blots Fig. in 3q nor is the rescue of the negative effect of TGF β 1 on P1. There is no quantification and it is not clear how reproducible this result is. Suppression of P2 by ROSI/PIO is clearer as are changes in RNA in the other panels but again some sort of quantification is needed.

Answer: We carefully quantified all western blots in the experiments done. Moreover, we repeated the experiments exposing Hep3B cells to TGF β 1, AREG (or the combination), in the presence or absence of Rosiglitazone and Pioglitazone. In this new set of experiments, Rosiglitazone clearly inhibited TGF β 1-mediated P1 downregulation. Even at baseline, Rosiglitazone increased HNF4A-P1 levels (**Figure below**). This confirms our previous results and indicates that Rosiglitazone also acts at the mRNA level. Conversely, in our repeated experiments, we could not see a strong effect induced by Pioglitazone. The revised version of Figure 3 has been modified accordingly (**Figure 3t, Page 14**).

Rosiglitazone rescues partially TGF β 1 and AREG mediated P1 downregulation
Hep3B cells were seeded and treated with TGF β 1 for 24h with the presence of Rosiglitazone. Western Blot of HNF4A isoforms. Quantification was normalized to GAPDH level for each condition. Relative values (Fold Changes) related to non-treated cells are here presented.

3. Down regulation of P1-HNF4a via EGF-like molecule AREG could be due to activation of Src kinase by EGF. Others have shown that Src (downstream of EGF) phosphorylates and subsequently down regulates the P1-HNF4a protein but not P2-HNF4a (Chellappa et al PNAS 2012 PMID: 22308320).

Answer: We performed new experiments treating cells with AREG and the c-Src inhibitor PP2. We found that AREG downregulates P1 protein levels in the presence of c-Src Inhibitor, suggesting that c-Src activation is not involved in AREG-mediated P1 reduction. We also repeated our experiment using recombinant EGF instead of AREG, using the same dose as described in Chellappa et al. As shown in the figure below, EGF downregulated P1 protein also in the presence of PP2 inhibitor, indicating that c-SRC is probably not the main mediator of EGF effect in hepatocyte cell lines. These discrepant results could be explained by cell differences (colon carcinoma cells vs hepatocyte cells lines). These results have been incorporated in the revised manuscript (**Supplementary Fig 7f, Page 13**)

c-SRC do not mediate AREG HNF4A-P1 inhibition.

Hep3B cells were pre-treated with c-Src inhibitor PP for 24h and treated overnight with AREG and EGF as indicated

4. The antisense HNF4a between P1 and P2 is examined in various panels but the relevance is not completely clear and it is not discussed. For example, Fig. 2c shows the down regulation of the AS and upregulation of P2-HNF4a, suggesting that P2-HNF4a might be repressing the AS. (P1-HNF4a does not appear to be affected in 2c.) If this is in fact the case, what role in AH do the authors propose for the AS?

Answer: We agree with the reviewer that this lncRNA could play a role in alcoholic hepatitis (AH). In the original manuscript, we showed that HNF4A-AS1 lncRNA expression was decreased in patients with AH. This antisense lncRNA uses the same promoter region as the HNF4A adult isoform (i.e. P1 promoter) and may play a role in deregulated P1 and P2-dependent gene expression in AH. The sequence of the 3'-end of a recent annotation of HNF4A-AS1 overlaps to the 5' end of the exon 1E. Through Cis and/or Trans mechanisms, AS1 could recruit repressors (transcription factors, transcription factor interacting proteins, histone modifiers or DNA methylases) to the P2 or exon 1D regions and regulate the expression and/or the splicing. Silencing this lncRNA is challenging due to its low expression and nuclear localization. In order to address the question raised by the reviewer, we studied the expression of HNF4A-AS1 in three of the cell lines (HepG2, Hep3B and HepaRG). HNF4A-AS1 was expressed at low levels and we failed to silence HNF4A-AS1. Due to time constraints, it is not possible to provide new data using Locked Nuclear Acid GapmeR technology. Due to the sequence length and the fact that the 3' end is not well known, we would need to perform 3'-RACE, possibly from human liver samples, before deciding the specific sequence to be cloned.

In summary, we agree that further studies should evaluate the functional role of this intriguing lncRNA in AH. Due to the complexity of the experiments and the time constraints, we think that this aim is beyond the scope of this study. We think that this lncRNA represents a potential target for therapy and that by keeping our data we can stimulate other investigators to explore its pathogenic role. These comments have been included in the revised manuscript (page 9).

5. Likewise, Fig. S4 – the predicted splicing events are interesting but the relevance to AH is not discussed.

Answer: As suggested, the relevance of these changes have been discussed in more detail in the revised manuscript. In particular, we have discussed how changes in exon 8 exclusion could impact AF-2 domain (Results & Discussion section, page 9). Patients with AH have in general less HNF4A exclusion events, which makes them have more HNF4A exon counts (Fig S4a) and higher correlation between exon 1D and the rest of exons (Fig S4b). When we studied exon exclusion phenomena in HNF4A transcript, we found less exclusion of exon 8 than in normal patients (**Supplementary Fig 4f**). Exon 8 codifies a portion of the Activation Domain 2 (AF-2) of HNF4A, which is closely related to transcriptional activity and protein turnover. We hypothesize that the lack of variability in this region could make HNF4A more susceptible to degradation in patients with AH. We have modified Fig S4, with a better representation of panel f.

Specific:

6. P.10, Fig. 3f-i – synergistic action via the TGFβ1RI/TAK1 – the TAK1 inhibitor does not affect P1 expression, although it does affect P2 – but p values for the appropriate comparisons are not given (TGFβ1 +/- TAK1 inhibitor) for P2 RNA (they look like they are significant, but this needs to be proven)

Answer: We have changed the panel, so results from cells untreated and treated with TAK1 inhibitor are shown. The differences did not reach significance (P=0.08) and is now presented in the graph.

7. Fig. S7 c-f. Which HNF4a is being probed for?

Answer: HNF4A-P1 isoform. We have modified the labelling of these WB.

8. The effect of AREG (EGF-like) on P2 expression is not very convincing (Figure 3e). This needs to be quantified? Also, the actin blots are oversaturated impeding proper

normalization. In contrast, in Fig. 3e and 3q the combined effect of TGFβ1 and AREG is substantial especially on P1-HNF4a but there is no mention of how the EGF and TGFβ1 pathway might be synergizing. Similarly, Fig. S7ab – TGFβ1+AREG synergistically decrease P1 but, unlike the statement on p. 10, in Fig. S7b does not show P2 RNA up in the presence of both. Rather it shows an increase only with TGFβ1.

Answer: We have quantified the experiments assessing the effect of AREG exposure and we performed some confirmatory western blots repeated again to have better housekeeping WBs (Fig 3d). We agree with the reviewer that, although there is a mild increase on P2 levels at early time points, AREG-induced effect on P2 induction is not significant. The combined administration of AREG and TGFβ1 synergistically downregulate P1. In contrast, AREG decreased TGFβ1-induced P2 up-regulation. We incorporated these new WBs in Figure 3 and changed the parts of the manuscript where we erroneously assumed a similar effect of AREG as that of TGFβ1 on P2 isoform induction. In addition, we have performed new experiments to explore the mechanisms underlying the synergistic effect by TGFβ1 and AREG on P1 downregulation. We tested the effect of MEK/ERK (UO126 inhibitor) and c-Src (PP2) inhibition on the effects of TGFβ1 and AREG on P1 levels. The results showed that MEK/ERK inhibition modulates the effects of both TGFβ1 and AREG, while C-Src inhibition only affect the response to TGFβ1 (see figure below). Therefore, both growth factors use common and differential pathways to repress P1 expression. Our results also suggest that MEK/ERK could participate in the additive/synergistic effect. These new results and comments have been included in the revised manuscript (**Supplementary Fig. 7 and page 13**).

Possible mechanisms of AREG and TGFβ1 synergy on HNF4A-P1 downregulation. Hep3B cells were pre-treated with a MEK/ERK inhibitor UO126, and a c-Src inhibitor (PP2) for 24h and treated overnight with AREG or TGFβ1 as indicated.

9. Fig. 1e shows that PPARγ is decreased in AH (second most downregulated TF after HNF4a) but this result is not discussed in the text. PPARγ does not go down in early ASH, when there is apparently no effect on HNF4a expression (Fig. 1e), but does decrease in AH, when there is increased P2-HNF4a (Fig. 1f). Could P2-HNF4a be down-regulating PPARγ activity??

Answer: To address this point, we overexpressed P2 in HepG2 cells and assessed the impact on PPARγ expression. Overexpression of P2 induced a slight, yet not significant, decrease in PPARγ. Conversely, we silenced P2 and assessed the effects on PPARγ expression. Knockdown of P2 expression induced a significant PPARγ overexpression. These results suggest that P2 could regulate PPARγ expression. Further studies should elucidate the mechanisms of P2 and PPARγ interdependence. These new results and comments have been included in the revised manuscript (**Fig. 3t and page 14**)

Silencing of HNF4A-P2 results in an increase of PPAR γ levels (a) HepG2 cells were transfected with a vector encoding for HNF4a2 (HNF4A-P1) and a8 (HNF4A-P2) isoforms; cells were collected 36h after transfection. RNA was extracted and expression of PPAR γ was measured by RT-PCR. (b) HepG2 cells were transfected with HNF4A-P2 siRNA for 48h and treated with TGFB1 for 12 and 24h.

10. Fig. 2m – an increase in P1 protein in the P2 KD is not apparent (the RNA effect is evident in 2n).

Answer: We have performed a new WB that show more clearly the effect of HNF4A-P2 silencing on HNF4A-P1 protein expression. These data have been included in the revised manuscript (Figures 2 and 3 and pages 11 and 13).

Silencing of HNF4A-P2 results in an increase of HNF4A-P1 levels HepG2 cells were transfected with HNF4A-P2 siRNA for 48h and treated with TGFB1 overnight. Nuclei were extracted using NE-PER kit (Thermo). (a) Western Blot was performed using specific N-terminal HNF4A antibodies (b) quantification of bands was performed using Li-COR Image Studio.

11. Fig. 2n suggests that P2-HNF4a represses P1 expression (but not the AS). Briancon et al JBC 2004 (PMID: 15159395) showed the reciprocal effect – namely, that P1-HNF4a represses the P2 promoter. This should be mentioned.

Answer: We did a number of additional experiments to address this question. In particular, we explored whether decreased HNF4A-P1 mediates P2 repression, as suggested by the quoted paper and the reviewer. First, we analyzed the time-course of TGFB1-induced changes in P1 and P2 expression. We found that TGFB1-induced P2 overexpression precedes HNF4A-P1 downregulation. This was seen both at RNA and proteins levels (6 and 12h, respectively) (see Figure below, panels a and b). Next, we explored the effect of HNF4A-P1 silencing in TGFB1-induced P2 overexpression. Knockdown of HNF4A-P1 using a specific siRNA did not result in the upregulation of HNF4A-P2 protein levels. Moreover, depletion of HNF4A-P1 did not affect TGFB1-induced HNF4A-P2 protein overexpression (see Figure below, panel d). Collectively, these findings suggest that the stimulation of HNF4A-P2 protein expression by TGFB1 is not mediated by HNF4A-P1 downregulation. These data have been included in the revised manuscript (Figure 3 and page 12).

HNF4A-P1 decrease is not the main driver of P2 induction by TGFB1

(a-c) Hep3B cells treated with TGFB1, AREG or a combination (same as in old Figure 3e) (a) RNA levels of HNF4A-P1 and P2 isoforms. (b-d) Protein was extracted. Western Blot of HNF4A Isoforms at early (12h) (b) and late (24-48h) time points(c). (d) Hep3B cells were seeded and silenced with specific P1 siRNA. After 48h of silencing and 24h of TGFB1 treatment, cells were collected and Total protein was extracted. Western blot of HNF4A isoforms.

12. Fig. 2r, 2s, p. 10 – synthesis and secretion of bile acids and glucose per se have not been examined. The authors have only measured the levels of these compounds in the primary hepatocytes – changes in those levels could be achieved by a number of different mechanisms.

Answer: We agree with the reviewer that bile acid concentration in the media could result from multiple steps. Almost all of the bile acids reabsorbed from the intestine are transported back to the liver where the hepatocyte re-conjugates them (Falany CN et al 1994, He D et al 2003). We measured glycochenodeoxycholate through mass (see figure below). Given that we do not supplement our media with glycochenodeoxycholic acid, the detection of glycine conjugated chenodeoxycholic acid in the culture media strongly suggests de-novo synthesis by hepatocytes. In the figure below, we show that the abundance of this particular conjugated bile acid is higher in P2 silenced primary human hepatocytes than in control cells. P2 silencing in HepG2 cells resulted in an increased expression of bile acid synthesis enzymes such as CYP7A1 as well bile acid transporters such as BSEP. This latter finding suggests that increased levels of bile acids in the media could be partially due to increase cell secretion. We have added these data and comments in the revised manuscript (**Supplementary Fig 7 Page 11**)

Glycochenodeoxycholate synthesis in hepatocytes HNF4A-P2 silenced Primary human hepatocytes were seeded at confluence and Transfected with siRNA anti-HNF4A-P2 and overlaid with Matrigel. Supernatant was collected at baseline, 8, 24 and 48h and samples were analyzed by Mass Spectrometry to measure Glycochenodeoxycholate.

Regarding glucose synthesis, Fig. 2r shows glucose concentrations in the media from primary hepatocyte cultures. No direct measurement of intracellular glucose was performed, as noted by the reviewer. Because hepatocytes were washed extensively before incubating in glucose-free media for this assay, the only possible source of glucose in the media was from hepatic glucose production. This production primarily reflects gluconeogenesis, since cells were washed and maintained overnight in low glucose media. This condition prior to the assay depletes hepatocytes of glycogen stores. The media used during the assay contained high concentrations of gluconeogenic substrates, primarily lactate, favoring gluconeogenesis (Yoon JC et al 2001, Madiraju AK et al 2014, see references at the end of this document). We have included these explanations and references in the methods and results section.

13. Fig. S6a – TCF3,4 and LEF1 are up in AH and AH explants; P2-HNF4a is also up. Others have reported that P1- and P2-HNF4a interact in a differential fashion with TCF4 in a colon cancer model (Vuong et al 2015 MCB, PMID: 26240283). The authors might want to see whether this paper is relevant to their story.

Answer: We appreciate this comment. This paper is very relevant. The RNAseq studies performed in our animal model of acute-on-chronic alcohol injury (CCl₄ + Ethanol treatment) showed predicted inhibition of TCF3 (which shares molecular and functional similarity with TCF4) (**Supplementary Fig 3**). We hypothesize that the fact that HNF4A is still active in these mice could be partially due to defective TCF3/4 repression activity over HNF4A. We have quoted this paper and added these comments in the revised manuscript (**Page 8**)

14. P. 11, Fig. 3l – need to show samples without TGFβ1 to show that “Hepatocytes transfection [sic – should be transfected] with siRNA targeting P2 isoforms abolished TGFβ1-mediated suppression of HNF4a-P1.”

Answer: As suggested, we have added the non-treated condition to the figure.

15. Supp Table 7 – catalog numbers for the antibodies must be given

Answer: Antibody references have been added to the Suppl table 7.

Editorial comments:

1. Abstract- last 2 sentences and elsewhere – “HNF4a-depending gene expression” should be “HNF4a-dependent (-driven) gene expression”

Answer: modified as suggested.

2. Some labels in the main figures are barely visible -- e.g., Fig. 3f-l – labels not very visible unless the pdf is zoomed in. In the paper version TGFbRI and TGFBRi cannot be easily distinguished. Fig. 4g labels are too small

Answer: modified as suggested.

3. Page 8- P2 HNF4a is introduced but P1 isoform is not described

Answer: we have described P1, as suggested.

4. Fig. 1b – text mentions NASH but the figure shows only NAFLD

Answer: the patients had NAFLD. We have changed the text as suggested.

5. Page 9 -“furtherly” ?

Answer: modified.

6. Fig. 1g, 1h – are not referenced in the text properly –Fig. 1g is referenced after Fig. 3o, Fig. 1h is not referenced at all

Answer: corrected as suggested.

7. Fig. S4d – label for “AH livers” is missing

Answer: we have added the label as suggested.

8. S4q – presentation is confusing. Means suggest more exon 8 in Control v. AH but AH ratio is 4.24

Answer: the figure was wrongly edited and has been corrected as suggested.

9. Fig. 2q – Cyp7A1 and 27A1 are not mentioned in the text

Answer: we have changed the text.

10. P. 10 -- “main” should be “potential” upstream regulators

Answer: we used “potential” as suggested.

11. Fig. S5d (RXR) is not discussed in the text

Answer: we have added an explanation of these results in the text.

12. P. 11 Fig. 3p does not show: “The PPAR-g agonists rosiglitazone and pioglitazone decreased the abundance of P2 isoforms and increased P1 isoforms (Fig. 3p).” Should be Fig. 3q?

Answer: modified.

13. Pg 11- “The effect of rosiglitazone on HNF4a-P1 mRNA levels was dose dependent (Fig. 3t)” -this is shown in 3u, not 3t

Answer: modified.

14. P. 11 – Fig. 3s should be 3t

Answer: modified

15. Fig. 3q – is there a reason for the HNF4a1-6 and 7-9 nomenclature used in the bottom blots instead of the P1- and P2-HNF4a as in the rest of the paper?

Answer: we have unified the nomenclature for HNF4A isoforms as suggested.

16. Pg. 11 PCK1, ALB mRNA panels in Fig 3t not referenced

Answer: we have referenced them in the text, as suggested.

17. Fig. S9d and e are not referenced

Answer: we have referenced them in the text, as suggested.

18. Fig. S10 is not referenced in the text

Answer: this Suppl Fig is referenced in methods section

19. Fig. S1 legend – not all p values are on the bottom left of the plots

Answer: modified as suggested.

20. Fig. S3 – red and blue should be defined. MEA should also be defined

Answer: we have added definition in the legend as suggested.

21. Fig. S4e – padj and Pearson's – up for Exon 1D compared to what?

Answer: The Pearson's correlation analysis was done comparing Exon 1D to itself ($p=0$, $R=1$) and to each one of the other exons.

22. Fig. 2 legend – in discussing genes, "that" should be used instead of "who"

Answer: modified as suggested.

Reviewer #4:

- 1. A weakness of this manuscript is that the major finding was not reproduced in the animal models of liver disease used in this study. Whereas earlier stages of ALD were recapitulated in the high fat diet/alcohol model with respect to activation of PPAR-gamma-dependent genes, later stages (modeled by alcohol/CCl₄) did not show inhibition of HNF4a-dependent gene expression. This observation is very briefly addressed by the authors but should be discussed more thoroughly.**

Answer: We fully agree with the reviewer. In fact, this criticism was also raised by reviewer #1. As explained to reviewer #1, data obtained in the animal model was only briefly described in the manuscript due to space constraints. The development of a true model of alcoholic hepatitis (AH) is one of the most urgent unmet need in this field. It is well known ethanol itself does not cause advanced fibrosis and liver failure in mice. To overcome this point, we performed a model of acute-on-chronic alcoholic liver disease in an attempt to reproduce the scenario in humans. For this purpose, mice with established cirrhosis were challenged by heavy alcohol administration (i.e. CCl₄ for 9 weeks and then EtOH after a wash-up period). Although we found liver damage and pericellular ("chicken-wire") fibrosis similar to the findings that we described in humans

(Altamirano et al, 2014, see references at the end of this document), our model did not show parameters indicative of liver failure (i.e. jaundice, coagulopathy and metabolic reprogramming). As suggested by this reviewer, the finding that mice had a relatively preserved HNF4A-dependent gene expression could explain why they do not develop liver failure. It is therefore plausible that manipulating HNF4A could favor the development of alcohol-induced liver failure in these mice. We think that these results could be beneficial in developing a useful preclinical model in the near future.

To expand the information on this model, additional data on degree of steatosis, inflammation and fibrosis as well as liver function tests have been included in the revised manuscript. We analyzed HNF4A isoform-specific targets by RNAseq in mice livers, as well as performed qPCR of selected genes. Hnf4a mRNA levels were not dysregulated along disease progression across different animal models, confirming the results obtained in human samples (see Figure below). Interestingly, only a specific sub-group of HNF4A-P1 targets were downregulated in the acute-on-chronic alcohol-related liver injury model (i.e. Pck1, see Figure below), while other targets remained unchanged or upregulated. In contrast, some known HNF4A-P2 target genes in mice were upregulated. These results indicate a partial defective transcription of HNF4A-P1 in mice, along with transcriptional activation of HNF4A-P2 in mice with acute-on-chronic alcoholic liver injury. Our results suggest that more profound changes in the overall HNF4A transcription activity are probably required to develop liver failure. We are currently working on a long-term project to manipulate HNF4A to favor alcohol-induced liver failure in mice. We have modified Supplementary Figure 3 and extended our description of the animal model and the transcription factor dysregulation in these animals. The new results and comments have been added to the revised manuscript (**pages 7-8, Supplementary Figure 3**).

Quantitative analysis of liver injury in alcohol- and fibrosis-associated mouse model. (A) Serum aminotransferase levels. (B) Liver injury score (see Methods). (C) Quantitative analysis of Oil RedO staining (five random fields at 200× magnification). (D) Triglyceride levels in liver tissue. (E) Quantitative analysis of Sirius Red staining (five random fields at 200× magnification). (F) Quantitative analysis of MPO-positive cell counts (five random fields at 200× magnification). (G) Representative liver sections from Hematoxylin-Eosin, Oil-Red-O and Sirius red stainings. (H) Detail of Sirius Red stained liver sections. All data are presented as mean±SD. Asterisks denote statistical significance as follows: a, $p < 0.05$, compared to control group; b, $p < 0.05$, compared to CCl4(6w) group; c, $p < 0.05$, compared to CCl4(9w) group; d, $p < 0.05$, compared to EtOH group.

Analysis of P1 and P2 isoforms and their targets in the CCL4+Ethanol animal model

12 week old C57Bl6J mice were treated with isocaloric diet + olive oil i.p. for 6 weeks (Control, N=3), intragastric ethanol + olive oil i.p. for 3 weeks (EtOH, N=3), isocaloric diet + CCL4 0.2 ml/kg i.p. for 6 weeks (CCL4 6W, N=2 for RNAseq and 3 for RT-PCR), isocaloric diet + CCL4 0.2 ml/kg i.p. for 9 weeks (CCL4 9W, N=2 for RNAseq and 3 for RT-PCR) or CCL4 0.2 ml/kg i.p. for 9 weeks + intragastric ethanol the last 3 weeks (CCL4 9W+EtOH, N=3). (a-f) Box Plot of RNA counts from liver RNA sequencing data. (a) Hnf4a levels, (b) HNF4A-P1 specific targets downregulated in mice treated with CCL4 and Ethanol. (c) HNF4A-P2 specific targets according to Vuong LM et al [REF] that were found upregulated in mice treated with CCL4 and Ethanol. (d) HNF4A-P1 specific targets not modified in mice treated with CCL4 and Ethanol. (e) HNF4A-P1 specific targets upregulated in mice treated with only ethanol 3 weeks. (f) Genes related to fibrosis (Col1a1, Lox), proliferation (Ccnb1), ductular reaction (Krt7) and macrophage infiltration marker (Cd68). (g) Real Time PCR of P1 and P2 dependent Hnf4a isoforms (h) Real Time PCR of HNF4A-P1 targets related to gluconeogenesis (Pck1), urea cycle (Otc) and clotting factor synthesis (F7).

2. Although the use of HepG2 and Hep3B cells is described in the Supplementary Materials and Methods and within the figure legends, it was not at all clear from reading the text of the manuscript that these cell lines were used. In fact, the way the cell culture experiments were described made it seem as though spontaneous de-differentiation of primary hepatocytes in culture was the only *in vitro* model employed. It should be made clear when, and for what purpose, HepG2 and Hep3B cells were used. The impact of the differentiation state of these cells should be discussed in the context of comparing results to those obtained with primary cells.

Answer: We have clarified the purpose of using specific cell types (primary vs cell lines) throughout the manuscript (**Pages 11-14**). For functional experiments such as the effect of P2 silencing on bile acid metabolism or glucose production, we mainly used primary hepatocytes. However, due to financial limitations and to ensure reproducibility, we mostly used two of the most well-characterized hepatocyte cell lines. We used Hep3B cells for experiments assessing effect of TGF β 1 and AREG on HNF4A-P1 and P2 levels, and HepG2 for experiments assessing the effect on HNF4A-P1 target genes. In our experience, HepG2 cells allow a better modulation of HNF4A downstream activity. We have performed the same experiments with Hep3B cells but the correlation between HNF4A-P1 expression and HNF4A-P1 activity (gluconeogenic, bile acid synthesis, clotting factor synthesis, etc.) was not always significant in this cell type. We observed that the more hepatocyte-like phenotype (i.e. HepG2) the more HNF4A-downstream effects we were able to document. On the other hand, the increase on HNF4A-P2 after treatment with TGF β 1 was more rapid and pronounced in Hep3B (6 fold) than in HepG2 (3 fold). This model allowed us to better interrogate TGF β 1-induced signaling in Hep3B cells. We agree with the reviewer that the differentiation state of cell lines is less modifiable. To overcome this limitation, we performed additional studies using an alternative *in vitro* model using retro-differentiated HepaRG (also discussed in reviewer #1 question #1). In this model using HepaRG-tdHep, we obtained similar results to those that were obtained with spontaneous hepatocyte de-differentiation, but in a more sequential fashion (see figure below). In addition, as the reviewer suggests, we have discussed in the Results and Discussion section, the potential impact of the differentiation state of the cells in the extrapolation of our results to primary human hepatocytes (**Pages 10-11**).

HNF4A isoform regulation in HepaRG-tdHep model of hepatocyte retrodifferentiation

HepaRG cells were cultured for 2 weeks in 10% FBS Williams E medium and then for additional 2 weeks supplementing the media with 2% DMSO. After this period of differentiation, cells were plated at low confluence and switched to media without DMSO. Cells were collected at the indicated time points. Significance was determined by two-tailed Mann-Whitney U. *P < 0.05, **P < 0.01. Box-and-whisker plots indicate 25th–75th percentile; midline, median; whiskers, minimum to maximum values; individual data points are represented. Gene expression is presented as relative values normalized to the mean of the control.

3. The increase in HNF4a-P2 (fetal) isoforms is seen only in patients with AH (Figure 3C). Notably, this increase was not seen in patients with compensated HCV-related cirrhosis. The authors cited a recent study showing that forced overexpression of a mature HNF4a isoform can reverse cirrhosis in CCl4-treated rats (Nishikawa 2015). The present findings should be discussed in the context of the Nishikawa paper, and the meaning of the observed restriction of the P2/P1 imbalance to AH patients only should be addressed.

Answer: we think this observation is very pertinent. We carefully reviewed the paper by Nishikawa et al, JCI 2015. In that study, rats with compensated cirrhosis had a relatively preserved HNF4a function, which was markedly downregulated in terminally decompensated cirrhosis. These results are in agreement with our human data showing that compensated HCV cirrhosis have preserved HNF4a-dependent gene expression. Of note, most patients with AH have decompensated disease along with liver failure, which makes AH a condition that resembles a terminally decompensated cirrhosis. Altogether, these human and rodent studies strongly suggest that HNF4a defective function is closely linked with the development of liver failure. Probably the fetal isoform is only re-expressed in conditions characterized by liver failure (i.e. ACLF, HA, etc), when HNF4A-dependent gene expression is clearly deficient. This hypothesis is supported by a recent study in different forms of advanced liver disease in humans (Guzman-Lepe J et al 2018). These comments have been included in the revised manuscript (**pages 19-20**).

4. An increase in the fetal isoforms of HNF4a may reflect an increase in proliferation that would be beneficial in terms of liver regeneration. Is there any indication that the loss of biological functions is accompanied by increased indicators of proliferation?

Answer: The reviewer raised a very important point. We have previously shown that AH livers are characterized by inefficient hepatic regeneration and impaired hepatocyte proliferation (Dubuquoy L et al 2015, Sancho-Bru P et al 2012). The fact that defective hepatic regeneration occurs in the setting of a massive P2 up-regulation suggest that P2 expression does not result in efficient hepatocyte proliferation. The molecular underpinning of defective hepatocyte proliferation in AH is largely unknown. In our study, transcription factors related with proliferation (SRF, SP1) were predicted to be activated in patients with AH, while other anti-proliferative factors (TP53) were also activated. In order to assess if P2 overexpression is part of a proliferative signaling pathway in injured hepatocytes, we performed additional experiments in cell lines and primary hepatocytes. A proliferation assay was performed in HepG2 and Hep3B cells as well in primary human hepatocytes transfected with siP2 or control siRNA. As shown in the Figure below, there were no differences between silenced cells and controls. We also measured P1 and P2 mRNAs in a model of liver regeneration after partial hepatectomy in mice that was recently published (Argemi J et al 2017). As previously described, IL6-KO mice display a defective regenerative response after partial hepatectomy due to liver necrosis and failure (Cressman DE 1996). We analyzed the expression of HNF4A isoforms in IL6 WT and KO mice at different time points after partial hepatectomy (2/3). Interestingly, impaired liver regeneration in IL6-KO was accompanied by increased levels of P2. We think these results are surprising and should be further studied. Overall, these results suggest that P2 does not play a major role in hepatocellular proliferation.

HNF4A-P2 and cell proliferation.

(a-c) Hep3B cells (a), HepG2 cells (b) and Primary Human Hepatocytes were transfected with siRNA anti HNF4A-P2. 6h after transfection, the thymidine analogue 5'ethynyl 2-deoxyuridine (EdU) was added to the culture medium. After 48h cells were treated with TGFβ1 (5 ng/ml) for additional 24h.

- Zhanxiang Zhou has shown that in animal models of ALD, the activity of HNF4a can be lost as a result of loss of zinc from the DNA binding domain zinc fingers. This post-translational modification is not seen by looking at mRNA or protein levels. The limitations associated with indirect measures of transcription factor activity should be addressed.**

Answer: we thank the reviewer for this interesting observation. We read the paper by Zhou Z et al, and we agree that loss of zinc from the DNA binding domain zinc fingers could play a role in defective HFN4a activity. In fact, it is well known that AH is associated with low zinc levels and there is mounting evidence that zinc supplementation exerts beneficial effects on experimental alcoholic liver disease. Based on the reviewer suggestion, we are planning to do studies in the near future correlating zinc levels and HNF4a activity. Moreover, we are currently performing a clinical trial in patients with AH within the NIH-funded AlcHepNet consortium including zinc supplementation. It is plausible that restoring HFN4A activity is one of the potential mechanisms of the beneficial effects of zinc supplementation in AH.

- Please include the catalog numbers for the antibodies listed in Supplemental Table 7.**

Answer: we have included them, as suggested

- In the description of Protein Extraction and Western Blotting in the Materials and Methods section, 2 clarification should be made. First, the units of the extract ratio are imprecise and should be changed to mg:microl or mg:L (whichever is correct). Second, it is unlikely that the “tissue was pestle and sonicated.” Correct as necessary.**

Answer: we have corrected the text.

8. **It is somewhat misleading to refer to 76 patients in groups of 9 to 18 as “a large series of patients” in the second paragraph of the manuscript.**

Answer: we have corrected the text.

9. **The first line of the manuscript is confusing. This isn't a paper about (primarily) addictions, mortality or cirrhosis, so why are those the 3 topics introduced first? Also, the first 4 references are not cited.**

Answer: We have corrected the text

We would like to thank the editors and the reviewers for their constructive criticisms and for the opportunity to resubmit a revisited version of our work. We hope you find the revised manuscript suitable for publication in Nature Communications.

Sincerely,

Ramon Bataller, MD, PhD
Associate Professor
Department of Gastroenterology Hepatology and Nutrition.
Biomedical Science Tower, BSTW1143
200 Lothrop St
University of Pittsburgh Medical Center
Pittsburgh, PA 15261
Phone: (919) 966-4812
Email: bataller@pitt.edu

References of Revision Letter (alphabetic order)

Aninat C, Piton A, Glaise D, Le Charpentier T, Langouët S, Morel F, Guguen-Guillouzo C, Guillouzo A. Expression of cytochromes P450, conjugating enzymes and nuclear receptors in human hepatoma HepaRG cells. *Drug Metab Dispos.* 2006 Jan;34(1):75-83. Epub 2005 Oct 4. PubMed PMID: 16204462.

Altamirano J, Miquel R, Katoonizadeh A, Abraldes JG, Duarte-Rojo A, Louvet A, Augustin S, Mookerjee RP, Michelena J, Smyrk TC, Buob D, Leteurtre E, Rincón D, Ruiz P, García-Pagán JC, Guerrero-Marquez C, Jones PD, Barritt AS 4th, Arroyo V, Bruguera M, Bañares R, Ginès P, Caballería J, Roskams T, Nevens F, Jalan R, Mathurin P, Shah VH, Bataller R. A histologic scoring system for prognosis of patients with alcoholic hepatitis. *Gastroenterology.* 2014 May;146(5):1231-9.e1-6. doi: 10.1053/j.gastro.2014.01.018. Epub 2014 Jan 15. PubMed PMID: 24440674

Argemí J, Kress TR, Chang HCY, Ferrero R, Bértolo C, Moreno H, González-Aparicio M, Uriarte I, Guembe L, Segura V, Hernández-Alcoceba R, Ávila MA, Amati B, Prieto J, Aragón T. X-box Binding Protein 1 Regulates Unfolded Protein, Acute-Phase, and DNA Damage Responses During Regeneration of Mouse Liver. *Gastroenterology.* 2017 Apr;152(5):1203-1216.e15. doi: 10.1053/j.gastro.2016.12.040. Epub 2017 Jan 9. PubMed PMID: 28082079.

Chou CH, Shrestha S, Yang CD, Chang NW, Lin YL, Liao KW, Huang WC, Sun TH, Tu SJ, Lee WH, Chiew MY, Tai CS, Wei TY, Tsai TR, Huang HT, Wang CY, Wu HY, Ho SY, Chen PR, Chuang CH, Hsieh PJ, Wu YS, Chen WL, Li MJ, Wu YC, Huang XY, Ng FL, Buddhakosai W, Huang PC, Lan KC, Huang CY, Weng SL, Cheng YN, Liang C, Hsu WL, Huang HD. miRTarBase update 2018: a resource for experimentally validated microRNA-target interactions. *Nucleic Acids Res.* 2018 Jan 4;46(D1):D296-D302. doi: 10.1093/nar/gkx1067. PubMed PMID: 29126174; PubMed Central PMCID: PMC5753222.

Cressman DE, Greenbaum LE, DeAngelis RA, Ciliberto G, Furth EE, Poli V, Taub R. Liver failure and defective hepatocyte regeneration in interleukin-6-deficient mice. *Science.* 1996 Nov 22;274(5291):1379-83. PubMed PMID: 8910279.

Dubois-Pot-Schneider H, Fekir K, Coulouarn C, Glaise D, Aninat C, Jarnouen K, Le Guével R, Kubo T, Ishida S, Morel F, Corlu A. Inflammatory cytokines promote the retrodifferentiation of tumor-derived hepatocyte-like cells to progenitor cells. *Hepatology.* 2014 Dec;60(6):2077-90. doi: 10.1002/hep.27353. Epub 2014 Oct 29. PubMed PMID: 25098666.

Dubuquoy L, Louvet A, Lassailly G, Truant S, Boleslawski E, Artru F, Maggioro F, Gantier E, Buob D, Leteurtre E, Cannesson A, Dharancy S, Moreno C, Pruvot FR, Bataller R, Mathurin P. Progenitor cell expansion and impaired hepatocyte regeneration in explanted livers from alcoholic hepatitis. *Gut.* 2015 Dec;64(12):1949-60. doi: 10.1136/gutjnl-2014-308410. Epub 2015 Mar 2. PubMed PMID: 25731872; PubMed Central PMCID: PMC4558407.

He D, Barnes S, Falany CN. Rat liver bile acid CoA:amino acid N-acyltransferase: expression, characterization, and peroxisomal localization. *J Lipid Res.* 2003 Dec;44(12):2242-9. Epub 2003 Sep 1. PubMed PMID: 12951368.

Falany CN, Johnson MR, Barnes S, Diasio RB. Glycine and taurine conjugation of bile acids by a single enzyme. Molecular cloning and expression of human liver bile acid CoA:amino acid N-acyltransferase. *J Biol Chem.* 1994 Jul 29;269(30):19375-9. PubMed PMID: 8034703.

Furuya S, Cichocki JA, Konganti K, Dreval K, Uehara T, Katou Y, Fukushima H, Kono H, Pogribny IP, Argemi J, Bataller R, Rusyn I. Histopathological and molecular signatures of a mouse model of acute-on-chronic alcoholic liver injury demonstrate concordance with human alcoholic hepatitis. *Toxicol Sci.* 2018 Dec 4. doi: 10.1093/toxsci/kfy292. PubMed PMID: 30517762.

Furuya S, Chappell GA, Iwata Y, Uehara T, Kato Y, Kono H, Bataller R, Rusyn I. A mouse model of alcoholic liver fibrosis-associated acute kidney injury identifies key molecular pathways. *Toxicol Appl Pharmacol*. 2016 Nov 1;310:129-139. doi: 10.1016/j.taap.2016.09.011. Epub 2016 Sep 15. PubMed PMID: 27641628; PubMed Central PMCID: PMC5323078.

Guzman-Lepe J, Cervantes-Alvarez E, Collin de l'Hortet A, Wang Y, Mars WM, Oda Y, Bekki Y, Shimokawa M, Wang H, Yoshizumi T, Maehara Y, Bell A, Fox IJ, Takeishi K, Soto-Gutierrez A. Liver-enriched transcription factor expression relates to chronic hepatic failure in humans. *Hepatology Commun*. 2018 Mar 23;2(5):582-594. doi: 10.1002/hep4.1172. eCollection 2018 May. PubMed PMID: 29761173; PubMed Central PMCID: PMC5944584.

Khan A, Fornes O, Stigliani A, Gheorghe M, Castro-Mondragon JA, van der Lee R, Bessy A, Chêneby J, Kulkarni SR, Tan G, Baranasic D, Arenillas DJ, Sandelin A, Vandepoele K, Lenhard B, Ballester B, Wasserman WW, Parcy F, Mathelier A. JASPAR 2018: update of the open-access database of transcription factor binding profiles and its web framework. *Nucleic Acids Res*. 2018 Jan 4;46(D1):D260-D266. doi:10.1093/nar/gkx1126. PubMed PMID: 29140473; PubMed Central PMCID: PMC5753243

Li ZY, Xi Y, Zhu WN, Zeng C, Zhang ZQ, Guo ZC, Hao DL, Liu G, Feng L, Chen HZ, Chen F, Lv X, Liu DP, Liang CC. Positive regulation of hepatic miR-122 expression by HNF4 α . *J Hepatol*. 2011 Sep;55(3):602-611. doi: 10.1016/j.jhep.2010.12.023. Epub 2011 Jan 15. PubMed PMID: 21241755.

Madiraju AK, Erion DM, Rahimi Y, Zhang XM, Braddock DT, Albright RA, Prigaro BJ, Wood JL, Bhanot S, MacDonald MJ, Jurczak MJ, Camporez JP, Lee HY, Cline GW, Samuel VT, Kibbey RG, Shulman GI. Metformin suppresses gluconeogenesis by inhibiting mitochondrial glycerophosphate dehydrogenase. *Nature*. 2014 Jun 26;510(7506):542-6. doi: 10.1038/nature13270. Epub 2014 May 21. PubMed PMID: 24847880; PubMed Central PMCID: PMC4074244.

Medvedeva YA, Lennartsson A, Ehsani R, Kulakovskiy IV, Vorontsov IE, Panahandeh P, Khimulya G, Kasukawa T; FANTOM Consortium, Drabløs F. EpiFactors: a comprehensive database of human epigenetic factors and complexes. *Database (Oxford)*. 2015 Jul 7;2015:bav067. doi: 10.1093/database/bav067. Print 2015. PubMed PMID: 26153137

Sancho-Bru P, Altamirano J, Rodrigo-Torres D, Coll M, Millán C, José Lozano J, Miquel R, Arroyo V, Caballería J, Ginès P, Bataller R. Liver progenitor cell markers correlate with liver damage and predict short-term mortality in patients with alcoholic hepatitis. *Hepatology*. 2012 Jun;55(6):1931-41. doi: 10.1002/hep.25614. Epub 2012 Apr 23. PubMed PMID: 22278680.

Satishchandran A, Ambade A, Rao S, Hsueh YC, Iracheta-Vellve A, Tornai D, Lowe P, Gyongyosi B, Li J, Catalano D, Zhong L, Kodys K, Xie J, Bala S, Gao G, Szabo G. MicroRNA 122, Regulated by GRLH2, Protects Livers of Mice and Patients From Ethanol-Induced Liver Disease. *Gastroenterology*. 2018 Jan;154(1):238-252.e7. doi: 10.1053/j.gastro.2017.09.022. Epub 2017 Oct 4. PubMed PMID: 28987423; PubMed Central PMCID: PMC5742049.

Yoon JC, Puigserver P, Chen G, Donovan J, Wu Z, Rhee J, Adelmant G, Stafford J, Kahn CR, Granner DK, Newgard CB, Spiegelman BM. Control of hepatic gluconeogenesis through the transcriptional coactivator PGC-1. *Nature*. 2001 Sep 13;413(6852):131-8. PubMed PMID: 11557972.

Reviewers' Comments:

Reviewer #1:

Remarks to the Author:

Authors conducted new experiments and provided comprehensive answers to my comments and suggestions.

Reviewer #2:

Remarks to the Author:

1) The authors added extensive experiments to address all the critics related to mechanism part.

2) The paper combines results and discussion sections. I guess the authors think this way may make the paper more readable.

3) The authors did not answer the eQTL and methylation QTL near HNF4a locus directly but showed substantial evidence in another way and it seems that the global epigenetic change is likely to be a major mechanism in dysregulation of HNF4a-dependent target genes in the development of AH.

Reviewer #3:

Remarks to the Author:

The authors have done an excellent job at responding to my concerns

Reviewer #4:

Remarks to the Author:

The authors have sufficiently addressed the list of recommendations by all the reviewers. Thus, I recommend publication.

Point-by-point Response to the Reviewers

REVIEWERS' COMMENTS:

Reviewer #1 (Remarks to the Author): Authors conducted new experiments and provided comprehensive answers to my comments and suggestions.

Reviewer #2 (Remarks to the Author): 1) The authors added extensive experiments to address all the critics related to mechanism part. 2) The paper combines results and discussion sections. I guess the authors think this way may make the paper more readable. 3) The authors did not answer the eQTL and methylation QTL near HNF4a locus directly but showed substantial evidence in another way and it seems that the global epigenetic change is likely to be a major mechanism in dysregulation of HNF4a-dependent target genes in the development of AH.

Reviewer #3 (Remarks to the Author): The authors have done an excellent job at responding to my concerns

Reviewer #4 (Remarks to the Author): The authors have sufficiently addressed the list of recommendations by all the reviewers. Thus, I recommend publication.